



# SnowPappus v1.0, a blowing-snow model for large-scale applications of Crocus snow scheme

Matthieu Baron[1,2], Ange Haddjeri[1], Matthieu Lafaysse[1], Louis Le Toumelin[1], Vincent Vionnet[3], and Mathieu Fructus[1]

[1]Univ. Grenoble Alpes, Université de Toulouse, Météo-France, CNRS, CNRM, Centre d'Études de la Neige, Grenoble, France
[2]Univ. Grenoble Alpes, Univ. Savoie Mont Blanc, CNRS, LECA, Grenoble, France
[3]Meteorological Research Division, Environment and Climate Change Canada, Dorval, QC, Canada

**Correspondence:** matthieu.baron@univ-grenoble-alpes.fr

**Abstract.** Wind-induced snow transport has a strong influence on snow spatial variability especially at spatial scales between 1 and 500 m in alpine environments. Thus, the evolution of snow modelling systems towards 100-500 m resolutions requires representing this process. We developed SnowPappus, a parsimonious blowing snow model coupled to the Crocus state-of-the-art snow model, able to be operated over large domains and entire snow seasons. SnowPappus simulates blowing snow

occurrence, horizontal transport flux and sublimation rate on each grid cell as a function of 2D atmospheric forcing and snow surface properties. Then, it computes a mass balance using an upwind scheme to provide eroded or accumulated snow amounts to Crocus. Parameterizations used to represent the different processes are described in detail and discussed against existing literature. A point-scale evaluation of blowing snow fluxes was conducted, mainly at the *Col du Lac Blanc* observatory in French Alps. Blowing snow occurrence evaluation showed SnowPappus performs as well as a currently operational scheme.

Evaluation of the simulated suspension fluxes highlighted a strong sensitivity to the suspended particles terminal fall speed. Proper calibrations allows the model to reproduce the correct order of magnitude of the mass flux in the suspension layer. Numerical performances of gridded simulations of Crocus coupled with SnowPappus were assessed, showing the feasibility of using it for operational snow forecast at the scale of the entire French Alps.

## 1  Introduction

Mountainous areas in temperate regions usually experience a seasonal snowpack. Its physical properties, depth and persistence influence many local processes such as surface energy balance, soil temperature and vegetation productivity (Choler, 2015). They are critical to forecast and anticipate snow-related hazards, especially avalanche triggering (Schweizer et al., 2003; Morin et al., 2020a). At a larger scale, snow melt-out is an important source of water for downflow hydrological catchment, with impacts on water availability for agriculture and ecosystems, human consumption and hydropower (IPCC, 2022). Besides,

topographic complexity of mountainous environment promotes a huge snowcover spatial variability. Variations in elevation and aspect, by influencing air temperature and radiative incoming fluxes are major predictors of this variability at very small to large scales. However these simple patterns are complexified by interaction between wind flow, precipitation patterns and various



post-depositional processes (Mott et al., 2018). These interactions, among other phenomena, include orographic effects which tend to enhance precipitation on the windward side of mountain ranges (at a scale 10-100 km), interaction between wind flow

and cloud formation processes and preferential deposition of snowfall at smaller scales (from dozens of meters to kilometers). Finally, at scales of meters to hundreds of meters, post-depositional processes, and in particular wind-induced snow transport have a big influence on snow depth and properties. This variability has consequences on the above-mentioned processes, and thus must be taken into account when studying them.

During the last 40 years, numerous models have been developed to simulate snowpack evolution. They range from simple

1-layer models, often used in Global Climate Modelling or Numerical Weather Prediction (NWP) models to detailed multilayer snowpack models explicitly representing processes like snow metamorphism, compaction, etc. These detailed models include Crocus (Vionnet et al., 2012), SNTHERM (Jordan, 1991), SNOWPACK (Bartelt and Lehning, 2002). Crocus has been used for large-scale applications (Vernay et al., 2022; Vionnet et al., 2016) but only at a very coarse resolution, which prevents representing adequately snow spatial variability. In particular it is currently used operationally for avalanche hazard forcasting

in French mountains at a massif range scale. High resolution applications including wind-induced snow transport were limited to very small domains and/or short periods of time (Vionnet et al., 2014) due to computational costs and limited availability of high resolution forcing data. However, growing computational power paves the way for moving large-scale operational systems towards resolutions of a few hundreds of meters, also sustained by the perspective of assimilation of promising high-resolution observations (Deschamps-Berger et al., 2022). It requires to represent phenomena driving snowpack variability at

this scale, including pre-depositional processes, which are increasingly represented in non-hydrostatic atmospheric models up to kilometric resolution and post-depositional processes such as wind-induced snow transport, which can be represented within the snowpack scheme.

Regarding wind-induced snow transport, various modelling approaches have also been developed for mountainous terrain including a fully explicit snow-atmosphere coupling (Vionnet et al., 2014; Sharma et al., 2021). However, this approach is not

affordable in terms of numerical cost on large temporal and spatial scales, explaining the use of much simpler parameterizations in snow hydrology applications (Bowling et al., 2004; Pomeroy et al., 2007; MacDonald et al., 2009), associated with more simple snow models than Crocus or SNOWPACK. In the above-mentioned context of increasing resolution of snow modelling systems, a numerically efficient representation of wind-induced snow transport that can be coupled to Crocus simulations is currently missing while this is a necessary condition to better account for the impact of snow transport in avalanche forecasting

over French mountains.

The goal of this paper is to present a novel blowing snow scheme, SnowPappus, coupled to Crocus in order to carry out simulations at the scale of the French Alps at a resolution of 250 m. Section 2 gives an overview of the modelling state of the art for the different involved processes in order to justify the methodological choices selected for SnowPappus and finally present model equations. Section 3 presents the methods used to run and evaluate simulations. Section 4 and 5 finally presents

and discusses the results.





## 2 Methodological choices

### 2.1 Target, opportunities and constraints

In this article, we present a wind-induced snow transport scheme designed to be coupled to Crocus snow model, part of SURFEX v9 land surface scheme (Masson et al., 2013), and to be used in 250 meters-resolution snowpack simulations at the scale of all French mountains (approximately 50 000 $km^2$ for French Alps) on entire seasons. The long-term operational purpose is to associate this modelling with a data assimilation framework probably requiring ensemble runs of 50-100 members (Largeron et al., 2020; Cluzet et al., 2021). The 250 m resolution has been chosen as a trade-off between the necessity of having a good representation of terrain parameters influencing mass and energy balance of the snowpack (slopes and aspects), and the expected computational cost. In order to achieve this goal using reasonable numerical resources, the transport scheme shall not be much more computationally intensive than the Crocus snow model itself. In terms of temporal and spatial scales, one must first consider that the typical time step of Crocus is 900 seconds. Then, meteorological forcings must be downscaled from kilometer scale NWP systems, as running operationally higher resolution atmospheric models is currently too costly for any weather service. Even if NWP models provide 3D atmospheric variables, the use of downscaling techniques forces us to use 2D wind fields as an input for our model (see Sect. 3.2).

These temporal and spatial scales influence methodological choices for model development, which are presented in details in the following sections. The processes and state variables represented in Crocus also impact the methodological choices. Crocus (Vionnet et al., 2012; Carmagnola et al., 2014) is a detailed multilayer snow scheme in which each snow layer is characterized by its mass, density, age, liquid water content, a historical variable stating if the layer has experienced liquid water in the past and microstructural properties: the optical diameter $D_{opt}$ and the sphericity $s$. These properties evolve in time by the representation of all main physical processes (heat diffusion and phase changes in relation with each layer energy budget, metamorphism, liquid water percolation, compaction). The ability of Crocus to distinguish different snow types at the surface may be an opportunity for the simulation of snow transport (Guyomarc'h and Mérindol, 1998; Lehning et al., 2000).

### 2.2 Blowing snow occurrence

#### 2.2.1 Theoretical background and previously used parameterizations

Transport is initiated when the fluid shear stress exerted near the surface exceeds the weight of the grains and their cohesion force (Schmidt, 1980) which occurs above a threshold wind speed. Grains can also be ejected by the impact of transported grains ("splash" process)(Comola and Lehning, 2017). Then, two threshold wind speeds can be defined: a threshold wind of transport initiation $U_i$ and a threshold wind speed necessary to maintain transport $U_e$. $U_i$ is expected to be higher than $U_e$ due to the splash process. However, in the case of snow, no systematic relationship could be found between $U_i$ and $U_e$ (Castelle et al., 1994; Michaux, 2003). It could be due to the evolution of snow microscale properties during transport, or to the fact that snow transport should necessarily be initiated by particle impacts, as cohesion forces are too strong to be broken by the wind alone (Schmidt, 1980). Therefore, in the case of snow, a single threshold value $U_t$ is commonly used.





The threshold wind speed varies strongly as a function of snow properties, ranging from 4 $m.s^{-1}$ for freshly fallen snow to 15 $ms^{-1}$ or even more for old refrozen or wet snow (Li and Pomeroy, 1997; Guyomarc'h and Mérindol, 1998; Clifton et al.,
2006). Different formulations have been used to assess the threshold wind speed. Following field observations in Saskatchewan plains, Li and Pomeroy proposed a parameterization depending only on temperature, arguing that most cohesion forces as capillarity, sintering and mechanical properties of snow are influenced by temperature.However this model only accounts for a small part of the variability of observed threshold wind speed (Li and Pomeroy, 1997). This formulation is used in Prairie Blowing Snow Model (PBSM) (Pomeroy et al., 1993) and in Marsh et al. (2020) model.

A second formulation is used in the SnowTran3D model (Liston et al., 2007), relating threshold wind speed with snow density. It covers a wider range of threshold wind speeds than the previous one, but was not extensively compared with field experiments.

Finally, snowdrift models coupled with detailed snow models simulating microstructural snow properties such as MAR (Gallée et al., 2001), Crocus-Sytron (Vionnet et al., 2018) and Crocus-MesoNH (Vionnet et al., 2014) use a formulation of
threshold wind speeds based on these microscale properties (dendricity, sphericity and grain size), proposed by Guyomarc'h and Mérindol (1998) and evaluated against blowing snow occurrence data collected at *Col du Lac Blanc* in French alps (Guyomarc'h and Mérindol, 1998), with different modifications depending on the considered model. The Alpine-3D model (Raderschall et al., 2008) also includes a dependency of the threshold wind speed on snow microstructure properties as proposed by Schmidt (1980).

Parameterizations based on snow surface properties (density or microstructure properties) seem to be the most promising in terms of blowing snow occurence, as they generate a wide range of threshold wind speeds (typically from 4 to 12 m s$^{-1}$ in the parameterization of Guyomarc'h and Mérindol (1998)) allowing to distinguish for example fresh and old snow case. However, they are consequently very sensitive to snow surface properties simulated by the underlying snow model (Crocus in our case) while near-surface snow density is highly uncertain (Helfricht et al., 2018) and microstructure properties can not be
directly evaluated as dendricity, sphericity and grain size are not observable variables. Therefore, a systematic bias in the model may be compensated by the threshold wind speed parameterization, limiting the possible transfer of such parameterization to a different snow model than the one for which it was developed. It leads us to choose the parameterization of Guyomarc'h and Mérindol for SnowPappus, which has been extensively tested with Crocus (Vionnet et al., 2013, 2018).

### 2.2.2   Atmospheric boundary layer

As two-dimensional wind fields are more commonly available from NWP outputs than 3d wind fields, for simplicity, we assume a neutrally stable and stratified flow, with the well-known logarithmic wind speed profile

$$U(z) = \frac{u_*}{k} \ln(\frac{z}{z_0}) \tag{1}$$

with $U$ the horizontal wind speed (m s$^{-1}$), $u_*$ the wind friction velocity (m s$^{-1}$) and $z$ the height above snow surface (m), $k$ the Von Karman's constant (dimensionless), found empirically to be equal to 0.41 and $z_0$ the roughness length of the surface (m).
We expect wind forcings to be given as a wind speed at a reference height $z_{\text{forc}}$. We deduce $u_*$ from it by inverting equation 1.





The presence of blowing snow is known to affect the wind profile, modifying the roughness length, which is thus supposed to increase with wind speed (Pomeroy and Gray, 1990; Nishimura and Hunt, 2000). However this increase was not experimentally confirmed in complex terrain (Doorschot et al., 2004; Vionnet, 2012), which is why we use for simplicity a constant roughness height $z_0 = 1.10^{-3}m$ which is the default SURFEX value for snow.

Most of transport flux occurs under a height $z \sim 1m$ above the surface in complex terrain with limited fetch (Pomeroy et al., 1993). The above assumption of neutrally stable flow is a good approximation under the following hypotheses: (i) the flow is steady (with respect to the time scale of the turbulent diffusion $\tau \sim \frac{\Delta z}{ku_*} \sim 1-10s$) (ii) the surface is plane, and no obstacle or terrain feature disturbs the flow upwind of the point (over a length $\sim \frac{zU}{u_*} \sim zln(\frac{z}{z_0}) \approx 10z$ with $z$ the maximum height where we want to simulate the wind profile) (iii) the flow is neutral, meaning turbulent kinetic energy production is

dominated by the surface shear stress. This is the case when the Richardson number $Ri << 1$. Hypothesis (iii) is reasonable in moderate to strong wind conditions associated with snow transport, however hypothesis (ii) is clearly questionable in complex mountainous terrain. However the terrain effect may mostly concern subgrid spatial scales at our target resolution, and anyway be too complex to be taken into account, so we keep this simple hypothesis, being aware of its limits.

### 2.2.3 Occurrence of snow transport in SnowPappus

We assume snow transport occurs when wind friction velocity exceeds a threshold friction velocity $u_t^*$ depending on the properties of the surface snow layer. Three cases are distinguished: (i) Following Vionnet et al. (2013) if the layer contains liquid water or if it has formerly contained some, meaning it is an ice crust, the snow is considered as non-transportable. (ii) In the case of dry snow aged of more than 1 hour, we use a threshold wind speed which can depend on snow microstructure. Two options were implemented:

– The default option GM98 : $u_t^*$ is calculated as a function of snow microstructure using the parameterization of Guyomarc'h and Mérindol (1998).

$$u_t^* = k \frac{U_t}{\log(\frac{h_{\text{ref}}}{z_0})} \tag{2}$$

$$\text{with } U_t = \begin{cases} 0.75d - 0.5s + 0.5 & \text{for dendritic snow} \\ -0.583g_s - 0.833s + 0.83 & \text{for non-dendritic snow} \end{cases} \tag{3}$$

$$\tag{4}$$

whith $U_t$ the wind velocity at a reference height $h_{\text{ref}}$=5 $m$. $d$, $s$ and $g_s$ are the dendricity, the sphericity and the grain size, which were the variables used to describe snow microstructure in the oldest versions of Crocus. They can be expressed as a function of sphericity and optical diameter $D_{opt}$ (see appendix B)

    – Option CONS : Threshold friction velocity is constant for snow aged of more than one hour.

(iii) If the snow layer age is inferior to 1 hour, we again follow Vionnet et al. (2013) fixing $U_t^{5m}$ to 6 m s$^{-1}$ during snowfall

events, arguing this threshold wind speed was observed in wind tunnel experiments by Sato et al. (2008).





Threshold wind speed depends strongly on the simulated microstructure properties of the surface snow layer, which are highly related to the parameterization of the properties of falling snow. For all SnowPappus simulations, we use the default parameterization of Crocus for falling snow, which makes it depend on wind speed (Vionnet et al., 2012).

The default SnowPappus way to calculate the threshold wind speed is strongly inspired by the method described by Vionnet et al. (2013), but a major difference is introduced. Indeed, the authors used the parameterization of Guyomarc'h and Mérindol (1998) in all dry snow conditions, even during snowfall. However, it leads to threshold wind speeds being too high during snowfall events due to wind dependence on falling snow properties. Consequently, they propose to modify the parameterization for falling snow, in order to get a more realistic threshold wind speed during snowfall. However, we preferred to modify the threshold wind speed during snowfall, rather than snow properties. To justify that, we argue that the GM98 parameterization should not be valid during snowfall events. Indeed, ice bonds which build cohesion of snow need some time to build up, and are necessarily weaker for freshly deposited snow. We consequently expect that a layer of freshly deposited wind-fracked snow particles has, for the same sphericity and grain size, a lower threshold wind speed during a snowfall event than after it. Compared evaluation of the two techniques will be presented in the next section. Note that this hypothesis leads to an instantaneous increase of the threshold wind speed when snow age reaches 1 h.

## 2.3 Horizontal blowing snow flux

### 2.3.1 Notations and geometric considerations

In the following section, we explain how the horizontal blowing snow flux can be estimated when transport occurs. We define $c$ as the concentration of snow particles in the air (kg m$^{-3}$), $u_p$ the horizontal speed of snow particles (m s$^{-1}$). Considering for simplicity the transport-related physical variables only depend on the height $z$, we can express the horizontal snow flux (kg m$^{-2}$ s$^{-1}$) as $q(z) = u_p(z)c(z)$. Then, the integrated horizontal blowing snow flux can be obtained by $Q = \int q(z)dz$ (kg m$^{-1}$ s$^{-1}$). $Q$ represents the total mass of snow transported horizontally by units of length and time. In this paper, we use $q$ for fluxes at a given height and $Q$ for integrated fluxes.

### 2.3.2 Blowing snow particles trajectories and transport modes

Snow transport by wind occurs when atmospheric forces are able to detach snow particles form the surface, forming a two-phase flow. The moment snow particles are detached from the snowpack, they are mainly submitted to gravity and drag force from the fluid. Electrostatic interactions with the surface could also play a big role close to the surface (up to a few centimeters), but they are rarely taken into account (Schmidt et al., 1999). The snow volumetric concentration within the air-snow mixture is low (under $10^{-3}$), so interactions between them are usually neglected (Bintanja, 2000; Kind, 1992). The turbulent nature of wind flow provokes fluctuations of wind speed experienced by particles. Therefore, they exhibit an apparently random motion due to the fluctuating drag force they are exposed to. This is usually modelled as a turbulent diffusion process (Bintanja, 2000; Gallée et al., 2001; Vionnet et al., 2014; Sharma et al., 2021).





The trajectory of transported particles can exhibit different shapes, corresponding to different transport modes. In the limit case where turbulent diffusion has a negligible influence on the trajectory, a particle falls back on the snow cover after a single jump of a few centimeters, with possible rebounds. This corresponds to "saltation" transport. In the opposite case, when turbulent diffusion plays an important role, particles exhibit a random motion on the vertical axis, so-called "suspension" and can reach high elevations, from decimeters to hundreds of meters above surface. Due to the combined effect of gravity and aerodynamic drag, particles in saltation motion cannot reach high heights, so that saltation is often considered to occur typically in a layer of a few centimeters above snow surface (Pomeroy and Gray, 1990; Nishimura and Hunt, 2000). Consequently, most blowing snow models solve saltation and suspension separately (Liston and Sturm, 1998; Pomeroy and Male, 1992; Bintanja, 2000; Vionnet et al., 2014; Marsh et al., 2020). They distinguish a saltation layer in the first centimeters above surface and a suspension layer located directly above it. However both processes can be described with the same dynamic equations (Nemoto and Nishimura, 2004) and the transition between them is not clear, leading some authors to introduce "modified saltation" for intermediate trajectories (Shao, 2005; Nemoto and Nishimura, 2004).

A third mode of transport, so-called reptation, has been described and corresponds to the rolling of big particles at the surface (Mott et al., 2018). However its contribution to the total snow flux was assumed to be negligible in most blowing snow models (Pomeroy et al., 1993; Vionnet et al., 2014; Sharma et al., 2021, ,...), which was based on experimental results of (Kosugi et al., 1992). We also followed this assumption, despite recent studies suggesting interactions with saltation transport (Aksamit and Pomeroy, 2016).

Furthermore, snow transport is not reduced to a sum of particle trajectories. In particular, saltation transport is a very complex phenomenon, where snow particles are detached from the snowpack by aerodynamic entrainment or ejection mechanism (Melo et al., 2022; Comola and Lehning, 2017), and perform centimetric-height jumps, with retroaction on the near-surface flow leading to the establishment of a steady-state flux. It is complexified in the case of snow by possible fragmentation of snow particles during impacts (Comola et al., 2017), leading to change in snow surface properties able to feedback on transport (Vionnet et al., 2013) and near-surface snow cohesion. It is still an open research topic, with still some recent fieldwork and experimental studies unravelling new saltation modes and mechanisms (Aksamit and Pomeroy, 2017; Mott et al., 2018).

### 2.3.3 Models developed to simulate the blowing snow flux

Numerous models with different degrees of complexity have been developed in order to simulate the air-blowing snow mixture. The most comprehensive and complex ones couple a Computational Fluid Dynamics model simulating the air movement from the near-surface to 5 - 20 meters height above the surface with the simulation of individual particles motion and interaction with the snow bed in a lagrangian mode (Nemoto and Nishimura, 2004; Groot Zwaaftink et al., 2014; Melo et al., 2022). These computationally intensive models are only applied in small domains (less than $100 \text{ m}^2$) in idealized situations, mainly for theoretical studies. Despite their complexity, they still rely on numerous simplifying assumptions, for example on particle shapes and ejection mechanisms (Melo et al., 2022; Nishimura and Hunt, 2000).

In order to deal with real-case applications, simplified models have been developed. In the latter, near-surface transport stated as saltation is usually represented by semi-empirical parameterizations, whereas snow concentration in the suspension layer is



computed in an eulerian mode. The equations governing particle concentrations usually include an advection term driven by the mean flow field, a sedimentation term, a diffusion term supposed to account for the effect of turbulent diffusion motion and a sink term accounting for sublimation. The associated three-dimensional advection-diffusion equation can be solved with different levels of approximation :

– full solving of a 3D advection-diffusion equation of the following shape, with some refinements to account for particle size distributions which are not presented here for simplicity.

$$\frac{\partial c}{\partial t} + (\mathbf{U}(\mathbf{x},\mathbf{t}).\nabla)c = \nabla.(K_{\text{snw}}(\mathbf{x},t)\nabla c(\mathbf{x},t)) - v_f \frac{\partial c}{\partial z} - s \tag{5}$$

with $K_{\text{snw}}$ the turbulent diffusion coefficient of snow particles (m$^2$ s$^{-1}$), $\mathbf{U}$ the wind speed (m s$^{-1}$), $v_f$ the terminal fall speed of snow particles (m s$^{-1}$) and $s$ the sublimation rate (kg m$^{-3}$ s$^{-1}$). This is the case for MAR (Gallée et al., 2001),
Crocus-MesoNH (Vionnet et al., 2014), SnowDrift3D (Schneiderbauer and Prokop, 2011a) and CryoWRF (Sharma et al., 2021) which solve Eq. 5 by fully coupling a snow model with an atmospheric model such as WRF or MesoNH. MAR and CryoWRF were applied to simulate entire snow seasons in the entire Antarctica Ice Sheet (Agosta et al., 2019; Hofer et al., 2021). However it was done at 27-35 km resolution, unadapted to mountainous environment. The high computational cost of atmospheric models in LES mode limits their applicability in complex terrain requiring finer
spatial and temporal resolutions (about 1 second). As a consequence, LES-based simulations of snow transport with 10 to 250 m resolutions were only carried out on much smaller domains or on a short time period covering a single event (Vionnet et al., 2014, 2017; Sharma et al., 2021).

   – In the work of Marsh et al. (2020), Eq. 5 was solved under a stationary state assumption, by neglecting $\frac{\partial c}{\partial t}$. Moreover, the wind profile in the $z$ direction was assumed to be logarithmic, allowing to use of a 2D wind field as an input. Solving
this equation in a stationary state allows solving the model with a much longer time step (here $\Delta t = 30$ min).

   – Finally, the effect of horizontal heterogeneity of wind speeds on the vertical concentration profile can be neglected, allowing to assess it independently on each point as the solution of a stationary 1D advection-diffusion equation :

$$\frac{\partial c}{\partial z}(K_{\text{snw}}(z)\frac{\partial c}{\partial z}) + v_f \frac{\partial c}{\partial z} + s(z) = 0 \tag{6}$$

This approach was used in Prairie Blowing Snow Model (Pomeroy et al., 1993), using a semi-empirical resolution
(Pomeroy and Male, 1992) and SnowTran3D (Liston and Sturm, 1998) based on a resolution of the above equation by Kind (1992).

To define a model suitable for our large-scale application, a trade-off between model complexity and accuracy was necessary. The numerical cost of fully coupled models prevents their use for our target domains and resolution. Moreover, several other factors may limit model accuracy. First, complex models exhibit a large number of physical parameters which are barely known
in real conditions, leading to high uncertainties. Besides, two-dimensional wind speed input may limit the added value of three-dimensional solving of Eq. 5. Finally, the target resolution (250 meters) is close or even bigger than the topographic scales able



to stop or enhance transport. As a consequence, the effects of subgrid variability of the wind field may dominate the effects of its resolved variability between grid cells. All these reasons led us to choose to solve the simple 1D advection-diffusion equation, as this approximation may not be the limiting factor for model uncertainty at our target resolution.

### 2.3.4 Suspension in SnowPappus

To compute the concentration profile in suspension, we follow a similar approach as Liston and Sturm (1998) and Pomeroy et al. (1993). SnowPappus (i) solves the 3D advection-diffusion equation (Eq. 5) in a 1-dimensional stationary state (Eq. 6), (ii) treats the sublimation in a specific routine (Sect. 2.4) and neglects its influence on the suspension concentration profile and (iii) assumes the diffusion coefficient of snow is proportional to the diffusion coefficient of momentum $K_{sca}$, leaving $K_{snw} = \frac{K_{sca}}{\zeta}$ with $\zeta$ a dimensionless quantity so called the Schmidt number (Vionnet, 2012; Naaim-Bouvet et al., 2010). As we assume a neutrally stable stratified flow, $K_{snw} = \frac{ku_* z}{\zeta}$. Besides, Eq. 5 assumes all blowing snow particles have the same terminal fall speed $v_f$ at a given space and time. However $v_f$ depends on their size and shape, which can exhibit a strong dispersion (Budd et al., 1966). This variability leads to enhanced particle concentration and fluxes at the top of the suspension layer, driven by particles with the smallest fall speed which reach more easily large height (Schmidt, 1982; Pomeroy and Male, 1992). SnowPappus follows the approach of Liston and Sturm (1998) and assumes homogeneous particle size and fall speed. It neglects the effect of $v_f$ dispersion in the suspension layer represented with a variable degree of complexity in other models (Bintanja, 2000; Déry and Yau, 1999; Vionnet et al., 2014; Déry and Yau, 2001; Yang and Yau, 2008; Pomeroy et al., 1993; Marsh et al., 2020).

All these hypotheses allow using Eq. 6 and reduce it to :

$$\frac{\partial \Phi}{\partial z} = 0 \tag{7}$$

$$\text{with } \Phi = \frac{ku^* z}{\zeta} \frac{\partial c}{\partial z} + v_f \frac{\partial c}{\partial z} \tag{8}$$

$\Phi$ is thus constant and represents the net vertical flux of blowing snow particles. We additionally hypothesize this flux can be neglected compared with other terms and obtain:

$$\frac{ku^* z}{\zeta} \frac{\partial c}{\partial z} + v_f c = 0 \tag{9}$$

$$c(z_r) = c_r \tag{10}$$

with $z_r$ a reference height (m) and $c_r$ a reference concentration (kg m$^{-3}$) used as a lower boundary condition further described in Sect. 2.3.6 dedicated to the transition between suspension and saltation. The solution of Eq. 9 and 10 is the following (Naaim-Bouvet et al., 2010):

$$c(z) = c_r (\frac{z}{z_r})^{-\gamma} \text{ with } \gamma = \frac{\zeta v_f}{ku^*} \tag{11}$$

Despite the strong assumptions necessary to obtain this power-law profile, it has been used successfully to fit observed concentration profiles (Guyomarc'h et al., 2019; Vionnet, 2012; Gordon et al., 2009; Mann et al., 2000). These authors point out that the decrease rate of the concentration with height depends on the particle fall speed and the Schmidt number.




In suspension motion, snow particles are embedded in the atmospheric turbulent air flow, consequently simple suspension models assume their horizontal mean velocity $u_p(z)$ is equal to wind speed $U(z)$ (Marsh et al., 2020; Liston and Sturm, 1998; Pomeroy and Gray, 1990). We also use this hypothesis in our work.

With all these elements, we can express the total suspension snow flux as :

$$Q_{susp} = \int_{z_r}^{h_{max}} c(z)U(z)dz \qquad (12)$$

which has the following analytical solution

$$Q_{susp} = \frac{c_r z_r}{k(1-\gamma(u_*))}[(\frac{h_{max}}{z_r})^{-\gamma(u_*)+1}(\log(\frac{h_{max}}{z_0}) - \frac{1}{1-\gamma(u_*)}) - (\log(\frac{z_r}{z_0}) - \frac{1}{1-\gamma(u_*)})] \qquad (13)$$

$h_{max}$ is the maximum height of the suspension layer, and has mainly an influence at very high wind speeds when the flux profile becomes non-integrable. The computation details are given in Sect. 2.3.8. Parameterizing $v_f$ and $\zeta$ is necessary to compute $Q_{\text{susp}}$ and will be the object of the following subsection.

### 2.3.5 Parameterization of the effective terminal fall speed of suspended snow particles

The concentration profile in the suspension layer, and thus the flux in the suspension layer, depends strongly on the $\gamma$ exponent, itself depending on the terminal fall speed and the Schmidt number. However, the only direct field measurement of blowing snow terminal fall speed was performed by Takahashi (1985) in Antarctica. Estimations of $\zeta$ are indirect and mostly rely on concentration profile analysis (Vionnet, 2012; Naaim-Bouvet et al., 2010). They might be affected by several phenomena such as turbulent kinetic energy destruction, incorrect estimation of $v_f$, etc. Thus, it is easier to rely on direct estimations of the $\gamma$ exponent from field observations of concentration profiles rather than estimating it from the physical parameters $\zeta$ and $v_f$. Therefore, an effective terminal fall speed $v_f^* = \gamma k u_*$ is defined to parameterize the suspension layer in SnowPappus. Analysis of concentration profile at *Col du Lac Blanc* in the French Alps and in Antarctica (Vionnet, 2012) at height of 0.1 - 1 m, as well as $v_f$ measurements from Takahashi (1985) suggest that (i) observed $v_f^*$ or $v_f$ have a large variability and range from 0.2 to 1.0 m s$^{-1}$ (ii) a "recent snow" and "old snow" regime are distinguished, $v_f^*$ increasing with snow age (iii) $v_f^*$ increases with wind speed, at least in the case of recent snow. In this latter case $v_f^*$ fits correctly with Naaim-Bouvet et al. (1996) parameterization. These observed trends have possible theoretical explanations. Indeed, stronger winds can make bigger particles enter suspension. Besides, older snow particles have a more rounded shape (and therefore a higher fall speed) than fresh dendritic snow flakes, due to the effect of sublimation and fragmentation occurring during transport events. These studies considered snow was recent either during a precipitation event (Takahashi, 1985; Vionnet, 2012) or when it was aged less than one day. Here, we define from Vionnet (2012) observations $v_{f,old}^* = 0.8$ m s$^{-1}$ and $v_{f,fresh}^* = \min(0.38u_* + 0.12, 0.8$ m s$^{-1})$ (parameterization from Naaim-Bouvet et al., 1996). Then, in SnowPappus, the effective terminal fall speed is set to :

$$\begin{cases} v_f^* = & v_{f,\text{fresh}}^*(u_*) & \text{if } d > 0 \text{ and } A < 0.05 \text{ days} \\ v_f^* = & v_{f,\text{old}}^* & \text{if } d = 0 \\ v_f^* = & v_{f,\text{old}}^*(1-F) + v_{f,\text{fresh}}^*F & \text{otherwise} \end{cases}$$





with $A$ the age of the surface snow layer and $F = \min(1, \frac{d}{d_m})$ where $d$ is the dendricity of the snow surface layer (Vionnet et al., 2012; Carmagnola et al., 2014). Our distinction between 'old' and 'fresh' regimes is arbitrary. We are always in the fresh snow case during a precipitation event, and move to the old snow regime more or less fast depending on $d_m$. This adjustable quantity is set by default to 0.5 (dimensionless), as the result of calibration on *Col du Lac Blanc* data, its sensitivity is assessed in the following evaluations (Sect. 3.3).

### 2.3.6 Lower boundary condition for suspension transport

In most blowing snow models, the transition between saltation and suspension layer is treated assuming the height of the saltation layer can be defined, and that particle concentration at the top of it can be used as a lower boundary condition for the suspension layer. However, detailed saltation models (Melo et al., 2022; Nemoto and Nishimura, 2004) indicate that (i) there is a change in the decay rate with the height of snow particles concentration in the transition zone (ii) a bimodal particle size distribution is observed in the transition zone, with one mode associated with the biggest particles vanishing above the transition zone, and the other mode associated with the smallest particles vanishing under it. Nemoto and Nishimura (2004) argue that in this transition zone, the smallest particles are yet in suspension whereas the biggest ones are still in a saltation motion. This suggests not all saltating particles are able to enter the suspension state, with only the smallest ones being picked up. Thus, we argue particle concentration at the top of the saltation layer cannot be simply taken as a boundary condition, as it is still a transition zone where part of particles are in saltation motion, without significant effect of turbulent diffusion on their trajectories, and part in suspension motion. The upper height of this transition zone can be estimated at 12-14 cm from Melo et al. (2022) results. In absence of precise information on this transition zone, the lower boundary condition for suspension transport (Eq. 10) should preferably be extrapolated from concentration measurements in the suspension layer rather than from estimation of the concentration in the saltation layer.

Pomeroy and Male (1992) fitted field-observed suspension flux and showed $(z_r = h^{P92}, c(Z_r) = c_{P90})$ is a suitable boundary condition to simulate fluxes in the suspension layer, with $c_{P90}$ the concentration predicted by the saltation model of Pomeroy and Gray (1990) which is detailed in the following subsection, and $h^{P92} = a u_*^{1.27}$ with a = 0.0834 m$^{-0.27}$ s$^{1.27}$ Consequently, we chose to use this boundary condition for our suspension model (Sect. 2.3.5). However, $h^{P92}$ is clearly located in the transition zone between saltation and suspension, so the concentration profile predicted by this model under 10-15 cm must be seen as an extrapolation of suspension behaviour.

### 2.3.7 Flux in the saltation layer and in the transition zone

In the following subsection, we will discuss how to compute the transport flux in the region under 10-15 cm height where the transport is a priori not pure suspension. It includes the saltation layer and the so-called "transition zone".

Simple semi-empirical parameterizations have been developed to simulate the flux and concentration of blowing snow in the saltation layer. Two of them, which were used in distributed snow transport models, are reviewed in detail in the following :





– Parameterization of Pomeroy and Gray (1990) (P90):

$$Q_{\text{salt}}^{P90} = h_{\text{salt}}^{P90} q_{\text{salt}} \tag{14}$$

$$= \frac{ce\rho_{\text{air}}}{g} u_t^*(u^{*2} - u_{*,t}^2) = A\frac{\rho_{\text{air}}}{u^* g} u_t^*(u_*^2 - u_t^{*2}) \tag{15}$$

with $h_{\text{salt}}^{P90} = 1.6\dfrac{u_*^2}{2g}$ (16)

With $Q_{\text{salt}}^{P90}$ the integrated saltation flux, $h_{\text{salt}}^{P90}$ an estimation of the height of the saltation layer and $\rho_{\text{air}}$ the air density (kg
m$^{-3}$). $c$ is the ratio between the saltating particle horizontal velocity $u_p$ and the threshold friction velocity $u_t^*$ and $e$ the
saltation efficiency, i.e. the proportion of energy conserved after a particle impact. Field measurements of the horizontal
snow flux at a height of approximately 2 cm in a large flat and undisturbed area of North America plains, with various
snow types and weather conditions, were assumed to be representative of $q_{\text{salt}}$ in the whole saltation layer. They were
used to calibrate the $ce$ product to $ce = \frac{A}{u^*}$ with $A = 0.68$m s$^{-1}$. Thus, this parameterization can be considered as a fit of
these experimental data. Arguing that $e < 1$, Pomeroy and Gray estimated a lower bound for $c$ based on $ce$ measurements
($c = 2.8$). This formulation was widely used in blowing snow models (Pomeroy et al., 1993; Liston and Sturm, 1998;
Bintanja, 2000; Gallée et al., 2001; Marsh et al., 2020), and the value $c = 2.8$, necessary to estimate particle concentration
in saltation, is used in every reviewed blowing snow models using semi-empirical modelling of saltation, without giving
further experimental validation of this value to the best of our knowledge (Pomeroy et al., 1993; Liston and Sturm, 1998;
Bintanja, 2000; Gallée et al., 2001; Vionnet et al., 2014; Marsh et al., 2020)

– Parameterization of Sørensen (2004) - Vionnet (2012) (S04):

$$Q_{\text{salt}}^{S04} = \frac{\rho_{\text{air}} u_*^3}{g}(1 - V^{-2})(a + bV^{-2} + cV^{-1}) \tag{17}$$

with $V = \frac{u_*}{u_{*t}}$ and $a$, $b$ and $c$ calibration parameters (Sørensen, 2004). They were calibrated as $a = 2.6$, $b = 2.5$ and $c = 2$
in the case of snow (Vionnet, 2012) in order to reproduce transport rates from a wind tunnel experiment (Nishimura and
Hunt, 2000). It was used in the coupling of Crocus with MesoNH (Vionnet et al., 2014).

P90 and S04 give very different results, S04 predicts a much higher flux, by a factor of about 10, which is already highlighted
by several authors (Melo et al., 2022; Doorschot and Lehning, 2002). Such discrepancy could also influence the suspension flux
via the boundary condition. Despite a physical basis, P90 and S04 are calibrated on measurements, from terrain observations
in the case of P90 in various conditions (and in particular wind speeds), and from a wind tunnel experiment in the case of S04,
conducted on a single, non-cohesive, snow type, at a single wind speed and air temperature (Nishimura and Hunt, 2000). Thus
P90 seems to have better empirical support than S04. However, there are high uncertainties in the measurement carried out to
calibrate P90, in particular concerning the measurement height (Pomeroy and Gray, 1990), and the fact that the more complex
saltation models support the magnitude of S04 predictions (Doorschot and Lehning, 2002; Melo et al., 2022). Besides, the use
of a non-cohesive snow with spherical grains in Nishimura and Hunt experimental set-up, which is not representative of natural
snow could at first be hypothesized as an explanation, but the theoretical work of Melo et al. (2022) suggests snow cohesion





has not such a big influence on saltation transport. Consequently we would not expect such discrepancies between S04 and
P90.

Numerical and experimental works supporting S04 use snow fluxes integrated up to 10 or 15 cm (Nishimura and Hunt,
2000; Melo et al., 2022) whereas P90 formulation is supposed to represent flux between 0 and $h_{\mathrm{salt}}^{P90}$ which is typically 1-4 cm
in the range of speed explored in the experiments of Pomeroy and Gray and Nishimura and Hunt. Thus, we argue that $Q_{\mathrm{salt}}^{S04}$
represents not only the saltation transport but also all the transition zone towards suspension transport, whereas $Q_{\mathrm{salt}}^{P90}$ gives
only the flux at the base of the saltation layer. Thus, both formulations cannot be compared directly. To overcome this difficulty,
in the following, we note $Q_{\mathrm{inf}}$ the blowing snow flux integrated up to a height of $h_{susp} = 10 - 15$ cm and propose two methods
to compute it, based respectively on P90 and S04 saltation model:

– **S04** : $Q_{\mathrm{inf}} = Q_{\mathrm{salt}}^{S04}$

– **P90 + SnowPappus** : We separate the lower atmosphere into two sublayers (1) between snow surface and $h^{P92}$ (defined
in Sect. 2.3.6, used for the lower boundary condition for suspension) (2) between $h^{P92}$ and $h^{susp}$. In layer 1, we consider
the behaviour of Pomeroy and Gray (1990) can be applied, so that the speed of snow particles is $u_p = 2.8u_{*,t}$ and
$c = c_{salt}^{P90}$. In layer 2, the SnowPappus suspension behaviour is extrapolated, with $u_p = U(z)$ and $c$ follows Eq. 11 with
the boundary condition described in Sect. 2.3.6. Thus the flux integrated between the surface and $h^{susp}$ is computed by:

$$Q_{\mathrm{inf}} = Q_1 + Q_2 \tag{18}$$


$$Q_1 = \int_0^{h^{P92}} c_{\mathrm{salt}}^{P90} u_p^{P90} dz = Q_{\mathrm{salt}}^{P90} \frac{h^{P92}}{h_{salt}^{P90}} \tag{19}$$

$$Q_2 = \int_{h^{P92}}^{h_{\mathrm{susp}}} c(z)U(z)dz \ \ \text{with} \ \ c(z) = c_{\mathrm{salt}}^{P90}\left(\frac{z}{h^{P92}}\right)^{\frac{-v_f^*}{ku_*}} \tag{20}$$

It has to be noted that this representation introduces a flux discontinuity at $h^{P92}$ which may be unrealistic and could be
improved by a smoother formulation in the future.

Both methods are represented schematically in Fig. 1
We now compare the outputs of both methods to compute the blowing snow flux up to $h_{\mathrm{susp}} = 10\text{-}15$ cm, as presented above.
The results of the method **SnowPappus+P90** depend on the value of $h_{\mathrm{susp}}$. Thus, we computed the blowing snow flux with this
method between 0 and 10 cm (height of Nishimura and Hunt experiment) and between 0 and 15 cm (height used by Melo et al.
to compare S04 with their more complex saltation model). Results are presented in Fig. 2 and shows that the predicted fluxes
by both models give close results for friction velocities lower than 0.6 m s$^{-1}$, and that SnowPappus fluxes tend to clearly curb
down at high wind speed (around 0.6-0.7 m s$^{-1}$ for 0-10 cm, 0.8-0.9 m s$^{-1}$ for 0-15 cm). We conclude that the huge difference
observed between S04 and P90 is resolved at low wind speed by adequately representing the bottom of the suspension layer
which is implicitly included in the S04 formulation. Both formulations give there a flux of the same order of magnitude and thus
experimental and theoretical validations of S04 are also in agreement with SnowPappus. At high wind speed, both formulation




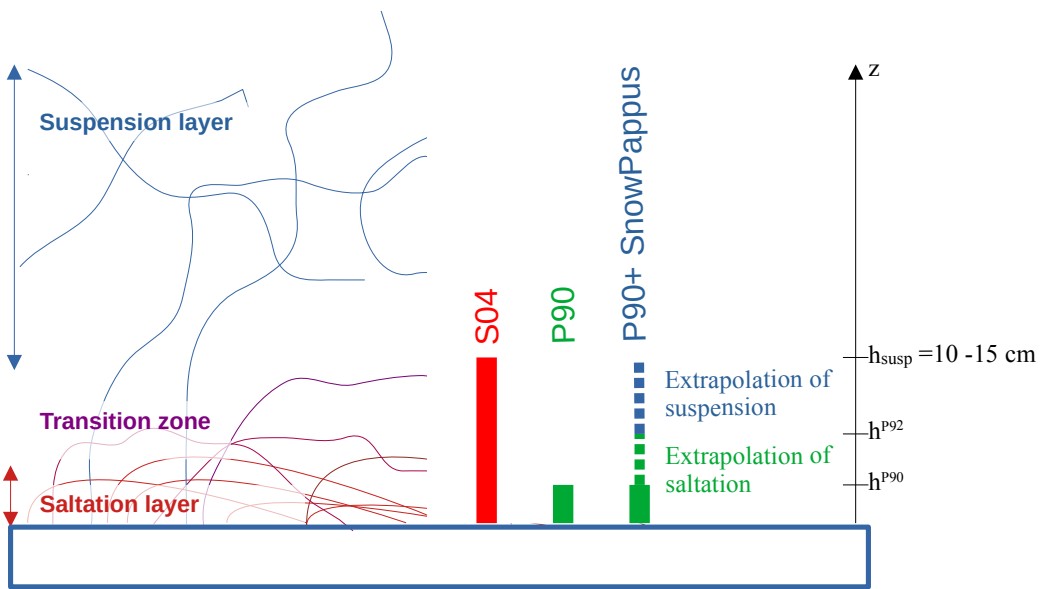

**Figure 1.** Schematic representation of the suspension and saltation layers and of the transition zone between them. The part of fluxes computed by the two saltation models S04 and P90 are represented schematically, as well as the 'P90+ SnowPappus' method, combining the P90 saltation model and the SnowPappus suspension model.

diverge, with near-surface SnowPappus fluxes tending to curb down. This behaviour is clearly due to the fast growth of $h^{P92}$

with wind speed, making the low P90 saltation flux applied to most of the 0 - 15 cm layer. It must be noted that observations and simulations used for validation of the formulations are in the range 0.2 - 0.85 m s$^{-1}$ (Pomeroy and Gray (1990), Pomeroy and Male (1992), Nishimura and Hunt (2000), Melo et al. (2022)). Thus, $h_{P92}$ might become unreasonably high and P90 flux inadequate at wind speed above $0.8\ ms^{-1}$. On the other hand, S04 could be representative of the flux integrated up to more than 10-15 cm above the surface at high wind speeds. Thus, we argue SnowPappus gives consistent results compared with

other parameterizations of the literature for low to moderate wind speed, additionally providing a coherent boundary layer for suspension transport. However, information is lacking on the saltation regimes at higher wind speeds, and P90+SnowPappus might simulate unrealistic fluxes in this case.

By default, we choose to use the P90+SnowPappus option in SnowPappus, as it gives a more coherent link between saltation and suspension, and because of better empirical support of P90. However S04 can also be used as an alternative option to

compute the blowing snow flux between 0 and 15 centimeters, however keeping the flux above 15 cm unchanged compared with the default option.



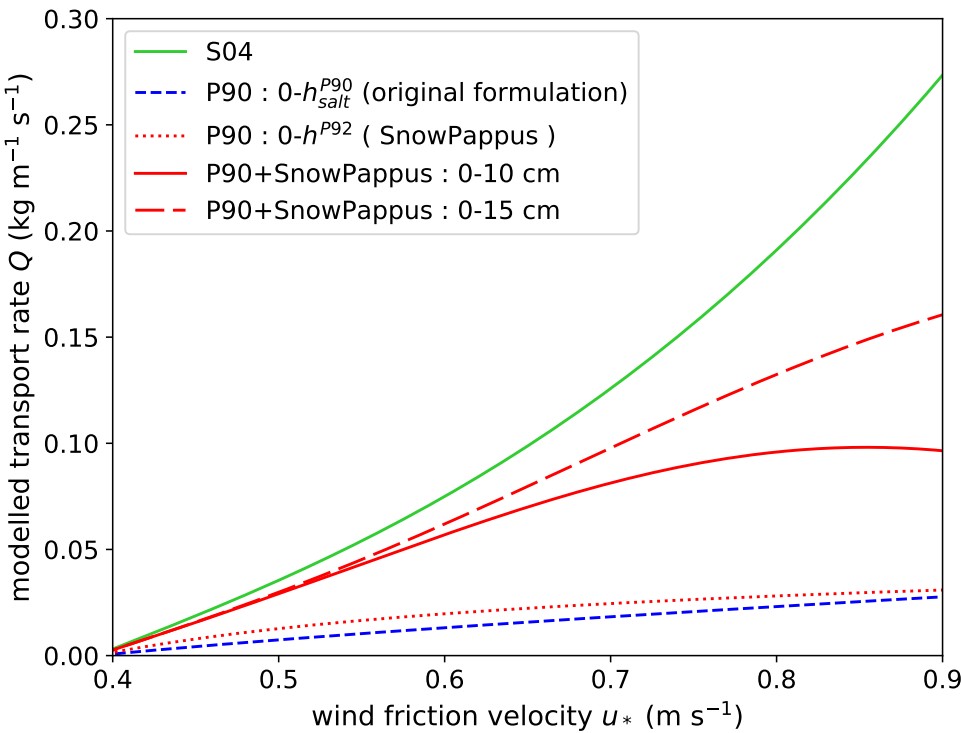

**Figure 2.** Comparison of the predicted flux in the saltation layer and the transition with suspension (up to 10 - 15 cm) with the two methods described in Sect. 2.3.7 based on P90 (blue curve) and S04 (green curve). P90-modelled flux is computed up to (i) the integration height $h^{P92}$, which corresponds to the entire "saltation" layer, (ii) 10 cm which is the height on which are integrated Nishimura and Hunt measurements used to calibrate S04 parameterization and (iii) 15 cm which is the height used by Melo et al. to compare S04 with a more complex saltation model, with good agreement shown (red curves). P90 formulation represents a flux integrated up to a height of $h_{\mathrm{salt}}^{P90} \approx 1-4cm$ which is not directly comparable to S04, whereas SnowPappus integrated up to 10-15 cm is. Here, all the modelled are computed for a threshold wind speed $u_{*t} = 0.39ms^{-1}$, and the type of snow used in SnowPappus is old (non-dendritic)

### 2.3.8 Influence of the upwind fetch distance on flux

In complex terrain, snow transport is influenced by the local topography, and in particular by the upwind distance where snow transport starts typically after a slope break or an obstacle preventing snowdrift. This is called the fetch distance $l_{\mathrm{fetch}}$. It has an influence on both saltation and suspension transport.

Indeed, the saltation models presented above are designed to describe steady-state saltation. However, fields measures on long and plane surfaces suggest that the saltation transport rate grows during approximately 300-500 meters before reaching a steady state (Takeuchi, 1980; Dyunin, 1967). This nonstationarity of saltation transport may seriously affect transport rates





in mountainous terrain, as slopes are rarely constant over several hundreds of meters. This makes comparisons between field
observations and wind tunnel experiments difficult, as the latter occurs with very limited fetches. The mechanism involved in
this instationarity remains unexplained, although it is taken into account in an empirical way in many blowing snow models
(Pomeroy et al., 1993; Liston and Sturm, 1998; Bowling et al., 2004; Marsh et al., 2020). We follow this literature and use the
parameterization proposed in SnowTran3D (Liston and Sturm (1998)) to modify the saltation flux :

$$q_{salt} = f(l_{\text{fetch}})q_{salt}^{P90} \quad \text{with} \quad f(l_{\text{fetch}}) = (1 - \exp(-3\frac{l_{\text{fetch}}}{l^*})) \tag{21}$$

where $l^* = 500$ m represents the fetch length at which the saltation flux reaches 95% of its steady state value (Pomeroy et al.,
1993).

Fetch distance also influences suspension transport. Indeed Pomeroy et al. (1993) assume the maximum height reached by
particles is limited by the time available to diffuse $t_d$, so that $h_{max} = ku_*t_d$. Consequently, suspension transport grows with
fetch distance. In SnowPappus, we follow these authors and additionally assume this is valid no matter the fetch distance.
We also simplified the expression used by Pomeroy et al. (1993) to get an analytical expression, obtaining $h_{max} = h_{salt} +$
$\frac{k^2}{\sqrt{log(\frac{h_{\text{salt}}}{z0})log(\frac{5m}{z_0})}}$. Appendix Fig. A1 compares the exact and approximated formula, and shows their difference is small. This
approach assumes an abrupt end of the suspension layer at the height $h_{max}$, which is not realistic. However it influences the
model outputs only if a significant flux would occur above $h_{max}$. We can show it happens only when wind speed exceeds 30-40
m s$^{-1}$ ($\gamma$ becomes lower than 1, making the flux profile not integrable) or if $h_{max}$ is inferior to $\approx 30$ cm, when $l_{fetch} < 10-20$
m.

Figure 3 shows the influence $l_{fetch}$ has on the total transport flux, which is very strong in the first hundreds of meters. In
mountainous environment, we can expect typical fetch distance to be in this range, and mostly inferior to our resolution (250
meters). We chose for simplicity to use by default a constant fetch distance on the whole grid $l_{fetch} = 250m$, which is a strong
hypothesis. Different methods and algorithms exist to compute fetch in complex topographies (Bowling et al., 2004; Marsh
et al., 2020) and could be included in a future version of SnowPappus.

### 2.3.9 Total blowing snow flux

To summarize, the final expression of the snow transport flux is:

$$Q_t = Q_{salt} + Q_{susp} \tag{22}$$

$$Q_{salt} = \int_0^{h^{P92}} q_{salt}^{P90}dz = h^{P92}q_{salt}^{P90} = Q_{salt}^{P90}\frac{h^P 92}{h_{salt}^{P90}} \tag{23}$$

$$Q_{susp} = \int_{h^{P92}}^{h_{max}} c(z)U(z)dz \quad \text{with} \quad c(z) = c_{salt}^{P90}(\frac{z}{h^{P92}})^{\frac{-v_f^*}{ku_*}} \tag{24}$$

The flux is highly dependent on the wind friction velocity $u_*$ and also depends on the threshold wind speed $u_{*t}$ and on snow
dendricity and age.



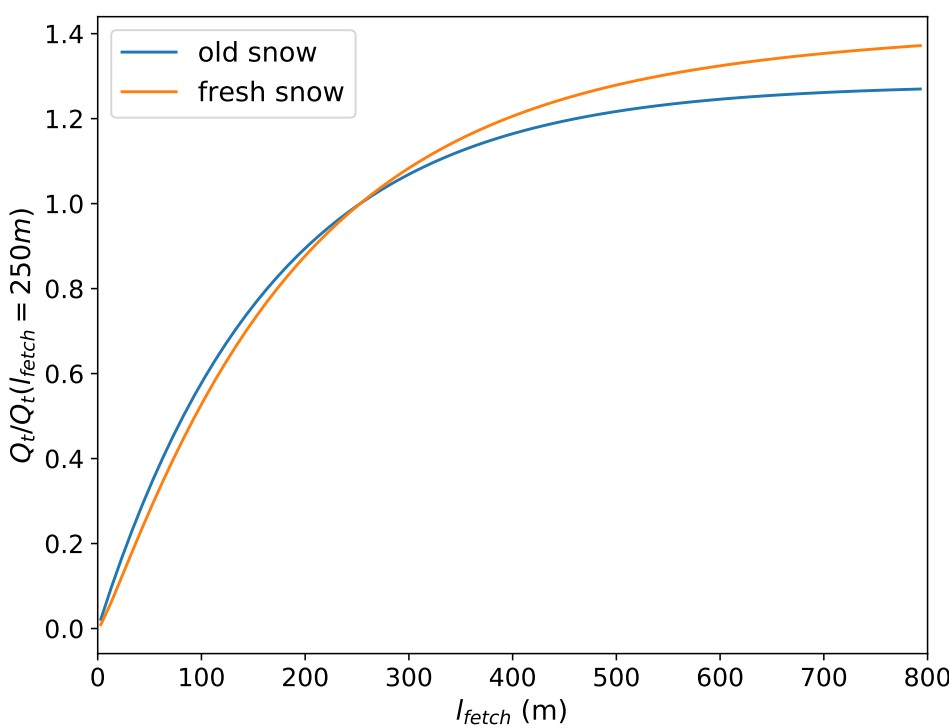

**Figure 3.** ratio between the total modelled transport rate $Q_t$ and its value for the default fetch value $l_{fetch} = 250m$, as a function of fetch distance. The case of fresh snow, i.e. dendritic snow with here $u_{*t} = 0.27$ m s$^{-1}$ (orange curve) and old snow, i.e. non-dendritic snow with here $u_{*t} = 0.39$ m s$^{-1}$ (blue curve) are compared, both for a wind friction velocity of 0.6 m s$^{-1}$ (which corresponds to approximately to a 2-m wind speed of 12 m s$^{-1}$)

## 2.4 Sublimation

Blowing snow sublimation is known to play an important role in the alpine water budget. Strasser et al. (2008) estimate
blowing snow sublimation loss in the Alps to 17-19% of cumulative snowfall. Wind blowing is known to increase sublimation, MacDonald et al. (2010) suggested that neglecting blowing snow sublimation can lead a hydrological model to overestimate snow accumulation by up to 30% in Canadian Rockies. According to these reports we choose to implement blowing snow sublimation among our modelled processes. Two different parameterizations of blowing snow sublimation rate $Q_{subl}$ (kg m$^{-1}$ s$^{-1}$) were implemented in SnowPappus with default choice to Simplified Blowing Snow Model (SBSM) parameterization
Essery et al. (1999).





The first is an implementation of the SBSM parameterization from Eq.6 of Essery et al. (1999) and the SBSM documentation:

$$Q_{subl} = \frac{\mu_{satc} \times 137.6}{F(T)} \times \frac{u_{wind}{}^5}{25000} \tag{25}$$

with F(T) beeing a scale factor function and $\mu_{satc}$ the under-saturation.

The second is from Eq.9 of Gordon et al. (2006) :

$$Q_{subl} = A \left(\frac{T_0}{T_a}\right)^4 U_t \rho_a q_{si} (1 - Rh_i) \left(\frac{U}{U_t}\right)^B, \quad \text{for} \quad U > U_t \tag{26}$$

with $Rh_i$ relative humidity to ice, $q_{si}$ the saturation specific humidity, $\rho_a$ (kg m$^{-3}$) the air density, $U$ and $U_t$ ($m$) wind and 5 meters wind threshold for transport at 5m and A, B constants defined in (Gordon et al., 2006)

## 2.5 Mass balance

We have seen in Sections 2.3 and 2.4 details about the computation choice of the different snow transport horizontal fluxes. This section describes how SnowPappus simulates mass exchanges between neighbouring grid cell, once the amount of horizontal transport flux has been computed for each pixel. In our model, the snow transport direction is always the same as the wind direction. Therefore the problem simplifies as solving the mass balance equation. The mass balance can be solved using the following continuity equation:

$$q_{dep}(x,y,t) = \nabla.Q_t(x,y,t) - q_{subl}(x,y,t) \tag{27}$$

with $q_{dep}$ (kg m$^{-2}$ s$^{-1}$) the pixel snow deposition *expressed by sloping snow surface unit*, $q_{subl}$ (kg m$^{-2}$ s$^{-1}$) the snow transport sublimation and $\nabla.Q_t$ the total snow transport flux divergence. We define $Q_t$ as the total vertically integrated horizontal blowing snow flux (kg m$^{-1}$ s$^{-1}$). In a given mesh grid we assume:

$$\nabla.Q_t = \left( \sum_{faces} \frac{Q_{t,in}(x,y,t)}{l_{px}} - \sum_{faces} \frac{Q_{t,out}(x,y,t)}{l_{px}} \right) \cos(\theta) = (\nabla.Q_t)_{\text{flat}} \cos(\theta) \tag{28}$$

with $l_{px}$ the pixel length. Eq 28 means that the divergence is expressed as the sum of the total snow transport flux $Qt$ leaving and entering the pixel faces (in and out). Within SURFEX grid configuration, each grid point has a defined slope angle $\theta$. We consider $l_{px} = l_{res} \cos(\theta)$ with $l_{res}$ the grid horizontal resolution (m). This assumes the transport flux is always parallel to the slope. This approximation was necessary to strictly preserve the total mass balance of the system. The SnowPappus model uses a regular cartesian mesh grid discretization with cell-centred storage. This means each simulation point is regularly disposed on the simulation zone with each simulation point representing a squared pixel of fixed size. Our mesh grid being cell centred, we do not compute the transport fluxes at the pixel faces, as needed for the continuity equation 28. To obtain these values, an upwind scheme (Patankar, 2018) has been implemented, i.e. the zonal and meridian components of the fluxes at the face are assumed to be equal to the zonal and meridian flux computed at the center of the upwind pixels (in both directions):

$$Q_t(i,j,t) = Q_t(i \pm \frac{1}{2}, j \pm \frac{1}{2}, t) \tag{29}$$



with $i,j$ being the grid coordinates of the center of pixels and $i \pm \frac{1}{2}, j \pm \frac{1}{2}$ pixel's border, as illustrated by Fig. 4.

This scheme was preferred to a linear interpolation of fluxes because there is no evidence on the variation of the blowing snow transport flux between two points 250m apart is linear (See Sect. 5.3.2 of discussion). The linear approximation seems more appropriate for much smaller mesh sizes. Supporting this assumption, the effect of fetch on the transport flux may cause the flux to respond to the change in the wind and snowpack conditions with a lag of a few hundred meters, the same order of magnitude as a grid cell. Also, the upwind scheme simplifies the steps needed to close the mass balance budget in preventing

snow mass creation. In our use case, the linear interpolation method would need extra steps with "gradient limiters" to ensure the boundedness of the mass balance. (Greenshields and Weller, 2022)

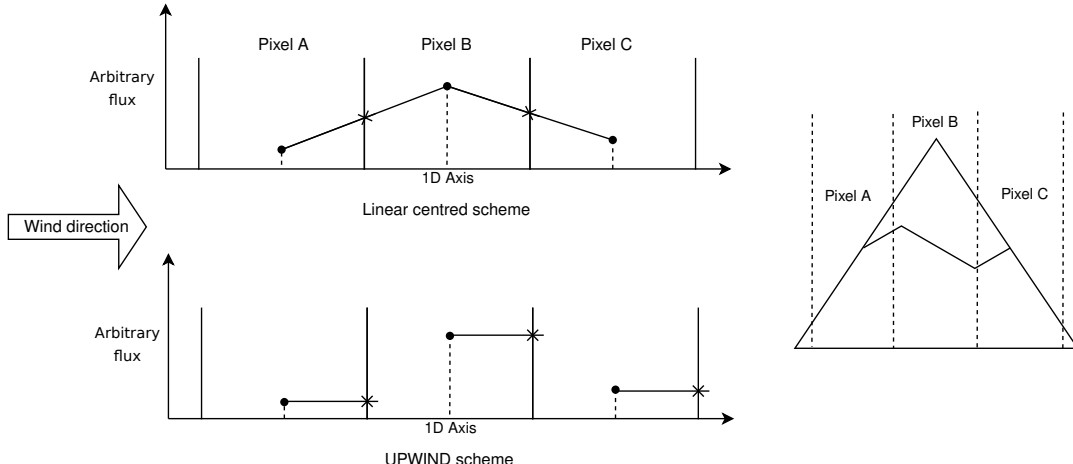

**Figure 4.** Illustration of the differences between the upwind scheme and a more classical linear scheme for a 1D ideal case. Dots represent the flux estimated for each pixel by Eq 30. Crosses (X) represent the flux value crossing the pixel face. In the linear scheme, the flux value leaving the pixel is the linear interpolation between the two-pixel cell center values. The flux crossing the pixel face can be very different from the cell center computed value. The border flux being linearly interpolated, the border value can be unrealistic. This behaviour can cause interpolation to move erosion out of the expected zone (usually summits, the windiest zone). In the upwind scheme, the flux value crossing the pixel is identical to the cell-centred computed value. This locks the border value to realistic values and gives the scheme added stability.

We consider a given grid cell, on which local horizontal transport rate is written $Q_{t,\mathrm{out}}$. We consider the four neighbouring cells located respectively north, south, west and east of the cell. We call $Q_{t,i}$ with $i = 1, \cdots, 4$ the horizontal transport rates on these cells, and call $\overrightarrow{n_i}$ the unitary normal vector going in the corresponding direction. With the upwind numerical scheme to

obtain the pixel face crossing values, we obtain the following continuity equation for our SnowPappus model :

$$(\nabla . Q_t)_{\mathrm{flat}} = \sum_{i=1,\cdots,4} Q_{t,i}(x,y,t) l_{px} \, min(\overrightarrow{W_{dir}}.\overrightarrow{n_i}, 0) - \sum_{i=1,\cdots,4} Q_{t,\mathrm{out}}(x,y,t) l_{px} \, max(\overrightarrow{W_{dir}}.\overrightarrow{n_i}, 0) \qquad (30)$$




$(\nabla . Q_t)_{\text{flat}}$ was defined in Eq. 28, $min(\overrightarrow{W_{dir}}.\overrightarrow{n_i}, 0)$ and $max(\overrightarrow{W_{dir}}.\overrightarrow{n_i}, 0)$ giving respectively the flux direction coefficient (same as wind direction) crossing each face normally for in and out the direction.

The code implementation of Eq. 30 is explained in more detail in paragraph 2.7. Distributed hardware requires the use of the
MPI interface (Clarke et al., 1994) with communication between different parts of the subdomain.

## 2.6 Influence of snow transport and deposition on snow surface properties

In the process of wind-induced snow transport, snowflakes experience multiple collisions with snow surface and sublimation (Comola et al., 2017). It leads to their transformation towards small rounded grains developing quickly a strong sintering cohesion and associated with compaction of near-surface snow layers. There is a negative feedback loop with transport by
changing the threshold wind speed and snow particles properties (Mott et al., 2018). Amory et al. (2021) observed a linear increase of snow density with time during a blowing snow event. Despite this qualitative knowledge of the phenomena, to our knowledge no other observation-based parameterization of this effect is available, in particular concerning microstructure. Thus, we chose to represent the process in a simple way :

– Snow deposited by a transport event (when $q_{\text{dep}} > 0$) has the properties of rounded grains with sphericity $s = 1$ and
dendricity $d = 0$, and relatively high density $\rho = 250 kg m^{-3}$.

– Wind-induced metamorphism might also originate either from subgrid snow transport or from horizontal displacement of snow with erosion and deposition flux compensating each other. Thus, the preexisting parameterization for wind-induced snow metamorphism from Vionnet et al. (2012) can still be activated, but considering the SnowPappus threshold wind speed instead of the original formulation. It makes the surface snow layers slowly become denser and evolving towards
small rounded grains approximately linearly with time. The impact is tested in Sect. 3.3

## 2.7 Implementation in SURFEX

SnowPappus is implemented inside the SURFEX/ISBA land surface scheme, which computes the evolution of soil and snow properties sequentially. The code is parallelized using MPI protocol, to be able to run distributed simulations over large domains in a short time.
At the beginning of a new simulation, the spatial domain is divided into subdomains. Each of them is associated with an MPI thread. Then, soil and snow models are called. They compute at each time step the evolution of soil and snow properties. For each time step, the SnowPappus routine is called before the snowpack scheme. It computes the horizontal transport rate, $Q_t$, and the blowing snow sublimation rate, $q_{\text{subl}}$, for each grid point, according to the surface properties computed in the previous time step. Then, once $Q_t$ and $q_{subl}$ are computed for all pixels, this information is shared with the processors associated with the
adjacent grid point by a blocking MPI communication. After this phase, the erosion/deposition rate can be computed with Eq. 30 and converted into an amount of snow to remove or add to the snowpack. If there is net erosion, snow is directly removed in the SnowPappus routine. Otherwise, the falling snow amount and properties are computed in the SnowPappus routine and then given as snowfall input for the Crocus snow scheme. The Crocus routine is called after SnowPappus. It deals with adding snow

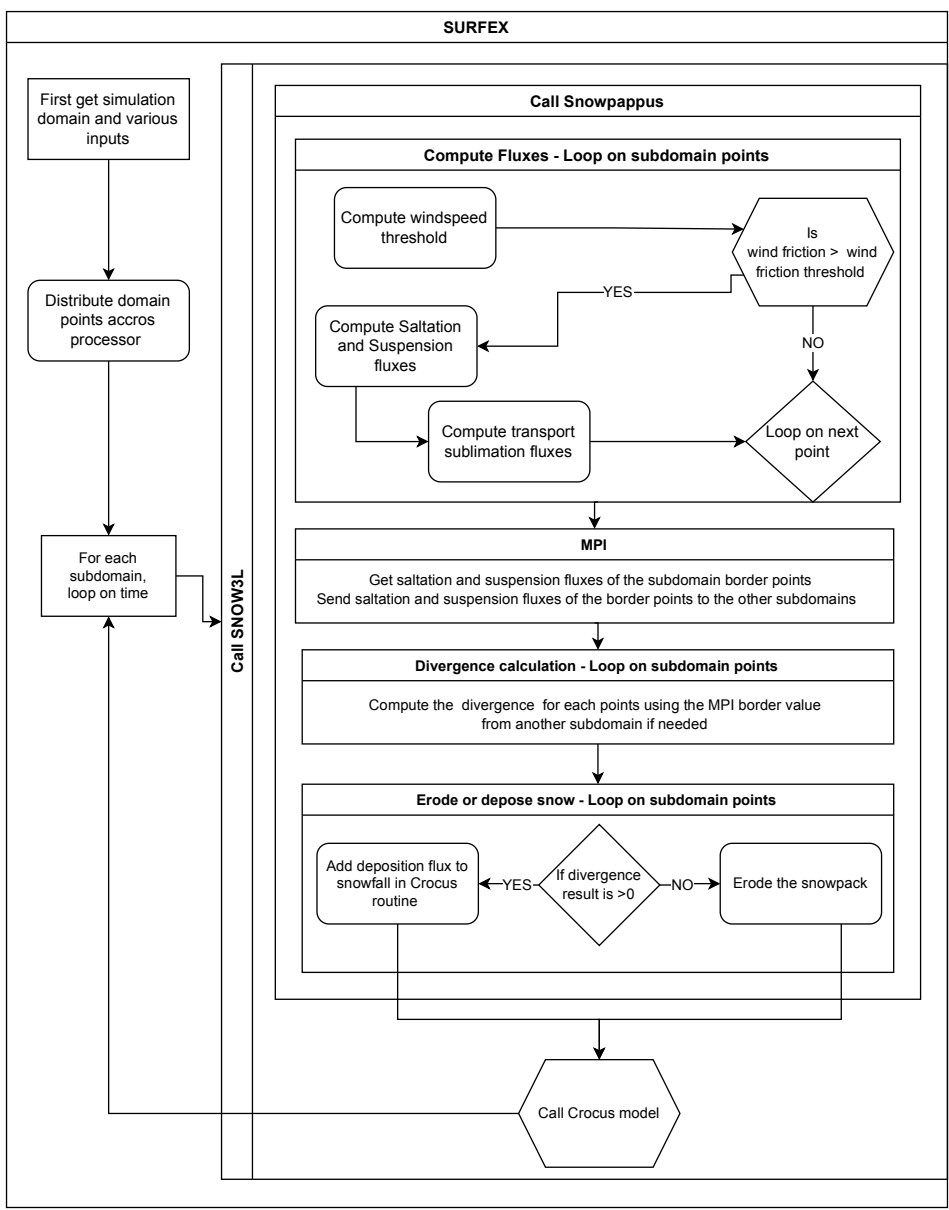

**Figure 5.** Description of how the SnowPappus routine is organized in the SURFEX framework and how it connects with the Crocus snow model.

to the snowpack and modifying snow layers accordingly to snowfall and SnowPappus outputs. If a snowfall and wind-driven
snow deposition occur simultaneously, the amount of added snow is the sum of both. Its density and microstructure variables
are a weighted average of falling snow and blowing snow properties. The detailed equations for this process are described in





appendix C. The Crocus routine then takes back the original Crocus model course and makes the properties of snow evolve through metamorphism, heat diffusion, compaction, percolation, etc. Those processes are summarized in Fig.5.

It is important to note that transport rate and snowpack evolution are solved in a decoupled mode (one after the other). Therefore, the deposition rate $q_{dep}$ is computed independently from the amount of snow available on the considered grid point, which can lead to the amount of snow removed from the point being higher than the available snow mass. In this case, the mass balance is not respected. The cumulative amount of this "ghost snow" is stored in a variable to check it remains small. To prevent this behaviour from occurring, limitations on $Q_t$ and $q_{subl}$ are made. The condition is the following :

$$q_{subl} \leq \frac{W}{t_{step}}$$

$$Q_t \leq (\frac{W}{t_{step}} - q_{subl})\frac{l_{px}}{cos(\theta)}$$

with $W$ being the snow mass for each pixel in kg m$^{-2}$, $Pt_{step}$ the computation time step in $s$, $l_{px}$ the pixel size in $m$ and $\theta$ the slope angle. Note that it is possible to deactivate this limitation.

## 3 Evaluation: Methods

### 3.1 Study area

For demonstration and evaluation purposes, we run simulations over a test zone covering the whole "Grandes Rousses" massif in the French Alps, with a spatial resolution of 250 meters. It covers 14443 grid points (3200 km$^2$). This area exhibits a complex topography, with elevation ranging from 700 to more than 3500 meters, involving a large range of temperature conditions and snow coverage duration. For this test zone, most winter storms come from North-Western flows. Besides, to demonstrate the ability of SnowPappus to be run over large domains, we also set up a simulation domain containing the whole French Alps with about 868000 simulation points. Both domains are illustrated in Fig. 6

### 3.2 Meteorological forcing

The Crocus snow model needs various atmospheric forcing variables: liquid and solid precipitations, incoming shortwave and long-wave radiations, air temperature and humidity, wind speed and direction. In this work, all these variables but the wind are given by the SAFRAN reanalysis (Vernay et al., 2022) over geographical units so-called 'massifs' of about 1000km$^2$ in which meteorological conditions only depend on elevation at a vertical resolution of 300 meters. Here, all meteorological variables are interpolated on a 250 m resolution simulation grid, at the exact elevation of each grid point, derived from a DEM at this resolution (as in Revuelto et al. (2018) and Deschamps-Berger et al. (2022)). The DEM was created by averaging on each grid point the 5 m resolution RGE Alti® DEM, provided by the Institut Geographique National at the scale of France. Snow transport modelling is strongly sensitive to the quality of the wind forcing (Musselman et al., 2015), consequently wind fields taking into account the effect of local topographic features were preferred to the very large-scale SAFRAN wind fields. Here, km-scale wind fields are first extracted from AROME NWP model at a 1.3 km resolution (Seity et al., 2011; Brousseau et al.,





2016), then downscaled at a 30 meters resolution using the DEVINE downscaling method (Le Toumelin et al., 2022) and finally resampled at 250m using a simple average. DEVINE method benefits from the use of convolutional neural networks to downscale kilometer scale winds from AROME to high-resolution local topography, based on preliminary training with

wind speeds simulated with ARPS atmospheric model (Xue et al., 2000) on a high number of synthetic topographies. Previous evaluations of DEVINE have shown that contrary to basic wind interpolation methods, DEVINE is able to reproduce several characteristics of terrain forced flow (speed-up on crests, windward deceleration, channelling through gaps and passes), that largely influence the onset and evolution of drifting snow episodes. Notably, model evaluation has shed light on improved wind speed estimations with DEVINE compared to raw NWP model outputs at elevated and exposed in-situ observation stations

(Le Toumelin et al., 2022).

### 3.3    Evaluation data

Blowing snow flux data is available for 3 stations in the Grandes Rousses test zone. One of these is the *Col du Lac Blanc* observatory where long-term monitoring of blowing snow fluxes and of various atmospheric forcings have been performed (Guyomarc'h et al., 2019). In particular, a vertical profile of Snow Particle Counters (SPC) (Sato et al., 1993) recording

blowing snow fluxes at four different heights, is located at a particularly wind-exposed location. Because of frequent snowfall and ablation events, the height of these sensors varies significantly with time. Consequently, a vertically integrated flux between 0.2 and 1.2 m is provided from these four measurements from 01 December to 01 April since 2010 (Guyomarc'h et al., 2019). The shapes of vertical profile concentration are classified into different categories: 'no flux' when no flux was recorded, 'inconsistent' when flux at the different height was physically inconsistent, 'power-law' when the concentration profile fits

with a power-law or 'mean' when the flux depends weakly on height. 'mean' flux is expected when the flux is dominated by solid precipitations, whereas 'power-law' profile is expected when it is dominated by wind-induced snow transport (see Sect. 2.3.5). We use these data to evaluate blowing snow occurrence and fluxes.

     The other sites are the Huez (FHUE) and Chambon (FCMB) stations from the ISAW network, which data were already used in blowing snow studies (Vionnet et al., 2018; He and Ohara, 2017). They are equipped with snow height, temperature, wind

measurements and Flowcapt sensors (Chritin et al. (1999)), which record integrated blowing snow from 0 to 2m above the ground surface. Trouvilliez et al. (2015) indicate that Flowcapt sensors of different generations give similar results with respect to SPC if a threshold value higher than 1 g m$^{-2}$ s$^{-1}$ is taken. However, they can be partially buried under snow depending on snow height and their reliability in terms of estimated flux is still debated in the literature (Cierco et al., 2007; Trouvilliez et al., 2015; Vionnet et al., 2018). Thus, similarly to Vionnet et al. (2018), we use them only to evaluate blowing snow occurrence.

Flowcapt data available in ISAW stations was specifically cleaned up as described in Vionnet et al., paragraph 3.1.





### 3.4 Model set up

#### 3.4.1 2D simulations

The two-dimensional simulations were performed with the "default" SnowPappus configuration, using GM98 option for wind speed threshold and deactivating wind-induced snow metamorphism. The simulations were run from 01 August 2018 06:00

UTC to 01 August 2019 06:00 UTC. Soil temperatures were initialized by a model spinup from 01 August 2008 06:00 UTC to 01 August 2018 06:00 UTC in a similar configuration except for the wind which also comes from SAFRAN during the spinup period. This simulation was used for numerical performance assessment and local evaluation of blowing snow occurrence. A reference simulation where snow transport was deactivated was also run over the same period with the same initial conditions.

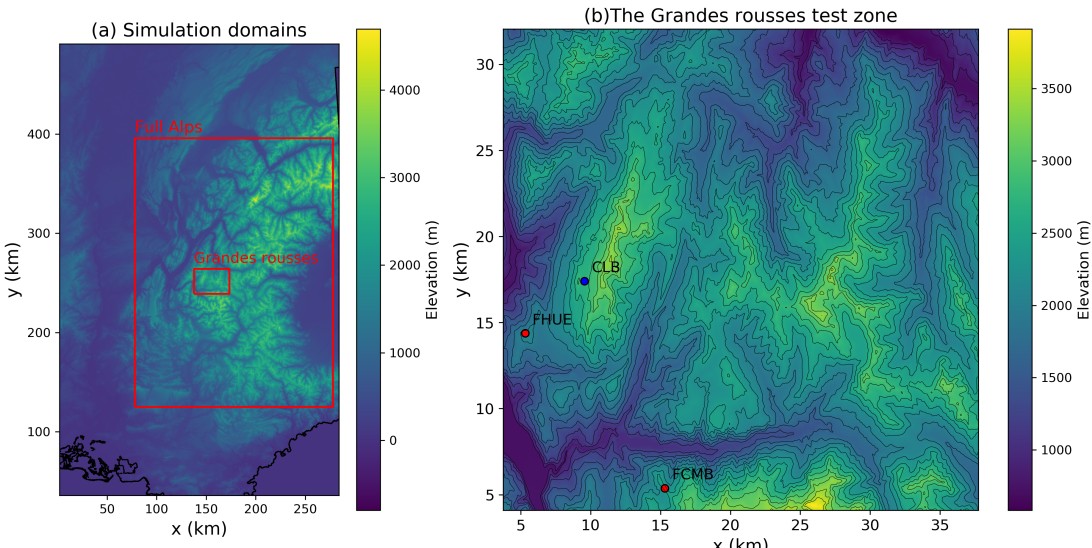

**Figure 6.** (a) Limits of the two domains used for simulations in the article. The small 'Grandes Rousses' test zone is used for two-dimensional simulations presented in Sect. 4.2 . The 'Full Alps' domain was used to test the possibility of running simulations on the entire French Alps (see Sect. 5.5). (b) Topographic map of the 'Grandes Rousses' The location of the *Col du Lac Blanc* experimental site (blue dot) and of the two flowcapt sensors of ISAW network (red dots) are indicated. Equidistant iso-elevation lines are represented. Data from RGE ALTI® was used to generate these maps.

#### 3.4.2 Local simulations

The accuracy of wind forcing is known to play a major influence on any evaluation of a snow transport model (Vionnet et al., 2018). Consequently, we also run and evaluate local scale simulations at *Col du Lac Blanc* forced by observed 5-m wind speed in order to distinguish the contributions of wind speed errors and model errors in our results. Wind speed sensors as well as SPC



are located on the same mast, at the station "AWS Col" described in Guyomarc'h et al. (2019). Their height above surface is variable and recorded continuously, allowing to estimate 5-m wind speed assuming a logarithmic wind profile and a roughness
length $z_0 = 2.3$ mm. The other meteorological variables are interpolated from SAFRAN reanalysis as for 2D simulations. In this configuration, blowing snow fluxes are computed by the model but do not result in any erosion or deposition. These local simulations were performed from 01 August 2010 to 01 August 2020 with a soil temperature initialized by a spinup from 2000 to 2010. At Col du Lac Blanc, fetch distance in the mean wind direction as defined in Sect. 2.3.8 was estimated to be $110 \pm 20m$, as the mean distance between the SPC sensors and the center of accumulation zones identified from Vionnet (2012,
figure4.24). We thus fixed $l_{fetch} = 110$ m in the blowing snow flux simulations.

Several model configurations were used to investigate simulated blowing snow occurrence, and are described here:

– **SnowPappus: Run A** SnowPappus default configuration with wind-induced snow metamorphism deactivated (see Sect. 2.6). 'GM98' option is used for threshold wind speed (see Sect. 2.2.3)

– **SnowPappus: Run B** SnowPappus default configuration with wind-induced snow metamorphism activated.

– **SnowPappus: Run C** SnowPappus with 'CONST' option (see Sect. 2.2.3), meaning 5-m threshold wind speed is constant and equal to 9 m s$^{-1}$ for snow aged more than one hour and 6 m s$^{-1}$ for falling or freshly fallen snow. Note the 9 m s$^{-1}$ was calibrated provide the optimal Heidke Skill Score (see in Sect. 3.5) among different tested values (not shown).

– **Vionnet 2013** SnowPappus with parameters putting it in the exact same configuration as Vionnet et al. (2013) for wind speed threshold calculation and falling snow properties (see Sect. 2.2.3)

Additionally, to investigate the sensitivity of the simulated fluxes of blowing snow to $d_m$ (Sect. 2.3.5), we tested three modified configurations of run B configuration with $d_m$ values of 0, 0.5 and 1.

## 3.5 Evaluation of blowing snow occurrence

We evaluate blowing snow occurrence using the same framework as Vionnet et al. (2018), allowing comparisons with the Sytron operational system, which can be considered as a benchmark. Contrary to SnowPappus, Sytron is based on an 8-
aspect idealized geometry Vionnet et al. (2018). Both systems share the Guyomarc'h and Mérindol (1998) parameterization for threshold wind speed calculation. Vionnet et al. (2018) used hourly averages of fluxes. They considered blowing snow is *observed* if the blowing snow flux measured by the SPC and integrated between 0.2 and 1.2m exceeded a threshold of 1 g m$^{-1}$ s$^{-1}$ and *simulated* if a non-zero blowing snow flux was simulated. They did not distinguish cases with or without snowfall, and defined blowing snow days as days with more than 4 consecutive hours of blowing snow. As the detection of
snow particles coming from solid precipitation may be sufficient to reach this threshold, this method may overestimate blowing snow occurrence during snowfall. However, complementary analyses limiting the occurrence of blowing snow to the additional condition of having at least one 'power-law' shape profile (Sect. 3.3) in a given hour, which is expected only when there is wind-induced snow transport, do not show any significant difference in our results (not shown).





Here, the study was conducted on the whole 2010-2020 period, while Vionnet et al. considered only the 2015-2016 season.
As in Vionnet et al. (2018), the false alarm rate (FAR), the probability of detection (POD) and the Heidke skill score (HSS) are used to evaluate the different setups:

$$POD = \frac{a}{a+c} \tag{31}$$

$$FAR = \frac{b}{a+b} \tag{32}$$

$$HSS = \frac{2(ad-bc)}{(a+c)(c+d)+(a+b)(b+d)} \tag{33}$$

with $a$, $b$, $c$, $d$ respectively the number of true positive, false positive, false negative and true negative events. HSS varies between -1 and +1, +1 for a perfect agreement and 0 for a random forecast.

These scores were first applied to the 4 model configurations previously described and to new runs of the Sytron system to cover the same evaluation period and share the same code version of SURFEX-Crocus. For Sytron, as in Vionnet et al. (2018) (although not mentioned in the original publication), the occurrence of blowing snow is considered detected if a non-zero flux is simulated on at least one of the 8 slope aspects.

The occurrence scores are then applied to the 2D simulation outputs for the 2018-2019 season, driven by simulated wind fields as described in Sect. 3.1. The evaluation is carried out at *Col du Lac Blanc* station and at both ISAW stations. Each station was associated with the closest grid point from its location.

### 3.6 Evaluation of blowing snow fluxes

Integrated blowing snow fluxes between $z_{min} = 0.2m$ and $z_{max} = 1.2m$, called $Q_{t,int}$ were evaluated against SPC data. Modelled flux is computed at each time step as :

$$Q_{t,int} = \int_{z_{min}}^{z_{max}} q_{susp}(z)dz = \frac{c_r z_r u_*}{k(1-\gamma)}[(\frac{z_{max}}{z_r})^{-\gamma+1}(\log(\frac{z_{max}}{z_0}) - \frac{1}{1-\gamma}) - (\frac{z_{min}}{z_r})^{-\gamma+1}(\log(\frac{z_{min}}{z_0}) - \frac{1}{1-\gamma})] \tag{34}$$

Note that the blowing snow flux below 20 cm in height can not be accounted for in this evaluation although it may represent a significant contribution to the total flux, especially for low or moderate wind speeds. Observed data are available with a 10-minute time step. For evaluation purposes, model output and SPC fluxes were first averaged hourly. As the observed fluxes cannot be used when precipitating particles prevail, periods when at least one 'mean' concentration profile was observed were removed. Considering the other missing data, 4947 hours of data are considered in the evaluation.

To assess the ability of SnowPappus to capture the long-term magnitude of wind-induced snow transport, monthly averages of simulated fluxes were compared to the observed ones, keeping only the months when at least 15 days of valid observed data are available. Hourly data were also classified distinguishing two cases : (i) days with snowfall according to SAFRAN reanalysis and (ii) days with no snowfall. Observation-derived wind friction velocity was also classified in 20 equally distant wind speed intervals (interval widths are about 0.064 $ms^{-1}$). Averaged observed and modelled flux by category and wind speed step were computed and compared. Less than 20 hours of data were available by wind steps for Wind speeds of more than 0.95 $ms^{-1}$ so these wind speeds were not considered.





These scores were applied to the 3 variants of Run B configuration (3 values of $d_m$).

## 4  Results

### 4.1  Comparison of simulated blowing snow flux with simple parameterizations in the literature

Before evaluating SnowPappus against observations, the relationship between $Q_t$ and wind speed is illustrated in Fig. 7 for fresh and old snow, and compared to other estimates from the literature. It stresses that, due to a lower terminal fall speed of
snow particles, fresh snow exhibits much higher transport rates than old snow. The model of Essery et al. (1999) exhibits results almost identical to the "old snow" case of SnowPappus. Compared with empirical observations of Mann et al., SnowPappus transport rates simulated in both cases are in the range of the very spread observed values, at least for wind speeds lower than 20 m s$^{-1}$. The fresh snow case seems to correspond approximately to the upper bound of observations. These observations show that at least the magnitude of SnowPappus flux is plausible.

### 4.2  Illustration of simulation output over Grandes Rousses study area

Synthetic simulation outputs of the simulation performed during the whole 2018-2019 winter season are presented in Fig. 8. The cumulated erosion/deposition rate during the whole season is low over a large part of the domain but it reaches absolute values of around 500 kg m$^{-2}$, which are comparable to the total amount of solid precipitation in the area, in some points. As expected, hot spots of transport are located around high alpine crests, generally limited to the crest summit and the two adjacent
grid cells. Higher deposition seems to usually occur on the Eastern side of the crest, consistently with the prevailing winter storms coming from North-Westerly Atlantic flows.

The relative average snow depth change compared to the simulation where snow transport was deactivated is also shown in Fig. 8. It is very similar to the deposition pattern, and also shows that some crests summits experience almost complete snow removal due to transport (relative change close to -1) whereas the amount of snow is more than doubled in some deposition
zones.

### 4.3  Evaluation of blowing snow occurrence at Col du Lac Blanc

HSS, FAR and POD of the different set-ups are compared on Fig. 9 for all days and for the days without snowfall. Probabilities of detection range typically from 80 to 90 %, and false alarm rates from 30 to 40 % considering the whole period. FAR are higher (40-60 %) considering only days without snowfall.
The SnowPappus default option (run A) exhibit slightly lower FAR and POD than the approach of Vionnet et al. (2013) (run D), which leads to similar in HSS, in particular when considering only days without snowfall. Accounting for wind-induced snow metamorphism option (run B) modifies only slightly the scores and did not imporove the detection of blowing occurrence. All these methods including a threshold wind speed that depends on the properties of surface snow exhibit high false alarm rates, and do not perform better than using a constant 5-m threshold wind speed equal to 9 m s$^{-1}$ in no-snowfall conditions and



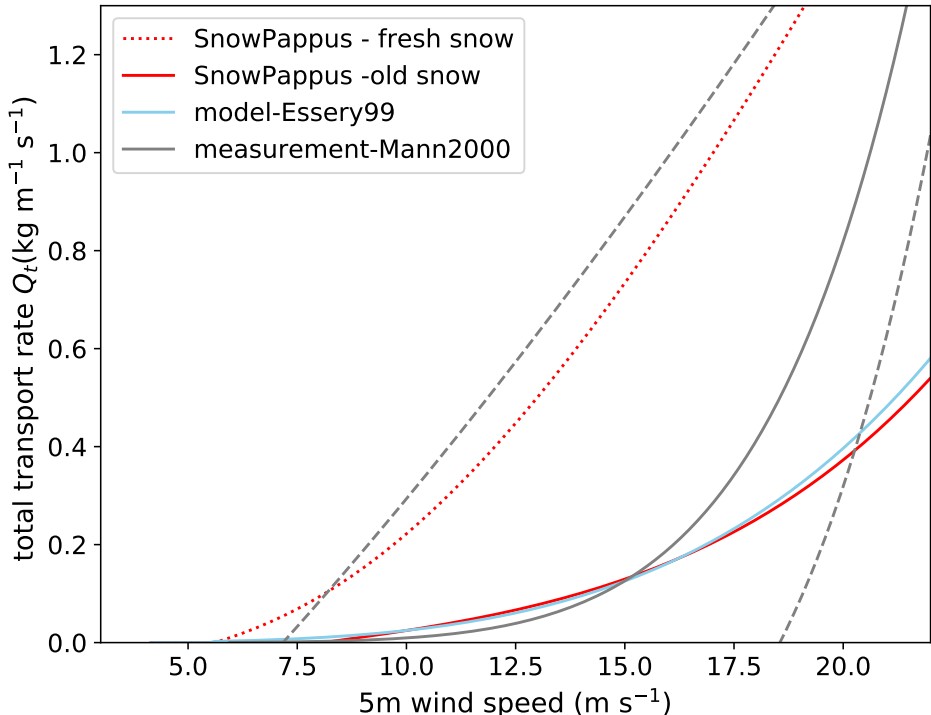

**Figure 7.** Total blowing snow flux $Q_t$ predicted by SnowPappus as a function of 5-m wind speed, for old and fresh snow (red lines, see Fig. 3 for old and fresh snow discrimination). The blue line represents the flux predicted by a simplified theoretical model (Essery et al., 1999), assuming $z_0 = 1mm$ and the grey line represents an empirical formula derived from observations in Antarctica (Mann et al., 2000) $Q = 1.504u_*^5.144$ and taking $z_0 = 5.6.10^{-5}m(\frac{u_*}{0.3ms^{-1}})^2$ as the authors of the studies. Dashed grey lines represent the upper and lower bounds of observed flux, roughly estimated from Fig. 6 of the article.

6 m s$^{-1}$ during precipitation events (run C). Thus, increased complexity of the parameterizations based on snow microstructure do not result in any visible amelioration of the quality of blowing snow occurrence simulations. In addition, SnowPappus, with its different options, and Sytron operational model exhibit similar scores.

### 4.4   Evaluation of blowing snow occurrence in 2D simulations

Scores of blowing snow detection obtained with 2D simulations at *Col du Lac Blanc*, Huez and Chambon stations are shown

in Fig. 10. HSS and POD are low in this configuration using wind speed downscaled at 250-m grid spacing using the DEVINE approach. At Col du Lac Blanc, the HSS is lower than the HSS obtained for point-scale simulation forced by observed wind speed, with the same threshold wind speed parameterization. This decrease in HSS mainly results from a strong decrease in POD (Fig. 10b) whereas the FAR remains similar (Fig. 10c). It suggests that the accuracy of downscaled wind speed and/or



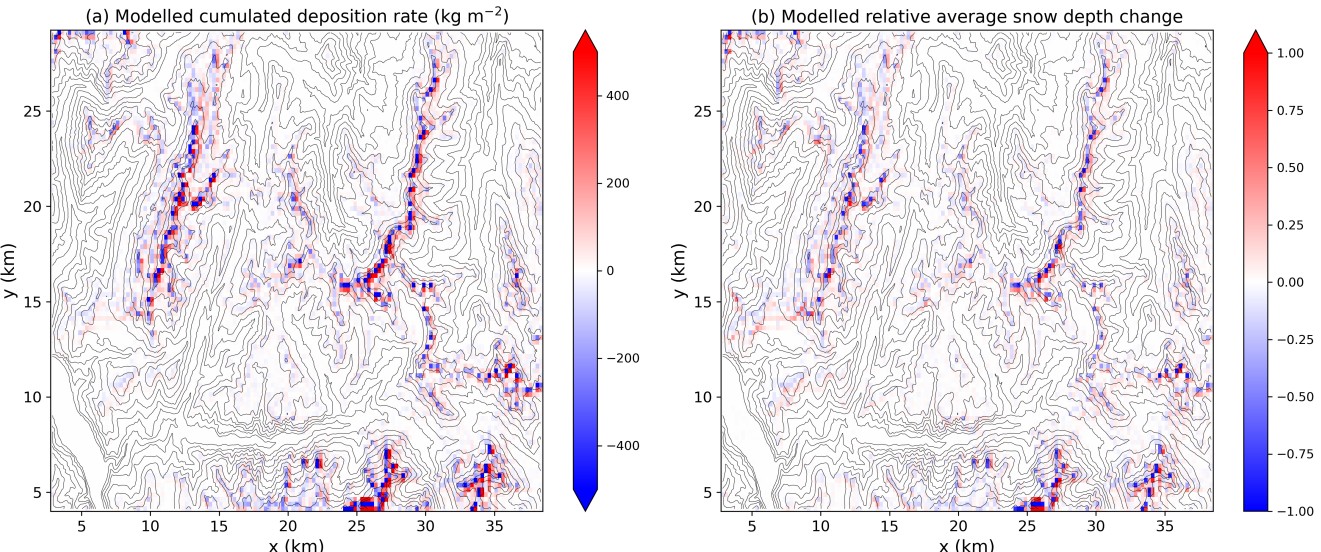

**Figure 8. (a)** Cumulated deposition rate $\sum_{t_i} q_{\text{dep}}(t_i)$ simulated by SnowPappus transport scheme during the whole 2018-2019 winter season in the whole Grandes Rousses test zone.**(b)** Relative difference in yearly-averaged snow depth $h_s$ between a simulation with transport (w.t.) activated and simulation without transport (n.t.) $\Delta h_s = \frac{h_s^{w.t.} - h_s^{n.t.}}{h_s^{n.t.}}$

the 250 m spatial resolution of the simulation are the main causes of the skill deterioration, as confirmed by the significant

discrepancies between observed and simulated wind speeds at the three stations (Fig. 11). A negative bias in simulated wind speeds is observed, in particular for strong winds, explaining the low detection rates.

### 4.5 Evaluation of blowing snow fluxes at Col du Lac Blanc

Figure 12a shows the simulated monthly averaged fluxes between 0.2 and 1.2 m $Q_{t,int}$ at *Col du Lac Blanc* as a function of the observed ones, for the 3 tested $d_m$ values. As expected, fluxes clearly increase when $d_m$ decrease, because it makes the

terminal fall speed closer to the fresh snow regime. Simulated $Q_{t,int}$ is of the same order of magnitude as the observed one. It is clearly overestimated when $d_m = 0$ case, clearly underestimated when $d_m = 1$ and slightly underestimated when $d_m = 0.5$. In all cases, modelled fluxes seem to correlate well with observed ones, however with a strong dispersion. In particular, one specific month has a simulated flux 8 times higher than the observed one regardless of the $d_m$ value. Fluxes and wind speeds during this month are shown in appendix Fig. A2. This figure shows that the discrepancy between observed and simulated

fluxes is caused by the occurrence of a storm with wind speeds larger than 15 m s$^{-1}$ and recent snowfall while no blowing snow flux was observed. All model configurations simulated large fluxes for this storm. The cause of this unusual situation (and the possible link with the surface snow properties) is not known and might be due to a SPC failure. These results are discussed in details in Sect. 5.1.

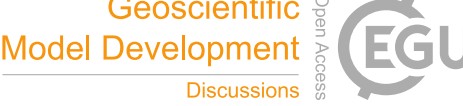

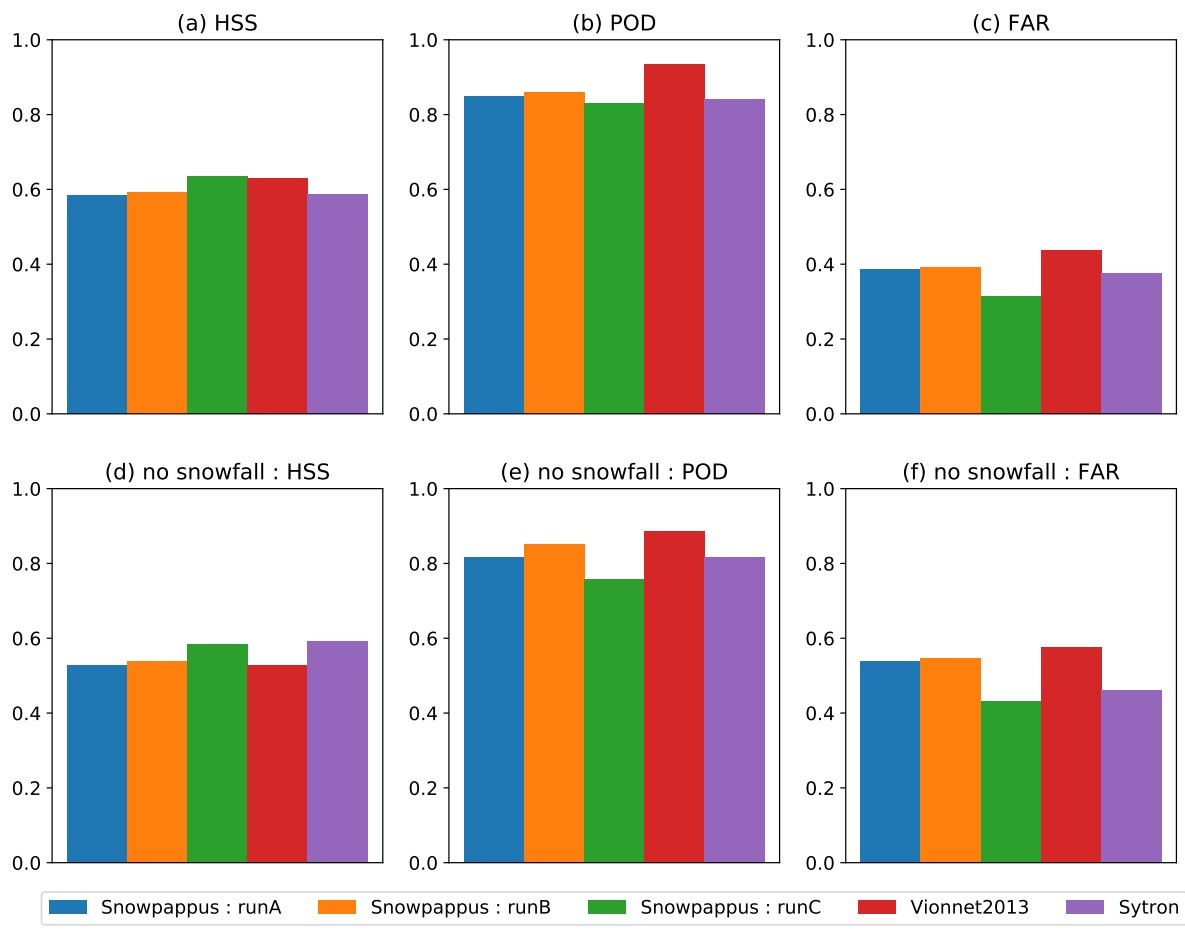

**Figure 9.** Evaluation of the detection of blowing snow days with point-scale SnowPappus simulations and Sytron operational system. HSS (a,d), POD (b,e) and FAR (c,f) of different SnowPappus configuration and other models, computed from all days independently (a,b,c) and only for days with no snowfall(d,e,f) are represented. As more detailed in Sect. 3.5 ,Run A is default SnowPappus configuration, run B is the same with wind-induced snow metamorphism, run C uses a constant threshold wind speed for dry snow aged more than 1 hour and 'Vionnet2013' is supposed to reproduce configuration described in Vionnet et al. (2013).

Figure 12b shows average simulated and observed fluxes as a function of the 5m-wind velocity $u_*$ for days with and without snowfall, for wind friction velocities between 0 and 0.95 m s$^{-1}$. Simulated and observed fluxes show the same dependency with $u_*$, the flux remaining negligible under a threshold wind speed and then increasing in a steady non-linear way. Moreover, both observed and simulated fluxes are higher during days with snowfall than during days without snowfall. Therefore, we can state that SnowPappus reproduces correctly the dependency of blowing snow flux on snow age and wind speed. In particular,



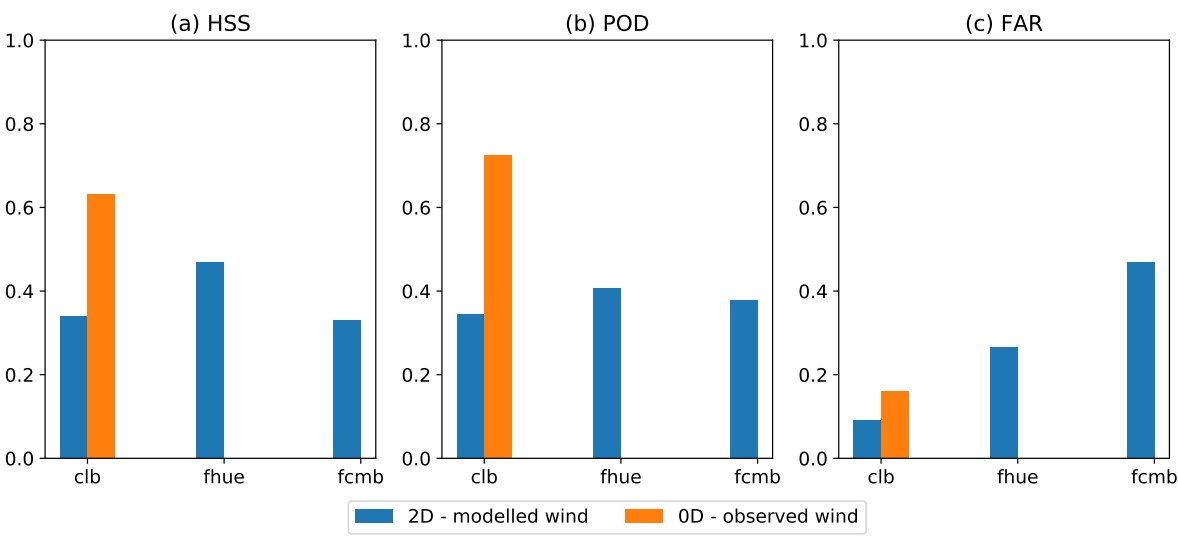

**Figure 10.** Blue bars: HSS, POD and FAR for the detection of blowing snow days in 250 m resolution simulations against Flowcapt data of ISAW stations of Huez (fhue) and Chambon (fcmb), and SPC data from *Col du Lac Blanc* (clb). Orange bars: same scores in point-scale simulations with the same SnowPappus configuration forced by observed wind speed (Run A in Fig. 9)

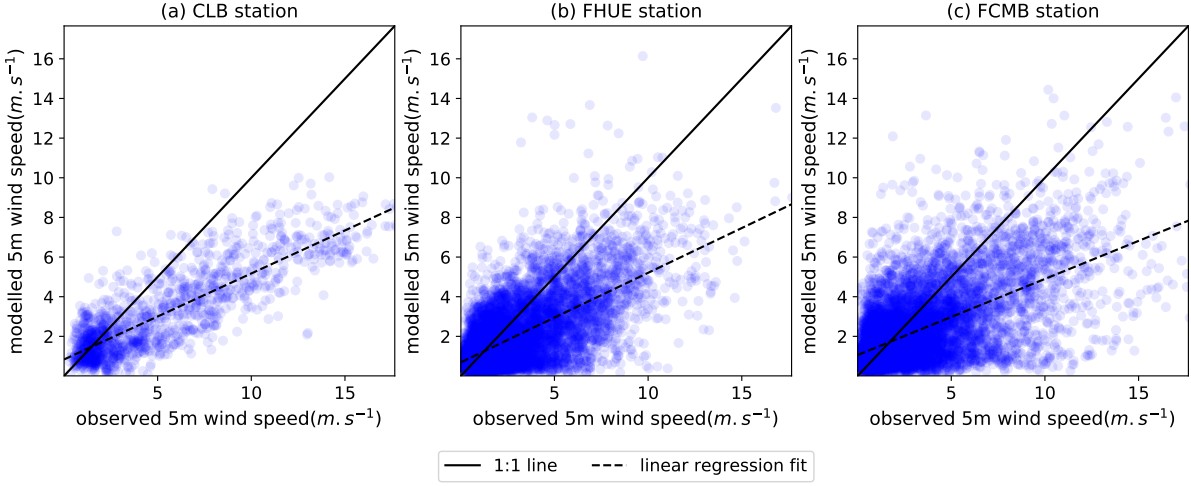

**Figure 11.** Comparison between observed wind at *Col du Lac Blanc* and at Huez and Chambon ISAW station and the 250m resolution DEVINE-modelled wind used as a forcing in the closest simulation points. Linear regression line fitting the modelled wind as a function of the observed ones are presented. The presented points are the ones used for blowing snow occurrence evaluation in Fig. 10. Wind observed data were filtered using the same algorithm as Le Toumelin et al. (2022)





the relationship between $Q_{t,int}$ and wind speed is very close to the observed one in the case of $d_m = 0.5$ for days without
snowfall.

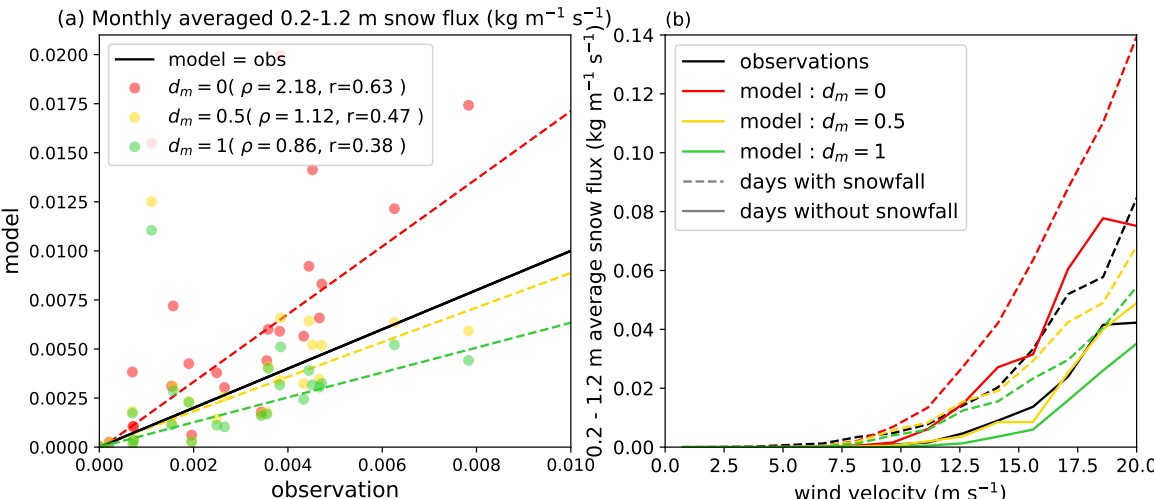

**Figure 12. (a)** Simulated monthly-averaged $Q_{t,int}$ at *Col du Lac Blanc* as a function of SPC-observed fluxes, for 3 different values of $d_m$. The 1:1 line representing equality between the model and observations is drawn in black. The dashed lines are Theil Sen regression fits (Wilcox, 1998) of each model configuration. Ratio model-observation of total cumulated fluxes ($\rho$) and correlation coefficient (r) are indicated in the legend. **(b)** Observed and modelled $Q_{t,int}$ (with the same model configuration as (a)) averaged by 5m-wind velocity intervals, as a function of the interval mean wind friction velocity. Dashed lines are computed from the data for days with snowfall and plain lines with the data for days without snowfall.

## 4.6   Numerical performance

The numerical choices of the SnowPappus scheme were driven by the goal of potential applicability in large-scale systems operated at hectometric resolution. In this section, we describe the numerical performance of the SnowPappus model to discuss its adequation with this objective.

Computation time was measured for the simulation domain covering the Grandes Rousses range (Sect. 3.1). Computations are done using one node of the current Meteo-France supercomputer. A node is made of 2 AMD Rome 2,2 Ghz CPU giving a total 128 computing cores and 256 Gb node RAM. In all of the following results we use a setting of one thread per computing core and will use the two words indistinctly.

   The SnowPappus blowing snow model is based on the MPI parallel libraries and a unpublished domain decomposition
already implemented in SURFEX for allocating parts of the simulation domain to computing cores, splitting the simulating domain in subdomain stripes. The algorithm is designed to balance as much as possible the number of grid cells between the different cores, but all the points with the same zonal coordinate are always gathered on the same core. Therefore, the



maximum number of subdomain stripes for an experiment is the number of lines of the domain because there must be at least one pixel line per stripe. For the Grandes Rousses domain the maximum number of stripes is 101 and so the maximum number

of parallel computing cores is 101. To time the execution of different parts of the code on a distributed set of cores, we use the DrHook profiling tool(Saarinen et al., 2005). To obtain the user run time of non overlapping and blocking code sections, we sum the maximum computing thread time for each code section.

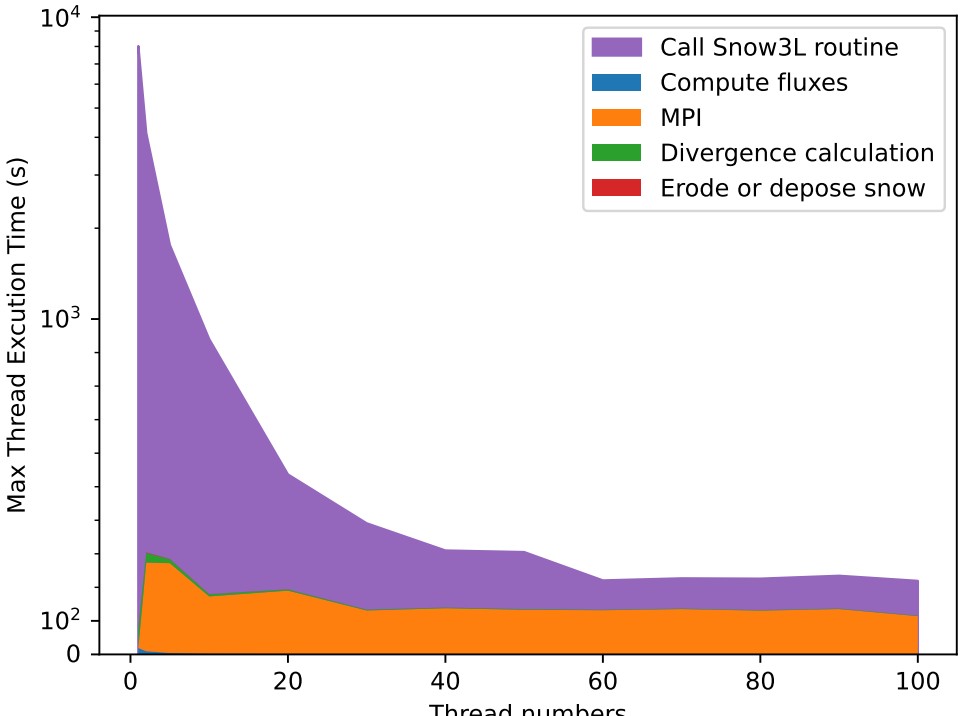

**Figure 13.** Visualisation of maximum thread execution time spent in the different parts of SnowPappus (named as in Fig. 5 ) and spent in others snow routines, in the SURFEX routine Snow3L. This time is mainly dedicated to run Crocus snow model.

Figure 13 shows the maximum thread execution time for each code section of the SnowPappus blowing snow model and compares these durations to the full snow routine Snow3L which contains the SnowPappus routine, the Crocus snow model

and the coupling instructions with the ISBA land surface model. The Snow3L execution time benefits greatly from parallel computing with a sharp decrease in execution time when computing on several threads. On the other hand, most of the computing time of the SnowPappus model is spent on MPI communications. The other parts of the SnowPappus routine have a comparatively negligible computing time. For our small experimental zone, the fastest simulation setting for the execution of



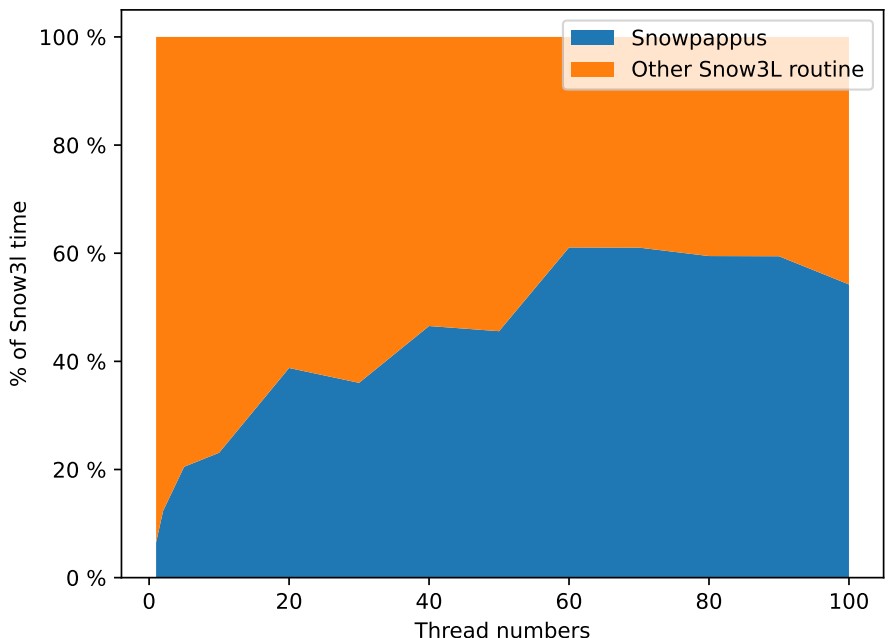

**Figure 14.** Visualisation of the proportion of time spent in the SnowPappus routine among the total snow-related snow3l routine computing time.

snow-related routines is using the maximum possible number of cores. However, most of the time gain is already obtained with
60 cores which seems to be the ideal configuration for this specific use case.

Figure 14 shows the proportional representation of the maximum computing thread time of the SnowPappus routine compared to the other snow-related routines in Snow3L. In a weakly parallelized case (core number <10), the proportional execution time of SnowPappus is low with less than 23% of computing time. This is caused by the weak parallelization creating large
subdomains per core, reducing the number of MPI communications in SnowPappus but increasing the number of computations per core for all snow routines, increasing the simulation time in heavy computing code sections as illustrated in Fig. 13. At worst, SnowPappus is using 60% of the total Snow3L time for 60 threads. For this configuration, most of the SnowPappus computing time is spent in MPI communications (Fig. 13) with other parts of the SnowPappus routine executing in negligible time. Indeed, classic SURFEX computing routines benefit greatly from parallelization with simulations points being computed
independently of one another. In SnowPappus however, we create a dependency on the neighbourhood simulation point. Logically, increasing the number of subdomains increases the number of blocking MPI communication. This can lead to an increase in MPI communication and waiting time, which become the limiting factor at high thread numbers. MPI communication time



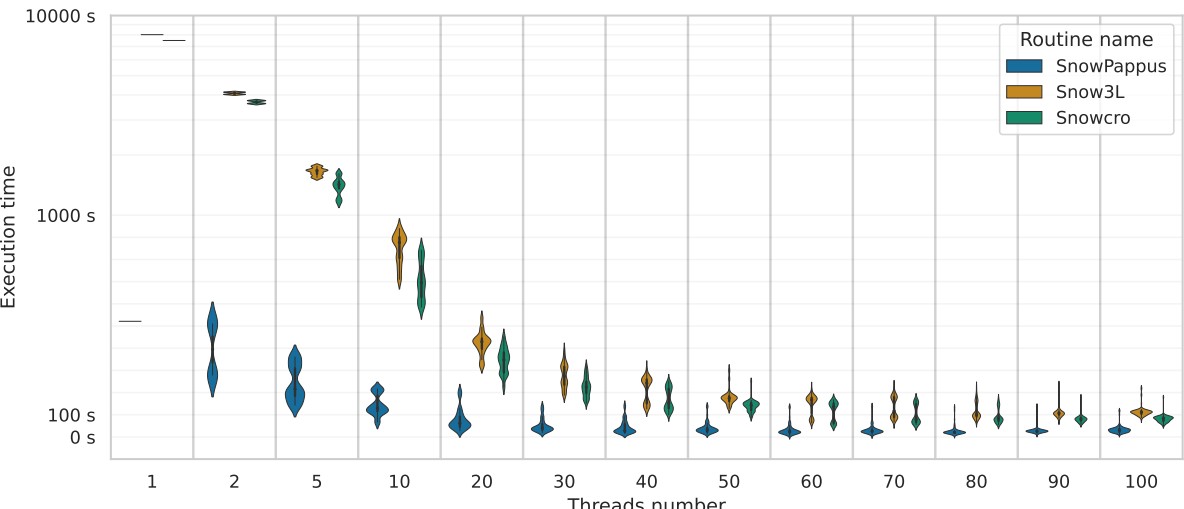

**Figure 15.** Visualisation of execution time for different threads run and various Fortran routines. This violin plot represents the distribution of execution time of various simulation routines according to different given threads number. The visualization is using a sym-log scale for the y-axis. We see on some distributions (ex 70 threads) a bimodal distribution for the execution time of snow3l and Crocus. This artifact is caused by the threads allocating algorithm cutting the simulation domain in an uneven number of points per thread. Some threads have more simulation points, they are spending more time on computation.

depends on the amount of data to communicate between threads, itself depending on domain shape, whereas waiting time depends on the equity of work sharing among computing threads.

Figure 15 shows the execution time distribution among threads depending on the chosen run threads number. The bimodal distribution obtained for some threads is due to the stripe domain decomposition of SURFEX, leading to a group of threads running on twice the number of points than other ones, with a direct impact on the computing time. Beyond this limitation, this graph also illustrates a high spread between the longer thread and the faster one for each run due to a large variability of snow coverage and snow transport between the points.





## 5 Discussion

### 5.1 Quality of point-scale flux prediction and comparison with other studies

We evaluated SnowPappus in terms of blowing snow occurrence against point-scale measurements. We also evaluated the simulated fluxes against observed fluxes in a part of the suspension layer. As model outputs are highly dependent on wind speed (Fig. 7), we first discuss the model skill of point-scale simulations at *Col du Lac Blanc* forced by observed wind speeds. This evaluation is carried over a long observation period of 10 years.

Long-term evaluation has already been performed for blowing snow occurrence by Vionnet et al. (2013) over the 2001-2011 period at the *Col du Lac Blanc* observatory. Their method (referred hereafter as VI13) was adapted in the Sytron operational system and evaluated on the same site on the 2015-2016 period, as a part of SURFEX v8 (Vionnet et al., 2018). As explained in Sect. 2.2.3, the SnowPappus blowing snow occurrence detection algorithm was also adapted from VI13 with a modification concerning treatment of blowing snow events with concurrent snowfall. We evaluated blowing snow occurrence in SnowPappus, Sytron and VI13 on the 2010-2020 period, reproducing the methods described in Vionnet et al. (2018) in order to get comparable results. All three methods are coded within the current version of SURFEX (v9). We interestingly obtained similar results as Vionnet et al. (2013). Moreover, we also evaluated Sytron and VI13 within the current version of SURFEX (v9) with the same dataset, and obtained similar results. All three methods give reasonable but perfectible detection scores with a large false alarm rate, even if (Vionnet et al., 2013) argued in a similar context that it may mainly concern events with very low simulated blowing snow fluxes. However, the previous evaluation of Sytron by Vionnet et al. (2018) showed better performances, with a perfect detection of blowing snow events at daily time scale at *Col du Lac Blanc* for one winter, which differs notably with our own Sytron evaluation. It remained the case when restricting our evaluation period to the 2015-2016 season (results not shown). Possible explanations for this discrepancy are (i) recent evolutions of SURFEX code (between v8 and the v9) interacting with blowing snow occurrence (ii) precise temporal window within the year of Vionnet et al. (2018) which was not given in their article and may differ from ours (from 1 December to 1 April) (iii) undescribed expert data cleaning performed on SPC data. We could retrieve the original simulation outputs of Vionnet et al. (2018) and applied our evaluation process to these data (see *code availability*). We obtained results very close to our own Sytron run with the original data. Consequently, hypothesis (iii) seems the most likely. Beyond this reproducibility issue, we can state that our model performs similarly to previously tested models in terms of blowing snow occurrence.

In terms of fluxes, evaluation of the monthly averaged blowing snow fluxes integrated between 0.2 and 1.2 m was performed with different parameterizations of the terminal fall speed $v_f$. It shows SnowPappus is able to simulate satisfactory average fluxes at *Col du Lac Blanc* if $v_f$ is adequately calibrated. However, simulated fluxes suffer from a high standard deviation of the error, leading to a low correlation between observed and simulated data. A particular outsider monthly value can be identified in Fig. 12, when the model overestimates the flux by a factor of 8-10. This corresponds to January 2018, when the simulated flux is mainly associated with two major blowing snow events separated by a few days. Wind speeds were high (up to 20 m s$^{-1}$), and the temperature was about -10°C. According to SAFRAN reanalysis, intermittent snowfall occurs during the event, providing continuously fresh and transportable snow that should be detected by the sensors (see the appendix Fig. A2). However, SPC





recorded no flux during this period, whereas the Flowcapt from the ISAW station of Huez (FHUE), only five kilometres away,
recorded high fluxes. Consequently, we hypothesize this outlier probably comes from a detector failure rather than from a
model overestimation. Moreover, Fig. 12a shows the flux dependency on wind speed and snow age is correctly reproduced by
SnowPappus. Overall, this point-scale evaluation of snow fluxes at *Col du Lac Blanc* shows SnowPappus provides good orders
of magnitude of blowing snow suspension fluxes and occurrence when an observed wind forcing is used. However, the model
exhibits a strong uncertainty at the event time scale.

A major strength of our evaluation is the long-term period encompassing 10 winter seasons while most previous studies
evaluating simulated blowing snow fluxes in seasonal snowpack conditions only considered a few blowing snow events taken
within a period of time of typically one month (Pomeroy and Male, 1992; Naaim-Bouvet et al., 2010; Vionnet et al., 2014).
Considering the dispersion of the monthly averaged fluxes we obtain, such short evaluations may be strongly biased. In partic-
ular, Naaim-Bouvet et al. (2010) evaluated a formulation of blowing snow flux which is very close to ours in the case of fresh
snow but assuming an infinite fetch. Contrary to us, their model overestimated the flux by an order of magnitude, which could
be due to the combined effect of the infinite fetch assumption and the specificity of the two considered events. Longer-term
evaluations of blowing snow fluxes could be found in Antarctica. For example, an evaluation of the monthly averaged flux
simulated by the MAR model in Antarctica was performed at 2 stations with respectively 2 and 8 years time series (Amory
et al., 2021). Correlation coefficients of 0.6 and 0.8 were obtained between observation and model. The correlation we obtain
is close or inferior to their results for the first station, although the results are not fully comparable (different seasonality of the
fluxes, use of simulated wind speed, etc.). Qualitatively, all the mentioned studies obtained a strong dispersion of model errors
which is consistent with our results.

However, some aspects of our methodology are questionable. Indeed, The use of Flowcapt and SPC data during precipitation
events for evaluation purposes is questionable. Indeed, some field evaluations suggest the less rounded snow particle shape in
these conditions leads to bias of the flux estimates (Sato et al., 2005; Trouvilliez et al., 2015). Moreover, both instruments do
not distinguish blowing snow particles from precipitating snowflakes, which by themselves can be responsible for fluxes of
typically 0.1 to 10 g m$^{-2}$ s$^{-1}$ (Vionnet (2012), Fig. 4.17b, Vionnet et al. (2017), Fig. 7). This may deteriorate our results for
blowing snow occurrence, as we did not take this possibility into account.

### 5.2 Sensitivity, added value and robustness of microstructure-dependent parameterizations

We performed blowing snow suspension fluxes evaluation making the parameter $d_m$, which influences the terminal fall speed,
vary within a range which is compatible with the current state of knowledge. Results show that (i) the value $d_m = 0.5$ allows for
simulations of realistic average fluxes and (ii) the value of the flux is strongly influenced by $d_m$, which can make its cumulative
value vary by almost a factor 3 in the explored range. $d_m$ controls the terminal fall speed of suspended particles, so it highlights
the extreme sensitivity of suspension to this parameter, which is for now imprecisely known.

On the other hand, results on the blowing snow occurrence at *Col du Lac Blanc* suggest that the differences between Vionnet
et al. (2013) method and SnowPappus do not lead to significant differences in the quality of the simulations. Consequently,
in the current state of knowledge, both methods can be used. Results also show that the wind-induced snow metamorphism





option seems to have only a very small effect on simulated blowing snow occurrence. It means it might not enhance the quality
of simulation in alpine environment, although complementary evaluations of its impact on the snow stratigraphy would be

required, in particular within 2D configurations of the model. Moreover, for the first time to our knowledge, we compared the
results of these parameterizations based on microstructure to a much more simple one where the wind speed threshold depends
only on whether or not snow was deposited less than 1 hour ago. It emphasizes that a well-calibrated constant threshold wind
speed performs as well or even slightly better than those parameterizations.

However, the above statements must be taken with caution as our study suffers important limitations. Indeed, only blowing

snow fluxes integrated between 0.2 to 1.2 m were evaluated, which tells nothing about what happens in the saltation layer and in
the saltation-suspension interface. Moreover, evaluations and calibrations on flux and occurrence were applied to only one site,
with particular climate and environmental conditions. The calibrations of $d_m$ and of the constant threshold wind speed may be
over-calibrated to this site and thus not directly valid or optimal at other sites. The microstructure-dependent parameterizations
of the threshold wind speed from Guyomarc'h and Mérindol (1998) might transfer better to other topographic and climatic

conditions but the lack of extended data to evaluate directly this process does not allow it to be affirmative. The absence of
added value of Guyomarc'h and Mérindol (1998) at *Col du Lac Blanc* may indicate that it does not captures the temporal
variability of threshold wind speed at this site, or that simulated snow surface properties are not relevant. This last case is
plausible due to uncertainties associated with forcings, and that occurrence is evaluated in a simple configuration with no
erosion or deposition.

## 5.3 Main limitations and improvement opportunities

SnowPappus model outputs depend on two major steps which are (i) computing the local blowing snow fluxes and (ii) com-
puting snow redistribution among grid points. Both steps are subject to large uncertainties which limit the accuracy of the final
snow cover simulations. In this section, we discuss the main sources of uncertainties. Snow redistribution in 2D simulations
has not been evaluated in this article, and will be the subject of a future study expected to provide complementary insights to

the following discussions.

### 5.3.1 Uncertainty on parameterizations

Local fluxes evaluation results suggest the complexity of parameterizations, at least for suspension fluxes and blowing snow
occurrence, is not directly linked with model accuracy. Indeed, our simple suspension model exhibited a very high sensitivity
to the snow particle effective terminal fall speed $v_f^*$, which is also involved in more complex suspension models (Bintanja,

2000; Vionnet et al., 2014). Thus, enhancing our knowledge of this parameter and its link to snow properties may allow a larger
improvement of suspension flux simulation than complexifying the models. Besides, simplification of the threshold wind speed
dependency on snow properties, as well as the inclusion of wind-induced snow metamorphism, did not change significantly
the blowing snow occurrence prediction skill. It could be explained by the fact that the interest in such parameterizations can
also be limited by intrinsic errors of the Crocus model in terms of surface properties. The hypothesis that a unique threshold

wind speed can be used for initiation and stop of the transport in given conditions may also affect this conclusion, as well as





the time step of the model which is longer than the duration of individual continuous transport events (Doorschot et al., 2004). Consistently with our results, it must be noticed that recent development in MAR by Amory et al. (2021), including among others a simplification of the threshold wind speed parameterization, led to an improvement of the model skill.

Finally, there are many "blind points" of the snow transport literature that limit the possibility to parameterize some phe-
nomena which may have a strong influence. For example, we could not find any study about the influence of the slope on snow saltation transport, whereas it was shown to influence sand transport (White and Tsoar, 1998) and steep slopes are common in complex terrain. Besides, a large part of wind-induced snow transport events occurs during snowfall (Vionnet et al., 2013). However, saltation fluxes and initiation were never studied during snowfall events to the best of our knowledge, whereas snowfall obviously changes snow cohesion, properties and interacts with grain ejection mechanisms. Finally, quantitative in-
formation about the action of transport on snow surface properties still lacks, despite some recent results on density evolution (Sommer et al., 2018; Amory et al., 2021). We think field or wind-tunnel measurements of snow SSA and density in snow deposition zones, as well as observation of their temporal evolution during blowing snow events, would maybe allow to test and improve the hypothesis we had to do for SnowPappus development.

### 5.3.2 Wind forcing

Simulated blowing snow fluxes increase quickly and not linearly with wind speed as shown in Fig. 7, and pointed out previously by numerous authors (Essery et al., 1999; Mann et al., 2000; Schneiderbauer and Prokop, 2011b, ,...) which found transport fluxes to be approximately proportional to power 4 to 5 of $u_*$. It consequently makes predicted blowing snow fluxes highly sensitive to the quality of wind forcings. In addition, poor results for blowing snow occurrence were obtained using 2-dimensional outputs forced by DEVINE modelled wind speed, which is partly explained by the important difference between
the value of this wind taken at the closest grid point and the local one. These results suggest the quality of the local wind forcing can easily become a major limiting factor for local blowing snow fluxes assessment. Moreover, it may become even more limiting for the prediction of the spatial erosion and deposition patterns (Musselman et al., 2015). The improvement of wind fields assessment in mountainous areas is beyond the scope of this study but has recently been investigated by several authors (Raderschall et al., 2008; Helbig et al., 2017; Dujardin and Lehning, 2022). Therefore, the possibilities of improvements in
snow transport modelling are fully dependent on the incoming advances in this area.

### 5.3.3 Spatial resolution

On the one hand, two-dimensional simulations on the Grandes Rousses test zone (see Sect. 4.2) showed that activating snow transport can have a very strong effect on snow accumulation in the most exposed zones. These effects can be of the same order of magnitude as the amount of annual precipitations in the region (1532 mm of precipitations recorded from 01 August
2018 and 01 August 2019 at the automatic weather station of Alpe d'Huez, coordinates 45.087833°N, 6.085667°E). It suggests that neglecting wind-induced snow transport at this scale is irrelevant for high alpine crests. On the other hand, the width of areas strongly influenced by transport encompasses typically a few grid points. Thus we can expect results to be hampered by strong numerical errors and discretization issues, as suggested by previous studies at 25-200 m resolution (Lehning et al.,





2008; Bernhardt et al., 2009; Grünewald et al., 2010). It highlights also the difficulty of carrying local evaluation of a distributed
model at such resolution in complex terrain, as gridded meteorological forcings, in particular wind, and consequently simulated
snow conditions do not necessarily correspond to local ones.Further evaluations of SnowPappus using distributed observations
from remote sensing (snow depth, snow cover) will provide further insights into the influence of spatial resolution on model
results.

### 5.4 Limits of applicability

Several of the choices we made in the model development were adapted to our constraints in terms of spatial and temporal
resolution (around 250 m, 900s) and the type of environment (complex terrain with seasonal snowpack). These choices could
become unsuitable in different contexts. Indeed, high-frequency fluctuations of the turbulent wind cause blowing snow flux
to fluctuate as well, over time scales inferior to 1 minute (Doorschot et al., 2004; Aksamit and Pomeroy, 2018). As their
dependency on wind speed is highly non-linear, the observed relationship between averaged wind speed and blowing snow
fluxes must depend on the time step on which measures are averaged. Most of the parameterizations we used are inferred from
observations averaged on a 7.5 - 10 min period (Pomeroy and Gray, 1990; Pomeroy and Male, 1992; Vionnet et al., 2012)
which is close to the model time step. Thus, using these parameterizations at a much smaller time step may introduce errors.

We can also note that we mostly rely on studies performed in regions with a seasonal snowpack (Pomeroy and Gray, 1990;
Pomeroy and Male, 1992; Naaim-Bouvet et al., 2010; Vionnet et al., 2013) which may also be biased in other conditions, like
Antarctica or Arctic environments. Finally, moving to higher resolutions or to areas where topography varies on a much larger
scale than the spatial resolution would require treating explicitly the influence of fetch, as it would cover several grid points.
Moreover, coming back on reasoning done in Sect. 2.3.3, simulations at higher resolutions may benefit more from the solving
of the 3D advection-diffusion equation.

### 5.5 Applicability at large scale

The technical possibility to apply SnowPappus in large-scale simulation was one of the target of this work. The detailed
evaluation of its computing performance provided in Sect. 4.6 shows that computing time of SnowPappus physical routine is
negligible compared with the one of Crocus. However, scalability issues caused by MPI communications in a highly parallel
environment are an important limitation. Reducing the number of communications and homogenizing the workload among
threads would alleviate this issue. To achieve it, optimal sub-domain cutting could use squares instead of stripes or even take
advantage of snow cover duration (SCD) climatology, as the numerical cost of Crocus-SnowPappus increases strongly with
SCD. Another important bottleneck is the duration of SURFEX Input/Output (I/O) operations, which are not yet parallelized.
This well-known problem was highlighted by Balaji et al. (2017). Despite these current limitations, we were able to perform a
yearly simulation on a domain covering the full French Alps (see Fig. 6) in 17h of simulation time using only one computing
node. It means daily operational simulations implying 8 days of simulations in the current French system (Morin et al., 2020b)
would require only 30 min computing time on one node, which is affordable. Therefore, the main criteria for using SnowPappus





in an operational system in a near future will be our ability to demonstrate its added value on the snow cover simulations rather than computation time limitations.

## 6    Conclusions

This paper presents SnowPappus, a new blowing snow model coupled with the Crocus state-of-the-art snow scheme. It aims
to be part of a future operational system running distributed snowpack simulations over the entire French mountains at 250-m grid spacing. SnowPappus is a simple model computing blowing snow fluxes using semi-empirical parameterizations to represent saltation and solving suspension in a one-dimensional stationary state, as models like PBSM (Pomeroy et al., 1993) or SnowTran3D (Liston and Sturm, 1998). It includes newer results on the terminal fall speed of snow particles (Naaim-Bouvet et al., 2010; Vionnet, 2012) which have a strong influence on the simulated snow fluxes, and a parameterization of the
threshold wind speed based on Crocus-simulated microstructure properties. Several options are available to represent threshold wind speed, suspension, sublimation and wind-induced snow metamorphism. MPI parallelization handles the data sharing between neighbouring points required to compute snow redistribution on parallel computers. Performance tests show that Crocus coupled with SnowPappus is able to run a simulation over the full French alps during an entire snow season within a reasonable computation time. However, MPI communications and waiting times raise significantly the computation time.

Local evaluations of suspension snow flux and blowing snow occurrence using observed wind fields to drive SnowPappus were also performed. They show that SnowPappus is able to simulate reasonable average suspension fluxes if the effective terminal fall speed of suspended snow particles is adequately calibrated, and that blowing snow occurrence is satisfactorily captured. However, the simulation outputs have a strong uncertainty, which is coherent with previous results obtained with other models. Numerous badly known physical parameters and understudied parameters limit improvements of the used parame-
terizations. Moreover, uncertainty linked to parameterization combines with uncertainties on forcing wind speed. Therefore, it may lead to local fluxes being strongly different from the simulated one. They will have to be understood more as a "first guess" than as a quantitative estimate.

      Future work will include an evaluation of the simulated snow spatial distribution against satellite data (snow depth maps from satellite stereo imagery, snow cover maps for satellite optical imagery, ...). Depending on the results of these future
studies, various improvements of SnowPappus could be done, such as representing the effect of subgrid topography on transport fluxes (Bowling et al., 2004) or testing different parameterization of deposited snow properties. Finally, the high and partly unavoidable uncertainty in the simulation outputs also stresses the need to include ensemble simulations and data assimilation in the future system.

*Code and data availability.* The SnowPappus blowing snow model is developed in the framework of the open-source SURFEX project.
The source files of SURFEX code are provided at https://doi.org/10.5281/zenodo.7687821 to guarantee the permanent reproducibility of results. However, we recommend that potential future users and developers access the code from its Git repository (http://git.umr-



cnrm.fr/git/Surfex_Git2.git, last access: 1 mars 2023) to benefit from all tools of code management (history management, bug fixes, documentation, interface for technical support, etc.). This requires a quick registration, and the procedure is described at https://opensource.umr-cnrm.fr/projects/snowtools_git/wiki/Install_SURFEX (last access: 1 mars 2023). The version used in this work is tagged as SnowPappus-

v1.0. A user manual, describing the SURFEX namelist options related to SnowPappus is available at https://doi.org/10.5281/zenodo.7681340. More general information about SURFEX use can be found at https://opensource.umr-cnrm.fr/projects/snowtools_git/wiki and https://opensource.umr-cnrm.fr/projects/snowtools_git/wiki (last accesses: 1 mars 2023) The DEM used in this study originate from RGE Alti® website. They can be downloaded freely at https://geoservices.ign.fr/documentation/donnees/alti/rgealti. ISAW Network stations raw data are freely available on http://iav-portal.com/index.php?nav=iodmisawlist&lang=en&search=¢er=&sort_field=center&sort_asc=1. The data from *Col du Lac*

*Blanc* station is available at https://doi.org/10.17178/CRYOBSCLIM.CLB.COL.csv and are described by Guyomarc'h et al. (2019). AROME downscaled wind forcing used for simulations on Grandes Rousses test zone is available at https://doi.org/10.5281/zenodo.7681661. input data , namelists and instructions to run the model and produce most of the plots and simulations presented in this paper are available on download at https://doi.org/10.5281/zenodo.7681551. In the same folder, codes generating some additional results not shown in the article are available (see Sect. 3.5 and 5.1)

**Appendix A:  Supplementary figures**

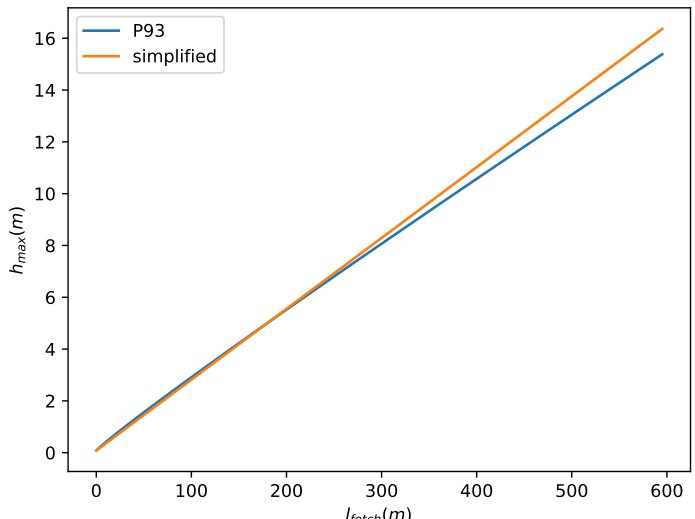

**Figure A1.** Comparison between the height $h_{max}$ given in Sect. 2.3.8 used as an upper bound for suspension transport (blue curve) and the one computed with Eq. 10 from Pomeroy et al. (1993) (orange curve). Both heights are plotted as a function of the fetch distance as defined in Sect. 2.3.8.



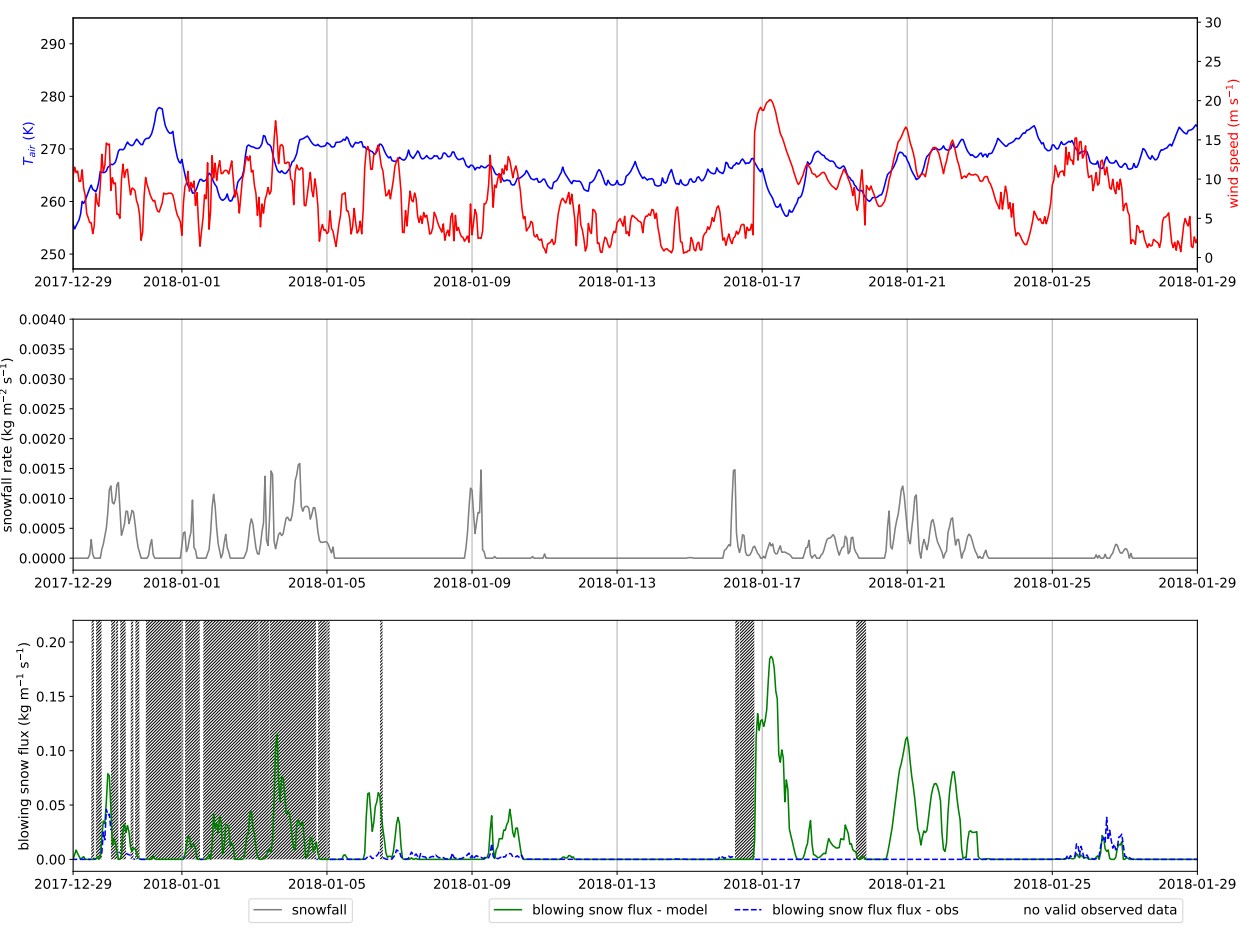

**Figure A2.** Modelled and SPC-measured 0.2 -1.2 m blowing snow flux during January 2018. SAFRAN-modelled air temperature and solid precipitations, as well as measured wind speed are also represented. This month is a clear outlier in the comparison between observed and modelled fluxes presented in Fig. 12. Here we see that, during the two main modelled events of the month (17 January and 21 January), no flux is detected by the SPC, despite high wind (15-20 m s$^{-1}$) , negative temperatures in the previous weeks, preventing the formation of an ice crust, and co-occurring heavy snowfall, which should bring continuously transportable snow. It seems to indicate either an undetected SPC deficiency or big errors in the forcing fields during this month, rather than a model failure.





**Appendix B: Conversion formula between old and new Crocus microstructure formalism**

In the current version of SURFEX, the microstructure of Crocus is described with two prognostic variables : the optical diameter $D_{opt}$ and the sphericity $s$. In this article, we often refer to the dendricity $d$ and grain size $g_s$ which were used in older versions of Crocus (Vionnet et al., 2012). Carmagnola et al. (2014) proposed formulas linking $D_{opt}$, $s$, $g_s$ and $d$. However some errors were detected in this work, leading us to use another formula to get grain size from $D_{opt}$ and $s$. This formula is presented here:

$$g_s = 2\frac{D_{opt} - 2\alpha(1-s)}{1+s} \tag{B1}$$

The formula from Carmagnola et al. (2014) is used to compute the dendricity $d$ from $s$ and $D_{opt}$. Details about this new conversions formulas and their impact on Crocus metamorphism will be described in a paper in preparation.

**Appendix C: Properties of deposited snow in case of simultaneous snowfall and wind-driven redeposition**

We consider here the case when during a time step, a mass $m_{SP}$ and $m_{BS}$ (kg m$^{-2}$) of snow coming respectively from solid precipitation and wind-driven redeposition has to be added to the snowpack during a simulation time step. Optical diameter $D_{\mathrm{BS}}$ and sphericity $s_{\mathrm{BS}}$ sphericity $s_{\mathrm{BS}}$ of the deposited wind-blown snow are given in Sect. 2.6. Solid precipitation sphericity $s_{\mathrm{SP}}$ and optical diameter $D_{\mathrm{SP}}$ are computed as in the default Crocus configuration, described in Vionnet et al. (2012). In this case, a layer of total mass $m = m_{\mathrm{SP}} + m_{\mathrm{BS}}$ is added to the snowpack. Its properties $s$ and $D$ are given by the weighted average of blowing snow and solid precipitation properties:

$$s = \frac{m_{\mathrm{SP}}s_{\mathrm{SP}} + m_{\mathrm{BS}}s_{\mathrm{BS}}}{m_{\mathrm{SP}} + m_{\mathrm{BS}}} \tag{C1}$$

$$D = \frac{m_{\mathrm{SP}}D_{\mathrm{SP}} + m_{\mathrm{BS}}D_{\mathrm{BS}}}{m_{\mathrm{SP}} + m_{\mathrm{BS}}} \tag{C2}$$



*Author contributions.*  A.H. and M.B. developed jointly the SnowPappus code with equal contribution. A.H. had a special involvement in coding the mass balance routine and MPI parallelization. M.B. redacted the introduction, discussion and conclusion of the article. He surveyed the bibliography, and made theoretical choices and most of the technical work concerning saltation and suspension snow flux simulations, and general evaluation of the model. A.H. did it for mass balance, sublimation and numerical performance tests. M.L. supervised the work, provided technical support on Crocus and SURFEX, ran Sytron simulations and was extensively involved in the proofreading process. L.LT provided wind forcing for the two-dimensional simulation and helped in writing the article. V.V. participated in theoretical discussions, in particular about Sytron, and proofread the article. M.F. participated in debugging and helped code development.

*Competing interests.*  The authors declare that they have no conflict of interest.

*Acknowledgements.*  This work was conducted using mainly Meteo France office and computing ressource. Main fundings are SENSASS project funded by the Region Auvergne Rhône-Alpes (France) and CDSN PhD grant. We thank **Hervé Bellot** for providing Blowing snow flux data at col du Lac Blanc, as well as **Yannick Deliot**, **Hugo Merzisen** and **Isabelle Gouttevin** who provided advice and technical information about their use and about *Col du Lac Blanc* experimental site in general. We also thank **Rafife Nheili** for her help and participation in coding the MPI parallelization routine. Generally, we thank the whole Centre d'Etude de la Neige modelling team, for multiple fruitful discussions and diverse technical support. CNRM/CEN and LECA are part of LabEx OSUG@2020.



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
