# Peer review of "SnowPappus v1.0, a blowing-snow model for large-scale applications of Crocus snow scheme"

_Geoscientific Model Development, 2023_

## Author Comment (AC1)

In the following, author comments are in black and reviewer comments are in **blue.** When we present text passages from the revised manuscript, they appear in "*italics*".

**There are many 1D models of blowing snow, several 2D models (operating on decametre grid scales as far back as the 1990s), and even 3D coupled snow-atmosphere models. An operational model that can be run on large scales is certainly of interest. The authors give a fair summary of previous work and a nearly complete description of the model being introduced.**

We first would like to thank the reviewer for their constructive comments. In the following, we will give a detailed answer to each of them.

**The paper is very long, however, for the amount of original material being presented; some of this is due to repetition and poor organization of material.**

We understand the major concerns of both reviewers about the length and lack of clarity of the paper. In order to address this issue, we propose to extensively  modify the structure of the paper to improve its readability. We will split Section 2 (Methodological choices) into two separate sections. The first one will be dedicated to a literature review largely reduced in length by focusing on the topics for which an added value is provided in SnowPappus compared with existing works. It includes discussions about the representation of saltation flux and lower boundary condition for suspension flux and terminal fall speed parameterizations. This will allow to make the logics of the manuscript clearer. For processes that are simply implemented in SnowPappus following existing models or previous literature, the literature overview will be considerably shortened, or even suppressed if it is not useful for the remaining part of the article, such as the discussion on sublimation (L449-452). Then, a second section will be dedicated to a concise model description without any theoretical interruptions between the description of the different implementations. We believe this organization will help the reader to understand more quickly our modelling choices.

In addition, several parts of the manuscript will be strongly shortened, primarily the description of numerical performance, various repetitions and unnecessary information or discussion will be withdrawn and a more concise writing style will be adopted when possible. Besides, the modification of the paper structure, which splits the section "Methodological choices" into 2 parts do not result in additional length, as the number of subsubsections does not changes a lot and repetitions will be avoided. Overall, the planned changes will allow to suppress Fig. 15 and move Fig. 14 in the appendix and to reduce overall text length by 20% (without taking appendix into account).

In the following, we will first list the text passages that will be strongly shortened, or withdrawn, in their order of appearance and identified by their section number and if necessary line number in the preprint. Then, we present a revised text outline to apply the described changes. New (sub)sections are written with their titles underlined, and there contains in plain text. Parts of the outline in which the organization remain identical with the preprint's outline are marked as UNCHANGED, although some length reductions have been also applied in these parts.

**shortened text passages (with paragraph and line number from the initial manuscript):**
   2.1 Target, opportunities and constraints

2.2.2, wind profile

2.3.2 Blowing snow trajectories and transport modes

2.3.4 L257-263 discussion on the influence of particle size distribution on the suspension transport

2.3.3 L207-241 - Different types of transport models

2.3.7 L334-358 equations of  simple saltation parameterizations P90 and S04

2.3.8  Influence of fetch distance

2.5 L502-508 Influence if snow transport on snow surface properties (state of the art)

3.2 L 560-570 Wind downscaling description

3.5 L 623-634 Methods for blowing snow occurrence measurements

4.6 L734-739, domain decomposition for parallel computing

5.1 L799-805 Discussion on the outlier in blowing snow fluxes evaluation

withdrawn text passages :

2.4 L449-452, Sublimation

4.5 L712-717, Evaluation of blowing snow fluxes at *Col du Lac Blanc,* discussion on the outlier (repeated in the discussion)

**new outline in the revised manuscript :**

1. Introduction

      General introduction followed by the target, opportunities and constraints of the development of SnowPappus (corresponds to Sect. 2.1 in the preprint)

2. Blowing snow flux computation : state of the art

2.1 Blowing snow occurrence

      Useful theoretical background for blowing snow occurrence detection (Sect. 2.2.1 in the
      preprint)

2.2 Horizontal blowing snow fluxes

      2.2.1 Notations and geometric considerations

      2.2.2 Blowing snow particle trajectories and transport modes

            State of the art on the trajectories of blowing snow particles and transport modes,
            focused on saltation and suspension (corresponds to Sect. 2.3.2 in the preprint)

      2.2.3 Suspension transport modelling

            -Existing types of suspension transport models (part of Sect. 2.3.3 in the preprint)
            -Literature review on the effective terminal fall speed (mainly informations in Sect.
            2.3.5 of the preprint)

      2.2.4 Transition between saltation and suspension

            - The way it is treated in other models
            - State of the art on this transition zone
            - Problems of definition of the lower boundary condition for suspension transport
             (corresponds mainly to Sect. 2.3.6 of the preprint)

      2.2.5 Simple saltation models

            -Short description of Sorensen et al., 2004 (S04) and Pomeroy et al., 1990 (P90)
            saltation parameterizations
            -Discrepancies between S04 and P90 and discussion of the possible causes
            (informations in Sect. 2.3.7 of the preprint)

3. Model description

3.0 Crocus description

      Very short description of Crocus snow model

**3.1 Blowing snow occurrence**

Equations of the different options implemented in SnowPappus for threshold wind speed for transport (Sect. 2.2.3 in the preprint)

**3.2 Horizontal blowing snow flux**

**3.2.1 Suspension transport**

- Reasons for the choice of the model type
- Logarithmic wind speed profile (Sect. 2.2.2 in the preprint)
- Equations for suspension transport in SnowPappus (corresponds to Sect. 2.3.4 in the preprint)
- Parameterization of terminal fall speed used in SnowPappus (informations in Sect. 2.3.5 of the preprint)
- Maximum height of suspension transport as a function of fetch distance (part of Sect. 2.3.8 of the preprint)

**3.2.2 Saltation transport and transition with suspension**

- Description of the two options (using S04 and P90) of flux computation in the saltation zone and the transition with suspension in SnowPappus (included in Sect. 2.3.7 of the preprint).
- Influence of fetch distance on saltation transport in SnowPappus (included in Sect. 2.3.8 of the preprint, with Fig. 3 appearing)

**3.3 Sublimation**

Sublimation options in SnowPappus

**3.4 Mass balance**

Mass balance implementation in SnowPappus

**3.5 Influence of snow transport and deposition on snow surface properties**

- Properties of deposited snow
- Blowing snow induced snow metamorphism

**3.6 Implementation in SURFEX**

How SnowPappus is included in SURFEX code (mainly Sect. 2.7 of the preprint, but includes the description of domain decomposition for parallel computing)

**4. Evaluation : methods**

UNCHANGED (with length reductions)

**5. Results**

mainly UNCHANGED but addition of 1 paragraph (see below) and Fig. 14 moved in appendix, as partly redundant with Fig. 13 .

**5.1 Comparison of saltation parameterizations**

Comparison of blowing snow fluxes obtained with S04 and P90 implementations in SnowPappus and implications for the comparison of S04 and P90 (corresponds to the end of Sect. 2.3.7 in the preprint).

**6. Discussion**

UNCHANGED (with length reductions)

**7. Conclusion**

UNCHAGED

**Having said that "Blowing snow occurrence evaluation showed SnowPappus performs as well as a currently operational scheme" in the abstract, there needs to be a statement about the need and benefits for SnowPappus.**

We agree with the reviewer that the needs and benefit of SnowPappus were insufficiently stressed out in the abstract. We thus propose to replace L 2-4 by "*Thus, the evolution of operational snow modelling systems towards 100-500 m resolutions requires representing this process at these resolutions, over large domains and entire snow seasons. We developed SnowPappus, a parsimonious blowing snow model coupled to the Crocus state-of-the-art snow model able to cope with these requirements.*"

Besides, the SYTRON operational scheme was indeed already available to simulate wind-induced snow transport coupled to Crocus, but it is designed to work on idealised geometries, where an altitudinal band of a mountain range is represented by 8 slopes with 8 orientations, where snow can be transported from one slope to the opposite one (Vionnet et al., 2018). Thus, this model is not suitable for distributed simulations. It is however used operationally for the detection of blowing snow occurrence. To clarify the added value of SnowPappus compared to this scheme, we propose to make less ambiguous the above-mentioned abstract phrase by changing "Blowing snow occurrence evaluation showed SnowPappus performs as well as a currently operational scheme" in "*Evaluations showed SnowPappus performs as well as currently operational scheme SYTRON in terms of blowing snow occurrence detection, while the latter does not give access to a spatialized information*" .

Moreover, mention to SYTRON and its difference with SnowPappus will be discussed briefly in the end of the introduction, and the goal and strategy of the paper will be clarified. Given our reorganisation of the end of the introduction (see our response on manuscript length and organization), we present below the whole end of the introduction in the revised manuscript, and highlight in **bold** the sentences that address specifically these two issues.

"*In the above-mentioned context of increasing resolution of snow modelling systems, the long-term project of CNRM aims at performing simulations with Crocus at the scale of the French Alps at 250m resolution in an operational purpose, associated with a data assimilation framework requiring ensemble runs of 50-100 members (Largeron et al., 2020; Cluzet et al., 2021). The 250 m resolution allows a trade-off between the need for precisely representing slopes and aspects, influencing mass and energy balance of the snowpack, and the expected computational cost. In this context, a numerically efficient representation of wind-induced snow transport that can be coupled to Crocus simulations is lacking while this is necessary to better account for its impact on avalanche forecasting over French mountains. **Two blowing snow scheme coupled with Crocus exist yet : SYTRON (Vionnet et al., 2018) and Crocus-Meso-NH (Vionnet et al., 2014). However, both are unadapted to this geometry and resolution.***

***Thus, the goal of this paper is to describe and present first evaluations of a novel blowing snow scheme, SnowPappus, coupled to Crocus and able to be included in the aforementioned large-scale simulation system. Point-scale evaluation of blowing snow flux will be presented to discuss the modelling choices**. In order to avoid a prohibitive computational cost, this scheme shall not be much more computationally intensive than the Crocus model itself, and it will be forced with 2D*

*wind fields downscaled from NWP systems rather than coupled with 3D high resolution atmospheric model.*"

**There is no evaluation of the model in large-scale applications beyond demonstrating its computational feasibility.**

This concern was also raised by Reviewer #1. In this model description paper, we considered that the evaluation of blowing snow fluxes was the main topic to address as this is the most direct observation of the newly simulated processes. These evaluations do not allow to determine if the spatial patterns of snow properties simulated are improved by the blowing snow module. This limitation is explicitly mentioned L858-860 "Snow redistribution in 2D simulations has not been evaluated in this article, and will be the subject of a future study expected to provide complementary insights to the following discussions". Such evaluations are definitely necessary but they raise major methodological difficulties which justify a dedicated and separated paper.

First of all, in large scale simulations, several sources of errors are superposed, making it hard to determine if differences between model and observations come from a misrepresentation of wind-induced snow transport or from other sources of errors. In particular, precipitation forcing at high altitudes suffer from high uncertainty partly due to a largely unexplained spatial variability in NWP precipitation outputs, a severe lack of observations to constrain meteorological analysis systems, and various issues in radar precipitation measurements over complex terrains. It usually leads forcing errors to prevail in snow simulation systems that are not forced by local observation (Raleigh et al., 2015; Schlögl et al., 2016; Günther et al., 2019). Then, as point-scale observations are often not representative of the spatial scale of a simulation system (i.e. 250 m resolution in our case), evaluations have to rely on satellite observations which also involve complex retrieval methodologies and an appropriate consideration of associated uncertainties. Finally, methodological developments are required to compare simulated and observed snow maps as consistent spatial patterns may be simulated with slight localization inaccuracies, making traditional evaluation scores often unadapted (Gilleland et al., 2009).

Therefore, we also prepared a dedicated evaluation paper of the spatial distributions of snow height and Snow Melt Out Date against satellite stereo-imagery (Deschamps-Berger et al., 2020) and optical products (Gascoin et al., 2019). In this evaluation, the contributions of model and forcing uncertainties in simulation errors are quantified and compared. As expected, the forcing uncertainties prevail. This paper (Haddjeri et al.) will be submitted in September 2023 to The Cryosphere and will be complementary to the model description and first evaluations provided in this GMD model description paper.

Moreover, we would like to stress that we follow a strategy similar to other published wind-induced snow transport models operating at similar or lower resolution. Due the same methodological challenges, the corresponding publications did neither describe evaluation of the simulated spatial patterns of snow depth or surface properties (Gallée et al., 2001; Amory et al. 2021; Sharma et al., 2021). These spatial evaluations, when performed, were sometimes conducted in a separate paper (Gerber et al., 2023). Smaller scale blowing snow models were often described along with a purely qualitative evaluation of snow depth patterns on areas ranging from hundreds of meters long transects to a few kilometre square test zones (Liston and Sturm (1998); Liston et al. (2007); Vionnet et al. (2014)), and sometimes without any spatial evaluation (Essery et al., 1999).

Conversely, we provide in the present paper quantitative evaluations of the blowing snow flux and transport occurrence simulated with SnowPappus against a 10-years long observation time series at Col du Lac Blanc. As stressed in our introduction, such evaluations are very unusual in the currently available literature (except in Amory et al., 2021 in Antarctica), so concerning this direct variable, the evaluations provided in our paper are more advanced than similar literature on blowing snow models. These analyses allow to evaluate and discuss the parameterization choices done to compute blowing snow occurrence and fluxes and to question the interest of microstructure-based parameterizations for which the added value had never been assessed. This is also an important added value of our paper.

We propose to clarify the goal and strategy in the introduction. Given the introduction will be merged with Sect. 2.1 (see above our response on manuscript length and organization), we give here the full paragraphs including this clarification. The part of the text which is the most dedicated to it appears in bold.

"*In the above-mentioned context of increasing resolution of snow modelling systems, the long-term project of CNRM aims at performing simulations with Crocus at the scale of the French Alps at 250m resolution in an operational purpose, associated with a data assimilation framework requiring ensemble runs of 50-100 members (Largeron et al., 2020; Cluzet et al., 2021). The 250 m resolution allows a trade-off between the need for precisely representing slopes and aspects, influencing mass and energy balance of the snowpack, and the expected computational cost. In this context, a numerically efficient representation of wind-induced snow transport that can be coupled to Crocus simulations is lacking while this is necessary to better account for its impact on avalanche forecasting over French mountains. Two blowing snow scheme coupled with Crocus exist yet : SYTRON (Vionnet et al., 2018) and Crocus-Meso-NH (Vionnet et al., 2014). However, both are unadapted to this geometry and resolution.*

***Thus, the goal of this paper is to describe and present first evaluations of a novel blowing snow scheme, SnowPappus, coupled to Crocus and able to be included in the above-mentioned large-scale simulation system. Point-scale evaluation of blowing snow flux will be presented to discuss the modelling choices***"

In addition, L858-860 in the second part of the discussion will be replaced by :
" *Snow redistribution in 2D simulations has not been evaluated in this article, due to several methodological challenges including dealing with the superposition of errors coming from the precipitation fields and finding relevant metrics. It will be the subject of a future study expected to provide complementary insights to the following discussions.*"

**But the main problem is that this manuscript requires major copy editing for clarity and concision, beyond what can be achieved in review**

As mentioned above, we propose a revised manuscript with very substantial changes, including a revised structure of the literature review and model description to improve the clarity of the manuscript. The length is also reduced by 20%.

**I will expect to have more scientific comments on an easier to read revision, but here are a few corrections that I have noted:**

**There is, I think, a u\* missing in equation 13**

Indeed, the start of the equation will be corrected. The corrected one will start with :
$Q_{susp}=c\_r z\_r u\_*/(k (1- gamma(u\_*) )$

**For reproducibility, quoting the values of F(T), A and B in equations 25 and 26 would save the reader some cross referencing. \mu_{satc} and Rh_i are essentially the same thing. Some signposting would help anyone hoping to find the SnowPappus code in the extensive SURFEX repository.**

For reproducibility, we propose the following changes in the revised manuscript
- L 462, "A, B constants defined in (Gordon et al., 2006)" will be replaced by "*,A = 0.0018 and B=3.6*"

- in Eq. 26, 1-RH_i will be replaced by  *\mu_{satc}*

- F(T) expression, which is quite complex, will be provided in appendix, to limit the increase of the size of the manuscript.

- -A Readme with path towards the most useful SURFEX routines for SnowPappus will be provided with the SURFEX repository.

**Wind is measured at Huez and Chambon, so why no simulations forced by observed wind speed in figure 10?**

We decided not to perform simulation with observed wind speed at Huez and Chambon as the wind data do not have the same quality as at the *Col du Lac Blanc* observatory. These date would required more treatment in order to get continuous wind time series. In particular, wind sensors are located at a fixed 3,5 m height above ground, so their height must be deduced from measured snow height. They are sometimes of very low quality, with numerous discontinuities and gaps.

**Red points masked by the legend in figure 12a appear pink. Move the legend.**

Will be fixed.

**Vincent Vionnet is a co-author of this paper, so it should not be necessary to be so speculative about differences from Vionnet et al. (2018) in section 5.1.**

We would like to apologize for this unclear formulation that did not correctly reflect our clear understanding of this issue. In fact we _were able to_ retrieve the original simulation outputs from Vionnet et al. (2018) and found there was not any issue about it. As said L793 we "applied our evaluation process to these data[ ...] We obtained results very close to our own Sytron run with the original data".  Regarding the dates, we were able to obtain the information on the precise temporal window used in Vionnet et al. (2018), which is from 01/11 to 15/04. Therefore, their temporal window is larger than ours, so it is not possible that the change in the temporal window by itself would cause a perfect detection in their case and not in ours. As a consequence, we are sure the reproducibility issue comes from an unreproducible data post-processing, that was applied when compiling the results at the daily time scale in Vionnet et al. (2018). This step was not applied in our evaluation dataset. To clarify this and taking into account the reviewer's call for concision, we propose to replace L789-794 by

"_We were able to retrieve the original simulation outputs of Vionnet et al. (2018) and applied our evaluation process to these data (see code availability), obtaining results very close from ours . Thus, after discussion with the authors, it is clear that the issue comes from unreproducible data post-processing applied to the SPC data to compile results at the daily time scale._"

Of course, this unreproducible data processing is not satisfactory and we took a special care in this publication to respect the FAIR principles with all our data and provide all details in the Code and Data availability section to prevent such inconveniences.

**REFERENCES**

Amory, C., Kittel, C., Le Toumelin, L., Agosta, C., Delhasse, A., Favier, V., and Fettweis, X.: Performance of MAR (v3. 11) in simulating the drifting-snow climate and surface mass balance of Adélie Land, East Antarctica, Geoscientific Model Development, 14, 3487–3510,830 https://doi.org/0.5194/gmd-14-3487-2021, 2021

Cluzet, B., Lafaysse, M., Deschamps-Berger, C., Vernay, M., and Dumont, M.: Propagating information from snow observations with CrocO ensemble data assimilation system: a 10-years case study over a snow depth observation network., Cryosphere Discussions, https://doi.org/10.5194/tc-16-1281-2022, 2021.

Deschamps-Berger, C., Gascoin, S., Berthier, E., Deems, J., Gutmann, E., Dehecq, A., ... & Dumont, M. (2020). Snow depth mapping from stereo satellite imagery in mountainous terrain: evaluation using airborne laser-scanning data. _The Cryosphere_, _14_(9), 2925-2940

Essery, R., Li, L., and Pomeroy, J.: A distributed model of blowing snow over complex terrain, Hydrological processes, 13, 2423–2438, https://doi.org/10.1002/(SICI)1099-1085(199910)13:14/15%3C2423::AID-HYP853%3E3.0.CO;2-U, 1999.

Gallée, H., Guyomarc'h, G., and Brun, E (2001).: Impact of snow drift on the Antarctic ice sheet surface mass balance: possible sensitivity to snow-surface properties, Boundary-Layer Meteorology, 99, 1–19, https://doi.org/10.1023/A:1018776422809

Gerber, F., Sharma, V., & Lehning, M. (2023). CRYOWRF-a validation and the effect of blowing snow on the Antarctic SMB. *Authorea Preprints*.

Gilleland, E., Ahijevych, D., Brown, B. G., Casati, B., & Ebert, E. E. (2009). Intercomparison of spatial forecast verification methods. *Weather and forecasting*, *24*(5), 1416-1430.

Günther, D., Marke, T., Essery, R. et Strasser, U. (2019). Uncertainties in Snowpack Simulations—Assessing the Impact of Model Structure, Parameter Choice, and Forcing Data Error on Point-Scale Energy Balance Snow Model Performance. Water Resour. Res., 55(4):2779–2800, https://doi.org/10.1029/2018WR023403.

Guyomarc'h, G. and Mérindol, L.: Validation of an application for forecasting blowing snow, Annals of Glaciology, 26, 138–143, https://doi.org/10.3189/1998AoG26-1-138-143, 1998

Largeron, C., Dumont, M., Morin, S., Boone, A., Lafaysse, M., Metref, S., Cosme, E., Jonas, T., Winstral, A., and Margulis, S. A.: Toward snow cover estimation in mountainous areas using modern data assimilation methods: a review, Frontiers in Earth Science, 8, 325, https://doi.org/10.3389/feart.2020.00325, 2020.

 Lehning, M., Doorschot, J., and Bartelt, P.: A snowdrift index based on SNOWPACK model calculations, Annals of Glaciology, 31, 382–386,https://doi.org/10.3189/172756400781819770, 2000.

Liston, G. E. and Sturm, M.: A snow-transport model for complex terrain, Journal of Glaciology, 44, 498–516, https://doi.org/10.3189/S0022143000002021, 1998.

Liston, G. E., Haehnel, R. B., Sturm, M., Hiemstra, C. A., Berezovskaya, S., and Tabler, R. D.: Simulating complex snow distributions in windy environments using SnowTran-3D, Journal of Glaciology, 53, 241–256, https://doi.org/10.3189/172756507782202865, 2007.

Menard, C. B., Essery, R., Krinner, G., Arduini, G., Bartlett, P., Boone, A., ... & Yuan, H. (2021). Scientific and human errors in a snow model intercomparison. *Bulletin of the American Meteorological Society*, *102*(1), E61-E79.

Morin, S., Horton, S., Techel, F., Bavay, M., Coléou, C., Fierz, C., Gobiet, A., Hagenmuller, P., Lafaysse, M., Ližar, M., Mitterer, C., Monti, F., Müller, K., Olefs, M., Snook, J. S., van Herwijnen, A., and Vionnet, V.: Application of physical snowpack models in support of operational avalanche hazard forecasting: A status report on current implementations and prospects for the future, Cold Regions Science and Technology, 170, 102 910, https://doi.org/10.1016/j.coldregions.2019.102910, 2020b.

Raleigh, M. S., Lundquist, J. D. et Clark, M. P. (2015). Exploring the impact of forcing error characteristics on phy-
sically based snow simulations within a global sensitivity analysis framework. Hydrol. Earth Syst. Sci., 19(7):3153–3179. https://doi.org/10.5194/hess-19-3153-2015.

Schlögl, S., Marty, C., Bavay, M. et Lehning, M. (2016). Sensitivity of Alpine3D modeled snow cover to modifications in DEM resolution, station coverage and meteorological input quantities. Environmental Modelling & Software, 83:387–396. https://doi.org/10.1016/j.envsoft.2016.02.017

Schweizer, J., Jamieson, J. B., and Schneebeli, M.: Snow avalanche formation, Reviews of Geophysics, 41, https://doi.org/10.1029/2002rg000123, 2003.

Sharma, V., Gerber, F., and Lehning, M.: Introducing CRYOWRF v1. 0: Multiscale atmospheric flow simulations with advanced snow cover modelling, Geoscientific Model Development Discussions, pp. 1–46, https://doi.org/10.5194/gmd-2021-231, 2021

Viallon-Galinier, L., Hagenmuller, P., & Lafaysse, M.  Forcing and evaluating detailed snow cover models with stratigraphy observations. *Cold Regions Science and Technology*, *180*, 103163, 2020

Vionnet, V., Guyomarc'h, G., Bouvet, F. N., Martin, E., Durand, Y., Bellot, H., Bel, C., and Puglièse, P.: Occurrence of blowing snow events at an alpine site over a 10-year period: Observations and modelling, Advances in water resources, 55, 53–63, https://doi.org/10.1016/j.advwatres.2012.05.004, 2013.

Vionnet, V., Martin, E., Masson, V., Guyomarc'h, G., Naaim-Bouvet, F., Prokop, A., Durand, Y., and Lac, C.: Simulation of wind-induced snow transport and sublimation in alpine terrain using a fully coupled snowpack/atmosphere model, The Cryosphere, 8, 395–415, https://doi.org/10.5194/tc-8-395-2014, 2014.

Vionnet, V., Guyomarc'h, G., Lafaysse, M., Naaim-Bouvet, F., Giraud, G., and Deliot, Y.: Operational implementation and evaluation of a blowing snow scheme for avalanche hazard forecasting, Cold Regions Science and Technology, 147, 1–10, https://doi.org/10.1016/j.coldregions.2017.12.006, 2018.

Walter, B., Weigel, H., Wahl, S., and Löwe, H.: Wind tunnel experiments to quantify the effect of aeolian snow transport on the surface snow microstructure, The Cryosphere Discuss. [preprint], https://doi.org/10.5194/tc-2023-112, in review, 2023.

---

## Author Comment (AC2)

In the following, author comments are in black and reviewer comments are in **blue.** When we present text passages from the revised manuscript, they appear in "*italics*".

**The manuscript "SnowPappus v1.0, a blowing-snow model for large-scale applications of Crocus snow scheme" by Baron et al., presents a model development for the Crocus snow model to include drifting snow processes. Given the operational applications of Crocus, it potentially is an important step forward. This would warrant publication in a journal like GMD.**

We first would like to thank the reviewer for their constructive comments. In the following, we will give a detailed answer to each of them.

**However, having said that, I think that the major drawback of the current manuscript is that the goal (which is somewhat implicitly stated in the introduction) is not corroborated by the right validation data to determine if the model developments regarding the drifting snow module are actually an improvement. In other words, I interpret the goal of the model development to be to better capture the spatial distribution of snow (l.23-28). However, the actual goal stated by the authors in the Introduction is vague (l.51): "to carry out simulations at the scale of the French Alps." One would expect here to read something like: "to carry out simulations that improve the spatial distribution of snow depth at the scale of the French alps". The only validation data presented are the blowing snow measurements, which are point measurements. This kind of point validation data makes it hard to justify if the spatially explicit, 2D treatment of drifting snow is in fact useful. However, if the only goal is to represent drifting snow mass fluxes, it would be necessary to evaluate if a 2D/3D approach is really necessary, or if simply the 1D approach, calculating mass fluxes based on snow cover properties and wind speed is sufficient (i.e, applying Eq. 22) to reproduce that.**

**I also would like to stress here that I think that for operational applications, there should also be a demand by the operational users for any validation of model output that is going to be used in an operational product. How would an operational team judge simulated spatial patterns of snow depth when they cannot be certain how well the model reproduces those? Observed blowing snow fluxes at only three points in the domain hardly provide confidence that spatial patterns of snow deposition are in fact correctly reproduced.**

**Note that drifting snow also impacts the snow microstructure and density profiles by forming wind slabs. This did not seem to be the focus of the authors, but it would require snow pits to validate the results. In any case, the Introduction should discuss this snow microstructural aspect in more depth, I think.**

**So unfortunately, I think that this is a more serious flaw of the study that makes it hard to further judge the study for possible publication. If I were to recommend major revisions, I would need to see a path forward for how revisions, including new analysis or simulations, could better support the conclusions. But since to me it is not clear at this point what the goal of the model development is, it is very hard to judge what is needed and if it can be deemed**

**feasible. I think either the focus needs to be on the concentration profiles at 1D simulations, and compare those with observations. Or include observations of spatial patterns of snow depth to investigate to what extent the model reproduces those patterns. However, all these options require major redesign of the study and a big overhaul of the manuscript.**

The reviewer raises here major concerns about the goals of the article and its relevance in reaching these goals. In the following, we first clarify these goals and then discuss our choice of presenting point-scale flux evaluation rather than spatial patterns of snow accumulation or stratigraphies.

First of all, we would like to apologize for the lack of clarity in our goals in this model development, and we will clarify them in the revised manuscript. Our general goal is to develop a simulation system of the snowpack evolution at 250 m resolution covering the whole French Alps, based on the Crocus operational snow model and for various applications (avalanche forecasting, water resource monitoring for hydroelectricity, snow climatology, ...). To be able to represent a realistic spatial variability of snow properties, the implementation of a blowing snow module coupled to Crocus is necessary. The specific goal of this paper is to present the new blowing snow module dedicated to this specific application, including (1) an accurate model description with appropriate scientific justification considering existing literature, (2) a direct evaluation of blowing snow fluxes based on the available data and (3) an assessment of the numerical applicability of this scheme at our target resolution and spatial domain.

In this model description paper, we considered that the evaluation of blowing snow fluxes was the main topic to address as this is the most direct observation of the newly simulated processes. We agree with the reviewer that these evaluations do not allow to determine if the spatial patterns of snow properties simulated are improved by the blowing snow module. This limitation is explicitly mentioned L858-860 "Snow redistribution in 2D simulations has not been evaluated in this article, and will be the subject of a future study expected to provide complementary insights to the following discussions". Such evaluations are definitely necessary but they raise major methodological difficulties which justify a dedicated and separated paper.

First of all, in large scale simulations, several sources of errors are superposed, making it hard to determine if differences between model and observations come from a misrepresentation of wind-induced snow transport or from other sources of errors. In particular, precipitation forcing at high altitudes suffer from high uncertainty partly due to a largely unexplained spatial variability in NWP precipitation outputs, a severe lack of observations to constrain meteorological analysis systems, and various issues in radar precipitation measurements over complex terrains. It usually leads forcing errors to prevail in snow simulation systems that are not forced by local observation (Raleigh et al., 2015; Schlögl et al., 2016; Günther et al., 2019). Then, as point-scale observations are often not representative of the spatial scale of a simulation system (i.e. 250 m resolution in our case), evaluations have to rely on satellite observations which also involve complex retrieval methodologies and an appropriate consideration of associated uncertainties. Finally, methodological developments are required to compare simulated and observed snow maps as consistent spatial patterns may be simulated with slight localization inaccuracies, making traditional evaluation scores often unadapted (Gilleland et al., 2009).

Therefore, we also prepared a dedicated evaluation paper of the spatial distributions of snow height and Snow Melt Out Date against satellite stereo-imagery (Deschamps-Berger et al., 2020) and optical products (Gascoin et al., 2019). In this evaluation, the contributions of model and

forcing uncertainties in simulation errors are quantified and compared. As expected, the forcing uncertainties prevail. This paper (Haddjeri et al.) will be submitted in september 2023 to The Cryosphere and will be complementary to the model description and first evaluations provided in this GMD model description paper.

Moreover, we would like to stress that we follow a strategy similar to other published wind-induced snow transport models operating at similar or lower resolution. Due the same methodological challenges, the corresponding publications did neither describe evaluation of the simulated spatial patterns of snow depth or surface properties (Gallée et al., 2001; Amory et al. 2021; Sharma et al., 2021). These spatial evaluations, when performed, were sometimes conducted in a separate paper (Gerber et al., 2023). Smaller scale blowing snow models were often described along with a purely qualitative evaluation of snow depth patterns on areas ranging from hundreds of meters long transects to a few kilometre square test zones (Liston and Sturm (1998); Liston et al. (2007); Vionnet et al. (2014)), and sometimes without any spatial evaluation (Essery et al., 1999).

Conversely, we provide in the present paper quantitative evaluations of the blowing snow flux and transport occurrence simulated with SnowPappus against a 10-years long observation time series at Col du Lac Blanc. As stressed in our introduction, such evaluations are very unusual in the currently available literature (except in Amory et al., 2021 in Antarctica), so concerning this direct variable, the evaluations provided in our paper are more advanced than similar literature on blowing snow models. These analyses allow to evaluate and discuss the parameterization choices done to compute blowing snow occurrence and fluxes and to question the interest of microstructure-based parameterizations for which the added value had never been assessed. This is also an important added value of our paper.

We propose to clarify this goal and strategy in the introduction. Given the introduction will be merged with Sect. 2.1 (see below our response on manuscript length and organization), we give here the full paragraphs including this clarification. The part of the text which is the most dedicated to it appears in bold.

"*In the above-mentioned context of increasing resolution of snow modelling systems, the long-term project of CNRM aims at performing simulations with Crocus at the scale of the French Alps at 250m resolution in an operational purpose, associated with a data assimilation framework requiring ensemble runs of 50-100 members (Largeron et al., 2020; Cluzet et al., 2021). The 250 m resolution allows a trade-off between the need for precisely representing slopes and aspects, influencing mass and energy balance of the snowpack, and the expected computational cost. In this context, a numerically efficient representation of wind-induced snow transport that can be coupled to Crocus simulations is lacking while this is necessary to better account for its impact on avalanche forecasting over French mountains. Two blowing snow scheme coupled with Crocus exist yet : SYTRON (Vionnet et al., 2018) and Crocus-Meso-NH (Vionnet et al., 2014). However, both are unadapted to this geometry and resolution.*

*Thus, the goal of this paper is to describe and present first evaluations of a novel blowing snow scheme, SnowPappus, coupled to Crocus and able to be included in the above-mentioned large-scale simulation system. Point-scale evaluation of blowing snow flux will be presented to discuss the modelling choices*"

In addition, L858-860 in the second part of the discussion will be replaced by :

" *Snow redistribution in 2D simulations has not been evaluated in this article, due to several methodological challenges including dealing with the superposition of errors coming from the precipitation fields and finding relevant metrics. It will be the subject of a future study expected to provide complementary insights to the following discussions.*"

Finally, drifting snow highly impacts snow stratigraphy and this is one of the main reasons why spatialized applications of Crocus intending to take benefit from the simulated snow stratigraphies (e.g. avalanche hazard forecasting) require the implementation of a dedicated blowing snow module. We agree with the reviewer that this aspect should be discussed in the introduction, and this will be done in the revised manuscript.

However, in the general case, the evaluation of simulated snow profiles against observations is still highly challenging and appropriate evaluation methodologies are complex to set up (e.g. Viallon-Galinier et al., 2020). This caveat is identified as one of the main unresolved issue in numerical snow modelling (Morin et al., 2020, Ménard et al., 2021) and it can not be resolved in our paper. In the case of wind-blown areas, an additional major limitation is the unavailability of snow pits that would be necessary to evaluate these features. We pointed out L879-883 in the discussion the lack of quantitative information about the interaction between wind-induced snow transport and snow surface properties, which explains why we chose very simple representations of this process, considering the impossibility of any accurate evaluation. As we wrote, observation or experimentation campaigns such as the very recent work of Walter et al., 2023, with measures of snow properties in wind-blown areas would be of primary interest to enhance the representation of these processes in a model.

In the revised manuscript, we propose to discuss this briefly in the introduction. We propose to include this paragraph :

"*A major interest in coupling Crocus with a blowing snow scheme is its detailed representation of snow stratigraphy and microstructure as it may be an opportunity for the simulation of snow transport occurrence (Guyomarc'h and Mérindol, 1998; Lehning et al., 2000). Therefore, we test the added value of microstructure-based parameterizations of snow transport occurrence in the evaluation section. Moreover, it allows Crocus to be used as a tool for avalanche forecasting (Morin et al., 2020). Given that wind slabs formed by wind-induced snow deposition are one of the main causes of avalanche triggering (Schweizer et al., 2003), a blowing snow scheme coupled with Crocus could become a powerful tool for avalanche forecasting, even if evaluation of the simulated stratigraphy is out of the scope of this study.*"

Note that clarifications asked about the interest of using snow microstructure in a blowing snow model are also included in this paragraph (see our response dedicated to it below)

**Another major concern is the length of the paper, which I think is mainly a result of insufficient organization and logic. On occasion there is too much detail given, and some discussion is too spread out. For example, Section 2.2.1 treats "Theoretical background", but**

**section 2.3.2 also reads like theoretical background. So while reading, the manuscript is jumping back and forth between theoretical considerations, and implementation details, which makes it somewhat cumbersome to read and follow. But the manuscript would need to be shortened massively and bring its length more in line with the amount of unique content and validation data. Otherwise, a lot of detail is provided which is not helpful to interpret the results. For example, the discussion on sublimation stand all by itself. It is not clear at all how it impacts the simulations.**

We understand the major concerns of both reviewers about the length and lack of clarity of the paper. In order to address this issue, we propose to extensively modify the structure of the paper to improve its readability. We will split Section 2 (Methodological choices) into two separate sections. The first one will be dedicated to a literature review largely reduced in length by focusing on the topics for which an added value is provided in SnowPappus compared with existing works. It includes discussions about the representation of saltation flux and lower boundary condition for suspension flux and terminal fall speed parameterizations. This will allow to make the logics of the manuscript clearer. For processes that are simply implemented in SnowPappus following existing models or previous literature, the literature overview will be considerably shortened, or even suppressed if it is not useful for the remaining part of the article, such as the discussion on sublimation (L449-452). Then, a second section will be dedicated to a concise model description without any theoretical interruptions between the description of the different implementations. We believe this organization will help the reader to understand more quickly our modelling choices.

In addition, several parts of the manuscript will be strongly shortened, primarily the description of numerical performance, various repetitions and unnecessary information or discussion will be withdrawn and a more concise writing style will be adopted when possible. Besides, the modification of the paper structure, which splits the section "Methodological choices" into 2 parts do not result in additional length, as the number of subsubsections does not changes a lot and repetitions will be avoided. Overall, the planned changes will allow to suppress Fig. 15 and move Fig. 14 in the appendix and to reduce overall text length by 20% (without taking appendix into account).

In the following, we will first list the text passages that will be strongly shortened, or withdrawn, in their order of appearance and identified by their section number and if necessary line number in the preprint. Then, we present a revised text outline to apply the described changes. New (sub)sections are written with their titles underlined, and there contains in plain text. Parts of the outline in which the organization remain identical with the preprint's outline are marked as UNCHANGED, although some length reductions have been also applied in these parts.

**shortened text passages (with paragraph and line number from the initial manuscript):**
   2.1 Target, opportunities and constraints
   2.2.2, wind profile
   2.3.2 Blowing snow trajectories and transport modes
   2.3.4 L257-263 discussion on the influence of particle size distribution on the suspension transport
   2.3.3 L207-241 - Different types of transport models
   2.3.7 L334-358 equations of  simple saltation parameterizations P90 and S04
   2.3.8  Influence of fetch distance
   2.5 L502-508 Influence if snow transport on snow surface properties (state of the art)

3.2 L 560-570 Wind downscaling description
3.5 L 623-634 Methods for blowing snow occurrence measurements
4.6 L734-739, domain decomposition for parallel computing
5.1 L799-805 Discussion on the outlier in blowing snow fluxes evaluation
withdrawn text passages :
2.4 L449-452, Sublimation
4.5 L712-717, Evaluation of blowing snow fluxes at Col du Lac Blanc, discussion on the outlier
(repeated in the discussion)

**new outline in the revised manuscript :**
1. Introduction
   General introduction followed by the target, opportunities and constraints of the development of SnowPappus (corresponds to Sect. 2.1 in the preprint)

2. Blowing snow flux computation : state of the art
2.1 Blowing snow occurrence
  Useful theoretical background for blowing snow occurrence detection (Sect. 2.2.1 in the preprint)
2.2 Horizontal blowing snow fluxes
   2.2.1 Notations and geometric considerations
   2.2.2 Blowing snow particle trajectories and transport modes
    State of the art on the trajectories of blowing snow particles and transport modes, focused on saltation and suspension (corresponds to Sect. 2.3.2 in the preprint)
   2.2.3 Suspension transport modelling
    -Existing types of suspension transport models (part of Sect. 2.3.3 in the preprint)
    -Literature review on the effective terminal fall speed (mainly informations in Sect. 2.3.5 of the preprint)
   2.2.4 Transition between saltation and suspension
    - The way it is treated in other models
    - State of the art on this transition zone
    - Problems of definition of the lower boundary condition for suspension transport (corresponds mainly to Sect. 2.3.6 of the preprint)
   2.2.5 Simple saltation models
    -Short description of Sorensen et al., 2004 (S04) and Pomeroy et al., 1990 (P90) saltation parameterizations
    -Discrepancies between S04 and P90 and discussion of the possible causes (informations in Sect. 2.3.7 of the preprint)

3. Model description
3.0 Crocus description
  Very short description of Crocus snow model
3.1 Blowing snow occurrence
  Equations of the different options implemented in SnowPappus for threshold wind speed for transport (Sect. 2.2.3 in the preprint)
3.2 Horizontal blowing snow flux
   3.2.1 Suspension transport
    - Reasons for the choice of the model type
    - Logarithmic wind speed profile (Sect. 2.2.2 in the preprint)

- Equations for suspension transport in SnowPappus (corresponds to Sect. 2.3.4 in the preprint)
- Parameterization of terminal fall speed used in SnowPappus (informations in Sect. 2.3.5 of the preprint)
- Maximum height of suspension transport as a function of fetch distance (part of Sect. 2.3.8 of the preprint)

3.2.2 Saltation transport and transition with suspension
- Description of the two options (using S04 and P90) of flux computation in the saltation zone and the transition with suspension in SnowPappus (included in Sect. 2.3.7 of the preprint).
- Influence of fetch distance on saltation transport in SnowPappus (included in Sect. 2.3.8 of the preprint, with Fig. 3 appearing)

3.3 Sublimation
Sublimation options in SnowPappus

3.4 Mass balance
Mass balance implementation in SnowPappus

3.5 Influence of snow transport and deposition on snow surface properties
- Properties of deposited snow
- Blowing snow induced snow metamorphism

3.6 Implementation in SURFEX
How SnowPappus is included in SURFEX code (mainly Sect. 2.7 of the preprint, but includes the description of domain decomposition for parallel computing)

4. Evaluation : methods
UNCHANGED (with length reductions)

5. Results
mainly UNCHANGED but addition of 1 paragraph (see below) and Fig. 14 moved in appendix, as partly redundant with Fig. 13 .

5.1 Comparison of saltation parameterizations
Comparison of blowing snow fluxes obtained with S04 and P90 implementations in SnowPappus and implications for the comparison of S04 and P90 (corresponds to the end of Sect. 2.3.7 in the preprint).

6. Discussion
UNCHANGED (with length reductions)

7. Conclusion
UNCHANGED

**l.76-77: "The ability of Crocus to distinguish different snow types at the surface may be an opportunity for the simulation of snow transport (Guyomarc'h and Mérindol, 1998; Lehning et al., 2000)." is vague. Do authors aim to validate this in their study or not?**

In this sentence, we point out that different studies used parameterizations based on snow microstructure for blowing snow occurrence. However, in the particular case of Crocus, added value of microstructure-based parameterizations compared with simpler ones was not demonstrated in earlier studies (Guyomarc'h and Merindol, 1998; Vionnet et al., 2013; Vionnet et al., 2018), hence our use of *may.* Therefore, in our paper we test this hypothesis in Sect. 4.3 by comparing Guyomarc'h and Merindol (1998) parameterization with a simpler one. We propose to clarify this point by changing L76-77 in

"*A major interest in coupling Crocus with a blowing snow scheme is its detailed representation of snow stratigraphy and microstructure as it may be an opportunity for the simulation of snow transport occurrence (Guyomarc'h and Mérindol, 1998; Lehning et al., 2000). Therefore, we test the added value of microstructure-based parameterizations of snow transport occurrence in the evaluation section.*"

Note that, due to the reorganisation proposed above, as the whole Sect. 2.1 this will be part of the introduction.

**On a final note, given that Vincent Vionnet is listed as co-author, it is actually very strange to read:**
**L.790-791: "precise temporal window within the year of Vionnet et al. (2018) which was not given in their article and may differ from ours" and l.792: "We could retrieve the original simulation outputs of Vionnet et al. (2018)". It shouldn't have been so problematic to resolve these issues, since Vincent Vionnet is co-author. In fact, given his co-authorship, a much more solid discussion of the SnowPappus results with his earlier work is expected at this point.**

We would like to apologize for this unclear formulation that did not correctly reflect our clear understanding of this issue. In fact we were able to retrieve the original simulation outputs from Vionnet et al. (2018) and found there was not any issue about it. As said L793 we "applied our evaluation process to these data[ ...] We obtained results very close to our own Sytron run with the original data".  Regarding the dates, we were able to obtain the information on the precise temporal window used in Vionnet et al. (2018), which is from 01/11 to 15/04. Therefore, their temporal window is larger than ours, so it is not possible that the change in the temporal window by itself would cause a perfect detection in their case and not in ours. As a consequence, we are sure the reproducibility issue comes from an unreproducible data post-processing, that was applied when compiling the results at the daily time scale in Vionnet et al. (2018). This step was not applied in our evaluation dataset. To clarify this and taking into account the reviewer's call for concision, we propose to replace L789-794 by :

"*We were able to retrieve the original simulation outputs of Vionnet et al. (2018) and applied our evaluation process to these data (see code availability), obtaining results very close from ours . Thus, after discussion with the authors, it is clear that the issue comes  from unreproducible data post-processing applied to the SPC data to compile results at the daily time scale.*"

Of course, this unreproducible data processing is not satisfactory and we took a special care in this publication to respect the FAIR principles with all our data and provide all details in the Code and Data availability section to prevent such inconveniences.

**REFERENCES**

Amory, C., Kittel, C., Le Toumelin, L., Agosta, C., Delhasse, A., Favier, V., and Fettweis, X.: Performance of MAR (v3. 11) in simulating the drifting-snow climate and surface mass balance of Adélie Land, East Antarctica, Geoscientific Model Development, 14, 3487–3510,830 https://doi.org/0.5194/gmd-14-3487-2021, 2021

Cluzet, B., Lafaysse, M., Deschamps-Berger, C., Vernay, M., and Dumont, M.: Propagating information from snow observations with CrocO ensemble data assimilation system: a 10-years case study over a snow depth observation network., Cryosphere Discussions, https://doi.org/10.5194/tc-16-1281-2022, 2021.

Deschamps-Berger, C., Gascoin, S., Berthier, E., Deems, J., Gutmann, E., Dehecq, A., ... & Dumont, M. (2020). Snow depth mapping from stereo satellite imagery in mountainous terrain: evaluation using airborne laser-scanning data. *The Cryosphere, 14*(9), 2925-2940

Essery, R., Li, L., and Pomeroy, J.: A distributed model of blowing snow over complex terrain, Hydrological processes, 13, 2423–2438, https://doi.org/10.1002/(SICI)1099-1085(199910)13:14/15%3C2423::AID-HYP853%3E3.0.CO;2-U, 1999.

Gallée, H., Guyomarc'h, G., and Brun, E (2001).: Impact of snow drift on the Antarctic ice sheet surface mass balance: possible sensitivity to snow-surface properties, Boundary-Layer Meteorology, 99, 1–19, https://doi.org/10.1023/A:1018776422809
Gerber, F., Sharma, V., & Lehning, M. (2023). CRYOWRF-a validation and the effect of blowing snow on the Antarctic SMB. *Authorea Preprints*.

Gilleland, E., Ahijevych, D., Brown, B. G., Casati, B., & Ebert, E. E. (2009). Intercomparison of spatial forecast verification methods. *Weather and forecasting, 24*(5), 1416-1430.

Günther, D., Marke, T., Essery, R. et Strasser, U. (2019). Uncertainties in Snowpack Simulations—Assessing the Impact of Model Structure, Parameter Choice, and Forcing Data Error on Point-Scale Energy Balance Snow Model Performance. Water Resour. Res., 55(4):2779–2800, https://doi.org/10.1029/2018WR023403.

Guyomarc'h, G. and Mérindol, L.: Validation of an application for forecasting blowing snow, Annals of Glaciology, 26, 138–143, https://doi.org/10.3189/1998AoG26-1-138-143, 1998

Largeron, C., Dumont, M., Morin, S., Boone, A., Lafaysse, M., Metref, S., Cosme, E., Jonas, T., Winstral, A., and Margulis, S. A.: Toward snow cover estimation in mountainous areas using

modern data assimilation methods: a review, Frontiers in Earth Science, 8, 325, https://doi.org/10.3389/feart.2020.00325, 2020.

Lehning, M., Doorschot, J., and Bartelt, P.: A snowdrift index based on SNOWPACK model calculations, Annals of Glaciology, 31, 382–386,https://doi.org/10.3189/172756400781819770, 2000.

Liston, G. E. and Sturm, M.: A snow-transport model for complex terrain, Journal of Glaciology, 44, 498–516, https://doi.org/10.3189/S0022143000002021, 1998.

Liston, G. E., Haehnel, R. B., Sturm, M., Hiemstra, C. A., Berezovskaya, S., and Tabler, R. D.: Simulating complex snow distributions in windy environments using SnowTran-3D, Journal of Glaciology, 53, 241–256, https://doi.org/10.3189/172756507782202865, 2007.

Menard, C. B., Essery, R., Krinner, G., Arduini, G., Bartlett, P., Boone, A., ... & Yuan, H. (2021). Scientific and human errors in a snow model intercomparison. *Bulletin of the American Meteorological Society, 102*(1), E61-E79.

Morin, S., Horton, S., Techel, F., Bavay, M., Coléou, C., Fierz, C., Gobiet, A., Hagenmuller, P., Lafaysse, M., Ližar, M., Mitterer, C., Monti, F., Müller, K., Olefs, M., Snook, J. S., van Herwijnen, A., and Vionnet, V.: Application of physical snowpack models in support of operational avalanche hazard forecasting: A status report on current implementations and prospects for the future, Cold Regions Science and Technology, 170, 102 910, https://doi.org/10.1016/j.coldregions.2019.102910, 2020b.

Raleigh, M. S., Lundquist, J. D. et Clark, M. P. (2015). Exploring the impact of forcing error characteristics on phy-
sically based snow simulations within a global sensitivity analysis framework. Hydrol. Earth Syst. Sci., 19(7):3153–3179. https://doi.org/10.5194/hess-19-3153-2015.

Schlögl, S., Marty, C., Bavay, M. et Lehning, M. (2016). Sensitivity of Alpine3D modeled snow cover to modifications in DEM resolution, station coverage and meteorological input quantities. Environmental Modelling & Software, 83:387–396. https://doi.org/10.1016/j.envsoft.2016.02.017

Schweizer, J., Jamieson, J. B., and Schneebeli, M.: Snow avalanche formation, Reviews of Geophysics, 41, https://doi.org/10.1029/2002rg000123, 2003.

Sharma, V., Gerber, F., and Lehning, M.: Introducing CRYOWRF v1. 0: Multiscale atmospheric flow simulations with advanced snow cover modelling, Geoscientific Model Development Discussions, pp. 1–46, https://doi.org/10.5194/gmd-2021-231, 2021

Viallon-Galinier, L., Hagenmuller, P., & Lafaysse, M.  Forcing and evaluating detailed snow cover models with stratigraphy observations. *Cold Regions Science and Technology*, *180*, 103163, 2020

Vionnet, V., Guyomarc'h, G., Bouvet, F. N., Martin, E., Durand, Y., Bellot, H., Bel, C., and Puglièse, P.: Occurrence of blowing snow events at an alpine site over a 10-year period: Observations and modelling, Advances in water resources, 55, 53–63, https://doi.org/10.1016/j.advwatres.2012.05.004, 2013.

Vionnet, V., Martin, E., Masson, V., Guyomarc'h, G., Naaim-Bouvet, F., Prokop, A., Durand, Y., and Lac, C.: Simulation of wind-induced snow transport and sublimation in alpine terrain using a fully coupled snowpack/atmosphere model, The Cryosphere, 8, 395–415, https://doi.org/10.5194/tc-8-395-2014, 2014.

Vionnet, V., Guyomarc'h, G., Lafaysse, M., Naaim-Bouvet, F., Giraud, G., and Deliot, Y.: Operational implementation and evaluation of a blowing snow scheme for avalanche hazard forecasting, Cold Regions Science and Technology, 147, 1–10, https://doi.org/10.1016/j.coldregions.2017.12.006, 2018.

Walter, B., Weigel, H., Wahl, S., and Löwe, H.: Wind tunnel experiments to quantify the effect of aeolian snow transport on the surface snow microstructure, The Cryosphere Discuss. [preprint], https://doi.org/10.5194/tc-2023-112, in review, 2023.

---

## Author Response (AR2)

We thank both referees and the editor for their constructive comments which will improve our manuscript. We regret that Referee #2 does not fully appreciate the added value of our work but we believe this is mainly because they would have expected a traditional scientific paper including both model description and extensive evaluations in the same paper, whereas our strategy was (1) to use the opportunity offered by GMD model description papers to provide a more accurate model description than in publications in other journals, and (2) to extend model evaluations in future papers. We understand from the editor recommendations that a compromise has to be found. Therefore, we tried to do our best to account for Referee #2 last comments and include new evaluations of our model outputs. Thus, we chose to include in the revised paper a 2-dimensional evaluation of simulated snow depth against a satellite-derived snow depth map. This preliminary evaluation is complemented with a paper recently submitted to The Cryosphere and providing more robustness in the associated conclusions (Haddjeri et al., 2023). Taking into account the editor's recommendation for further conciseness in the manuscript, we chose to move our study about of the numerical performance of SnowPappus to the Appendix. Although this is an important component of the applicability of the model at large scale, the consistency of the whole paper is not affected by this choice. We also made other small adjustments to avoid increasing the length of the paper despite the new material requested by Referee #2. We hope that the editor and referees would find our manuscript now fully fills the requirements of GMD, and that the publication process can follow its way so that ongoing works based on SnowPappus will have the possibility to be published with reference to a high quality model description paper.

Our detailed response to both referees reports are given below. Author comments are in black and reviewer comments are in **violet**. When we present text passages from the new revised manuscript, they appear in "*italics*". When we refer to a Sect., Fig. or line number, it refers to the version of the manuscript we will resubmit along with this response (the Third version). Second version of the manuscript will be referred as "the second version")

**Report #1 (Anonymous referee #2)**

**The authors presented a thoroughly revised manuscript. However, the authors decided not to modify the manuscript considering one of my main criticisms. They have decided to maintain the full storyline introducing a 2D drifting snow model framework, while still refraining from providing any validation for that part. I cannot find good arguments to do that, because what is the value of the manuscript for the broader scientific community in that case? Even when comparing with field data is difficult, still some kind of validation would be required in my opinion.**

As explained in our previous response, the 2D evaluations of Crocus-SnowPappus were intended to be presented in a separate paper due to the need for detailed geospatial analyses of the results and to explore the robustness of the analyses to different precipitation inputs. As this additional paper (Haddjeri et al., 2023) would have met the reviewer's expectations, we initially preferred not to include the 2D evaluations in this GMD article to favour in-depth presentation of 2D evaluations. However, considering the new arguments provided by the reviewer and editor, we decided to include a preliminary 2D evaluation of SnowPappus in this study. We also refer to Haddjeri et al., 2023 in the revised version of the manuscript to extend the scope of the conclusions. We also took into account the reviewer's comment about our lack of assessment of sublimation impact on simulations (see below).

Therefore, in the revised manuscript, the part of the article relying on 2D simulation now includes (i) a sensitivity analysis comparing simulated snow depth at the end of accumulation

season with three simulation set-ups (CTRL: no transport, no sublimation, TRANS: transport, no sublimation, TRANS+SUBL: transport, no sublimation) and (ii) a comparison of simulated snow depth with CTRL and TRANS simulations with observed snow depth obtained by state-of-the-art stereo-imagery from Pléiades satellites (Deschamps-Berger et al. 2020), which covers approximately 150 km2 in our test zone. The spatial correlations between observed and simulated snow depth and snow depth distributions above 2700 m are also compared.

The main new results (Fig. 12 and 13) are (i) blowing snow sublimation has much less impact on simulations than blowing snow transport (ii) Wind-induced snow transport enhances snow depth variability at high elevation, making simulated snow depth distribution closer to observations (iii) The spatial correlation between observed and modelled snow depth is significantly improved, although a large part of the observed variability remains unexplained and SnowPappus may overestimate erosion/deposition near high alpine ridges. These results demonstrate a significant added value of blowing snow simulation with SnowPappus. To our knowledge, it is the first study to demonstrate quantitatively an added value of a wind-induced snow transport model on snow spatial patterns for a complete winter season at 250 m resolution and in complex terrain. Two studies had however been conducted at 30-50 m resolution (Bernhardt et al., 2012; Vionnet et al., 2021).

However these results raise many additional questions, including the likely superposition of precipitation patterns errors and need to be strengthened by the use of more images. As explained above, these questions are addressed in a separate paper (Haddjeri et al., 2023) where a detailed sensitivity analysis of spatialized snow simulations to precipitation forcing, blowing snow representation, and model resolution is provided. This study shows that both components highly interact in any evaluation of 2D simulations and that results ignoring these uncertainties should be considered with caution. However, the added value of SnowPappus to simulate the spatial variance of snow depth and snow melt out date at high elevations and around crests is confirmed by Haddjeri et al., with more satellite observations than the ones used in this paper.

In order to include this new work in the manuscript, Sect. 4 (Methods), 5(Results) and 6(Discussion) are re-organized to describe the new methods, results and discussion associated with these 2D evaluations. Besides, to avoid making the article longer than it already was, the description of numerical performance (Sect. 5.7 in the previous version) is moved in appendix, and various small unnecessary text passages are removed, including Sect. 5.3 in the second version, which gives an illustration of 2D simulation output (Sect. 5.3 in the second version). The new outline is given below, with the included modifications highlighted in **bold**:

4. Evaluation : Methods
 [UNCHANGED START OF SECT. 4]
 4.3 Evaluation data
 Description of point-scale flux measurements
 **Description of Pleiades satellite snow depth maps**
 4.4 Point-scale evaluations
   4.4.1 Local simulations set-up
   4.4.2 Evaluation of blowing snow occurrence
   4.4.3 Evaluation of blowing snow fluxes
 4.5 2D evaluations
 **2D simulation set-ups**
 **methods of 2D evaluations**

5. Results
[UNCHANGED START OF SECT. 5]

**A suggestion I thought of when reading the revised manuscript is to compare the simulated snow cover with simulations using another model, such as SnowModel/SnowTran-3D, or SnowDrift3D. Or the SYTRON or Crocus-Meso-NH (mentioned in L56) schemes that are apparently available in Crocus. Then at least readers would get some understanding of how well the model performs compared to other models.**

Performing a model intercomparison would be beyond the scope of this paper which already contains additional 2D evaluations. It is important to mention that SYTRON and Crocus-Méso-NH could not be applied for these comparisons because (1) by design SYTRON can not run on 2D domains  (Vionnet et al. 2018), (2) Méso-NH can not be run over a full snow season due to its very high numerical cost. Applying SnowModel/SnowTran-3D would be possible but this comparison would raise many other questions due to the major differences between the snow schemes themselves, again beyond the scope of this publication.

**The problem is now as a reader, I just have no idea if this is a useful model framework. It is also important in this context that most, if not all parameterizations and numerical schemes were implemented from other studies. Obviously there is nothing wrong with learning from, and building upon existing literature, but it means that the only trust I can have in SnowPappus comes from the fact that it is so heavily based in existing literature. But that means that the real value right now is in these other studies, not this particular one.**

We would like to highlight again that our work presents several new elements compared to previous works. In terms of threshold wind speed for transport, we propose and evaluate a modification of the threshold wind speed formulation developed by Vionnet et al. 2013. In terms of saltation flux, we use the Pomeroy et al. 1990 and Sorensen 2004 formulations only after a precise analysis of the discrepancy between both formulation, whereas previous studies only used one or the other without further justification (Liston et al. 1998, Gallée et al. 2001, Vionnet et al. 2014) or revealed this

discrepancy without proposing detailed explanation for it (Doorschot et al. 2002, Melo et al. 2021). Finally, to our knowledge we are the first snow transport model to include a dependence of the terminal fall speed of suspended particle on snow microstructure. Although this work is based on already published experimental studies, we proposed a new parameterization to fill in the gaps in knowledge and calibrated it to new observational data. Furthermore, We show the strong impact it has on the suspension flux, showing that neglecting this dependency could have important effects on the performance of other models. We thus strongly believe that there is a scientific value in this study, which is further strengthened by the inclusion of 2-dimensional evaluations in the new version of the manuscript.

**Furthermore, I also mentioned that the option for drifting snow sublimation is introduced, but it's not clear at all how this impacts the simulation. The authors don't seem to have done anything with my comment.**

We apologize if we did not initially understand the reviewer's request. The available data do not allow providing evaluations of sublimation fluxes but we can indeed show the impact of this parameterization between 2 simulations. As mentioned above when describing new 2-dimensional evaluations of SnowPappus, we now present the difference between snow depth simulated with TRANS and TRANS+SUBL. This result is shown in Fig. 12b.

**I would like to briefly provide feedback on three arguments I could distill to not provide further validation:**

**1) Methodological challenges obtaining and comparing snow depth data.**
**As I pointed out, it is also possible to compare with other existing models. Fig. 8 shows up to and over 4m of depositions at the lee-side of ridges. Is that realistic? When the resolution of Fig. 8 is 250x250m, that means that there is a lot of additional accumulation in the lee side. I'm not convinced that that is realistic. Locally behind ridges, corniches can form, or maybe some small patches that fill in with high accumulation. But I'm not sure that it is realistic that on a scale of 250x250m, that there is so much additional accumulation. Similarly, up to 4m erosion, or even more, is simulated in certain areas, which also comes across as excessive over such large grid cells. For example, Fig. 9c in Mott et al., 2010 (doi: 10.5194/tc-4-545-2010) shows that accumulations up to 4m only occur on scales much smaller than 250x250m.**

As explained above, comparison with existing models is not so obvious, not possible with all models and beyond the scope of this paper.
Then, the reviewer seems to have misinterpreted our results. In fact,  Fig. 8a indicates maximum erosion/deposition of 400 kg/m2 of snow over one year of simulation. At common snow densities, this would correspond to  snow depth of 1 to 3 m, but certainly not 4 m. The orders of magnitude obtained in Mott el al. 2010 are not comparable with ours because in our case, the magnitude of erosion is obtained after the whole winter while the study of Mott et al. is performed on a much shorter time period (a single blowing snow event).  A more comparable study would be the one of Vionnet et al. 2021 with CHM-PBSM3D where wind-induced snow transport effect can be of more than 2 meters over areas of several hundreds of meters, which is comparable to our results. In the new discussion section discussing the added value of SnowPappus in 2D simulation, we will mention this study by stating:

*"Two-dimensional simulations on the Grandes Rousses test zone (see Sect. ...) showed that activating snow transport has a significant influence on snow height spatial distribution at high elevation at the end of accumulation season, reaching up to 2-3 m of erosion/deposition near high alpine crests. Comparable snow height differences where obtained with PBSM-3D (Vionnet et al, 2021)."*

**2) From the author's response: "Due the same methodological challenges, the corresponding publications did neither describe evaluation of the simulated spatial patterns of snow depth or surface properties (Gallée et al., 2001; Amory et al. 2021; Sharma et al., 2021)."**
**Regarding Amory et al. (2021), they in fact do show an evaluation of surface mass balance with the drifting snow enabled MAR (see Fig. 10 in that paper). Sharma et al. (2021) discusses the differences between the default surface scheme NoahMP and the newly introduced scheme SNOWPACK. Sharma et al. (2021) also additionally shows for example the differences between simulations with and without drifting snow sublimation. So, in my opinion, these examples actually provide some inspiration of how credibility can be given to the model development in the manuscript.**

The comparisons between CryoWRF and NoahMP in Sharma et al. 2023 only include evaluations of atmospheric variables such as atmospheric humidity or temperature, which are not direct transport evaluations, on a dozen of point stations. This cannot be seen as an 'evaluation of the simulated spatial patterns of snow depth or surface properties'. In fact, Sharma et al. 2023 does not include any evaluation of wind-induced snow transport, and has recently been accepted in GMD. We believe that this paper do provide insights for the community despite the lack of snow transport evaluations.

However, we did miss a spatial evaluation of snow mass balance conducted on a transect in Antarctica by Amory et al. 2021 and thank the reviewer for pointing this out. This evaluation is carried out on one transect with regular measurements of surface mass balance. Nevertheless, this brief evaluation of a snow transport-related 1D spatial pattern does not dismiss our arguments that a broader study is needed to properly address the issues of spatial evaluation of blowing snow modelling (robustness of conclusions with meteorological forcing, detailed spatial analysis with topography, etc.). This is the case thanks to our recently submitted paper (Haddjeri et al., 2023). A first overview of the results is now presented in the current manuscript and appropriately discussed in light of the most extensive evaluations of Haddjeri et al.

**3) The uniqueness of the drifting snow mass flux observations as validation data.**
**Here I would like to recall that I wrote in my review: "However, if the only goal is to represent drifting snow mass fluxes, it would be necessary to evaluate if a 2D/3D approach is really necessary, or if simply the 1D approach, calculating mass fluxes based on snow cover properties and wind speed is sufficient (i.e, applying Eq. 22) to reproduce that." Later on I wrote: "I think either the focus needs to be on the concentration profiles at 1D simulations, and compare those with observations."**
**I cannot find a response to that point in the response document. It's still not clear to me if a 2D model is required to reproduce the observed mass fluxes, or if a 1D approach would already yield satisfactory results.**

We believe that the goals of our modelling framework are now clearly explained in the introduction of our manuscript. Obviously the intent of SnowPappus is to be applied on 2D domains and to be able to simulate erosion / accumulation which is not possible with 1D approaches. However, it is still useful to check whether 2D models are able to simulate realistic drifting snow mass fluxes at the local scale. Indeed, this is a more direct evaluation than 2D erosion / accumulation patterns which cannot be easily disentangled from other processes explaining the observed variability in snow depth. As mentioned above, we decided to include a 2D evaluation of the model, but we believe that the improvements shown in the 2D simulations are highly strengthened by the confidence in the simulations of blowing snow fluxes demonstrated in our paper (i.e. improvements obtained for a good reason because the realism of the physical process has been checked).

**Report #2 (Anonymous referee #1)**

**I think that this revision is adequate on the large review questions, but I still have many minor comments.**

We thank Referee 1 for the improvements they have noticed on our revised manuscript and for the last comments which keep improving our manuscript.

**80 (and throughout)**
**units should not be in italics**

fixed

**140 (and throughout)**
**Use exponents rather than vertical inline equation.**

In order to make it easier to read, the equations written in the text L140 and 141 were rewritten as an equation separated from the text (Eq. 3).

**141**
**friction velocity not yet introduced**

fixed

**234**
**If this were "the main novelty introduced in SnowPappus", it would not merit publication**

We apologize for this incorrect formulation which was only referring to the novelties introduced in the threshold wind speed computation. We will replace "the main novelty introduced in SnowPappus" by "*a novelty introduced in SnowPappus*".

**253**
**value of z0 stated twice**

fixed

**264**
**use ln for natural logarithm**

fixed

**312**
**T is Ta elsewhere, u_wind is U**

$T_a$ was replaced by T and $u_{wind}$ by U where it appeared

**316**
**5 m stated twice**

fixed

**347**
**GMD style is vectors in bold italics**

fixed

**395**
**14443 250 m grid cells do not make 3200 km^2**

fixed (changed towards 900 km²).

**463**
**What is "was calibrated provide" intended to mean?**

We corrected this expression by "was calibrated to provide"

**468**
**How is d_m = 0 possible?**

In fact in this case the factor F defined L278 is equal to 1, although the expression we gave is singular. We replaced the F expression L278 by:

$$F = [\max(1, \frac{d_m}{d})]^{-1}$$

It is equivalent in the case d_m>0 and works for d_m =0 (given we define F only when d>0)

**494**
**Equation 26 could be replaced by reference to equation 13**

fixed

**Figure 7 caption**
**.144 should be superscript**

fixed

**Figure 11c does not show anything.**

We understand the reviewer emphasizes that simulated and observed wind speeds do not exhibit a good agreement at Chambon station (FCMB) compared to the other stations. We believe this result must be shown as the difficulty to simulate realistic small scale wind speed largely explains the difficulty to simulate snow transport as discussed in Section 6.2 of the revised manuscript. To account for this comment we rephrased as follows the description of this result:

*"It suggests that the accuracy of the downscaled wind speed and/or the 250 m spatial resolution of the simulation are the main causes of the skill deterioration, as confirmed by the significant discrepancies between observed and simulated wind speeds at the three stations (Fig. 11), with a variable skill between Col du Lac Blanc (R²=0,71, RMSE=3,3 m/s), Huez (R²=0,49, RMSE=2,5 m/s and Chambon(R²=0,42, RMSE=3,0m/s) stations and a significant underestimation of the highest wind speeds at all sites."*

In addition, we noticed a small mistake in Fig. 11b and c of the second version: wind speeds were not evaluated on the same time period as the blowing snow occurrence evaluation (from 01/12/2018

to 04/01/2019). We corrected this issue, which explains the small difference between the old and new figures. However this correction does not affect our conclusions.

**Figure 12 needs space between a and b.**

fixed

**571**
**u\* is not wind velocity**

fixed

**Figure A2**
**Snow3l not explained**

We apologize for this missing definition. 'Snow3l' was changed in 'full snow routine' which is defined in main text

**Figure A3**
**Shading for times with no valid observations is not explained**

The legend of Fig. A4 (Fig. A3 in the second version) was completed to explain the shading.

**REFERENCES**

Amory, C., Kittel, C., Le Toumelin, L., Agosta, C., Delhasse, A., Favier, V., and Fettweis, X.: Performance of MAR (v3.11) in simulating the drifting-snow climate and surface mass balance of Adélie Land, East Antarctica, Geoscientific Model Development, 14, 3487–3510, https://doi.org/10.5194/gmd-14-3487-2021, 2021.

Bernhardt, M., Schulz, K., Liston, G. E., & Zängl, G. The influence of lateral snow redistribution processes on snow melt and sublimation in alpine regions. *Journal of Hydrology*, *424*, 196-206, https://doi.org/10.1016/j.jhydrol.2012.01.001, 2012.

Deschamps-Berger, C., Gascoin, S., Berthier, E., Deems, J., Gutmann, E., Dehecq, A., Shean, D., and Dumont, M.: Snow depth mapping from stereo satellite imagery in mountainous terrain: evaluation using airborne laser-scanning data, The Cryosphere, 14, 2925–2940, 2020.

Doorschot, J. J. and Lehning, M.: Equilibrium saltation: mass fluxes, aerodynamic entrainment, and dependence on grain properties, Boundary-Layer Meteorology, 104, 111–130, https://doi.org/10.1023/A:1015516420286, 2002.

Gallée, H., Guyomarc'h, G., and Brun, E.: Impact of snow drift on the Antarctic ice sheet surface mass balance: possible sensitivity to snow-surface properties, Boundary-Layer Meteorology, 99, 1–19, https://doi.org/10.1023/A:1018776422809, 2001

Haddjeri, A., Baron, M., Lafaysse, M., Le Toumelin, L., Deschamp-Berger, C., Vionnet, V., Gascoin, S., Vernay, M., and Dumont, M.: Exploring the sensitivity to precipitation, blowing snow, and horizontal resolution of the spatial distribution of simulated snow cover, EGUsphere [preprint], https://doi.org/10.5194/egusphere-2023-2604, 2023.

Melo, D. B., Sharma, V., Comola, F., Sigmund, A., and Lehning, M.: Modeling snow saltation: the effect of grain size and interparticle cohesion, Journal of Geophysical Research: Atmospheres, 127, e2021JD035 260, https://doi.org/10.1029/2021JD035260, 2022.

Mott, R., Schirmer, M., Bavay, M., Grünewald, T., and Lehning, M.: Understanding snow-transport processes shaping the mountain snow-cover, The Cryosphere, 4, 545–559, https://doi.org/10.5194/tc-4-545-2010, publisher: Copernicus GmbH, 2010

Pomeroy, J. and Gray, D.: Saltation of snow, Water resources research, 26, 1583–1594, https://doi.org/10.1029/WR026i007p01583, 1990.

Sharma, V., Gerber, F., and Lehning, M.: Introducing CRYOWRF v1.0: multiscale atmospheric flow simulations with advanced snow cover modelling, Geosci. Model Dev., 16, 719–749, https://doi.org/10.5194/gmd-16-719-2023, 2023.

Sørensen, M.: On the rate of aeolian sand transport, Geomorphology, 59, 53–62, https://doi.org/10.1016/j.geomorph.2003.09.005, 2004.

Vionnet, V., Guyomarc'h, G., Bouvet, F. N., Martin, E., Durand, Y., Bellot, H., Bel, C., and Puglièse, P.: Occurrence of blowing snow events at an alpine site over a 10-year period: Observations and modelling, Advances in water resources, 55, 53–63, https://doi.org/10.1016/j.advwatres.2012.05.004, 2013.

Vionnet, V., Martin, E., Masson, V., Guyomarc'h, G., Naaim-Bouvet, F., Prokop, A., Durand, Y., and Lac, C.: Simulation of wind-induced snow transport and sublimation in alpine terrain using a fully coupled snowpack/atmosphere model, The Cryosphere, 8, 395–415,1075, https://doi.org/10.5194/tc-8-395-2014, 2014.

Vionnet, V., Marsh, C. B., Menounos, B., Gascoin, S., Wayand, N. E., Shea, J., Mukherjee, K., and Pomeroy, J. W.: Multi-scale snowdrift-1085 permitting modelling of mountain snowpack, The Cryosphere, 15, 743–769, https://doi.org/10.5194/tc-15-743-2021, 2021.

Liston, G. E. and Sturm, M.: A snow-transport model for complex terrain, Journal of Glaciology, 44, 498–516, https://doi.org/10.3189/S0022143000002021, 1998